# Bacterial immune activation via supramolecular assembly with phage triggers

Tong Zhang[1], Yifei Lyu[2], Christina R. Beck[1], Naseer Iqbal[2], Renee Barbosa[1], Alireza Ghanbarpour[2✉] & Michael T. Laub[1,3✉]

Bacteria use diverse mechanisms to protect themselves against phages[1–6]. Many antiphage systems form large oligomeric complexes, but how oligomerization is regulated during phage infection remains mostly unknown[7–12]. Here we demonstrate that the bacterial immunity protein ring-activated zinc-finger RNase (RAZR) assembles into an active, 24-meric ring around the circumference of large ring structures formed by two unrelated phage proteins: a putative recombinase and a portal protein. Each multi-layered, megadalton-scale complex enables RAZR to cleave RNA nonspecifically to inhibit translation and restrict phage propagation. The recognition of unrelated phage proteins that form rings with similar diameters indicates that these proteins not only bind to RAZR but also enforce a geometry crucial to activation. The lack of large ring structures in the host probably prevents auto-immunity and RAZR activation before infection. The infection-triggered oligomerization of RAZR mirrors pathogen-induced oligomerization in eukaryotic innate immune complexes[13], underscoring a common principle of immunity across biology.

A common theme in eukaryotic innate immunity is the assembly of large, higher-order oligomeric protein complexes[13]. Upon detection of pathogen-associated molecular patterns, innate immune sensors known as pattern recognition receptors can assemble into megadalton-sized supramolecular complexes along with other immune proteins, triggering activation of downstream immune responses[13]. The ability to oligomerize at multiple steps of immune signalling enables signal amplification above a response threshold, and the cooperativity in assembly allows the innate immune system to respond to pathogens in an all-or-none manner[13].

Bacteria also encode diverse innate immune systems to protect themselves against bacteriophage infection[1–6], but how they detect phages and become activated remains poorly understood. A handful of antiphage defence systems in bacteria oligomerize upon binding a small-molecule second messenger[14–16], but how phage infection initially triggers second messenger production remains unclear. Recently, many innate immune systems in bacteria have been shown to sense pathogen-associated molecular pattern-like proteins from phages. The antiviral STAND systems bind to phage-encoded proteins and then tetramerize to form active complexes, at least in vitro and following co-expression[17]. Structural studies indicate that many antiphage systems form oligomeric complexes of up to 10 MDa on their own[7–12], but the phage signal sensed by each is unclear.

Here we investigated an antiphage defence system called PD-T7-4 discovered through a functional metagenomic selection using diverse *Escherichia coli*[5]. We renamed this system RAZR based on the studies presented below. Homologues of RAZR are widespread across different bacterial phyla, including both Gram-negative and Gram-positive bacteria[5]. RAZR contains a predicted N-terminal zinc-finger domain (ZFD) and a C-terminal DUF4145 domain that is a divergent higher eukaryotes

and prokaryotes nucleotide-binding (HEPN) RNase[18]. HEPN domains typically dimerize to precisely position their catalytic residues at the dimer interface and form a composite active site[19]. RAZR functions through an abortive infection mechanism in which infected cells die or stop growing but prevent spread of virions through the bacterial population[5,20]. It is unknown how RAZR gets activated during phage infection and how it functions to restrict phage replication.

## RAZR is activated by multiple phage proteins

We focused on a RAZR system in *E. coli* UMB1727 isolated from a female urinary bladder[5]. Consistent with previous work, RAZR provided robust defence against phages T3, T7, SECΦ18 and SECΦ27 when expressed from its native promoter on a low-copy number plasmid in *E. coli* MG1655 (Fig. 1a and Extended Data Fig. 1a). To gain insight into its phage-defensive function, we used HHpred[21] to identify remote homology and AlphaFold3 (ref. 22) to predict the structure of RAZR (Fig. 1b,c). The N-terminal region of RAZR contains a ZFD closely related to Cys₄ zinc ribbons (Extended Data Fig. 1b). The C-terminal region includes a DUF4145 domain that is distantly homologous[18] yet structurally somewhat similar to HEPN proteins with a DALI $Z$-score of 8.7 (Extended Data Fig. 1c). In prokaryotes, HEPN domain-containing RNases are essential components of many phage-defensive elements, such as CRISPR–Cas systems, toxin–antitoxin systems and other newly discovered antiphage systems[5,6,19]. HEPN RNases typically contain an α-helical core and catalyse RNA cleavage through a conserved Rx4-6H motif[18,19]. We identified a conserved histidine residue (H154) and arginine residue (R146) in the DUF4145 domain of RAZR (Fig. 1c and Extended Data Fig. 1c,d) and found that alanine substitutions of these residues abolished defence against phage SECΦ27

[1]Department of Biology, Massachusetts Institute of Technology, Cambridge, MA, USA. [2]Department of Biochemistry and Molecular Biophysics, Washington University School of Medicine, St. Louis, MO, USA. [3]Howard Hughes Medical Institute, Massachusetts Institute of Technology, Cambridge, MA, USA. ✉e-mail: alirezag@wustl.edu; laub@mit.edu

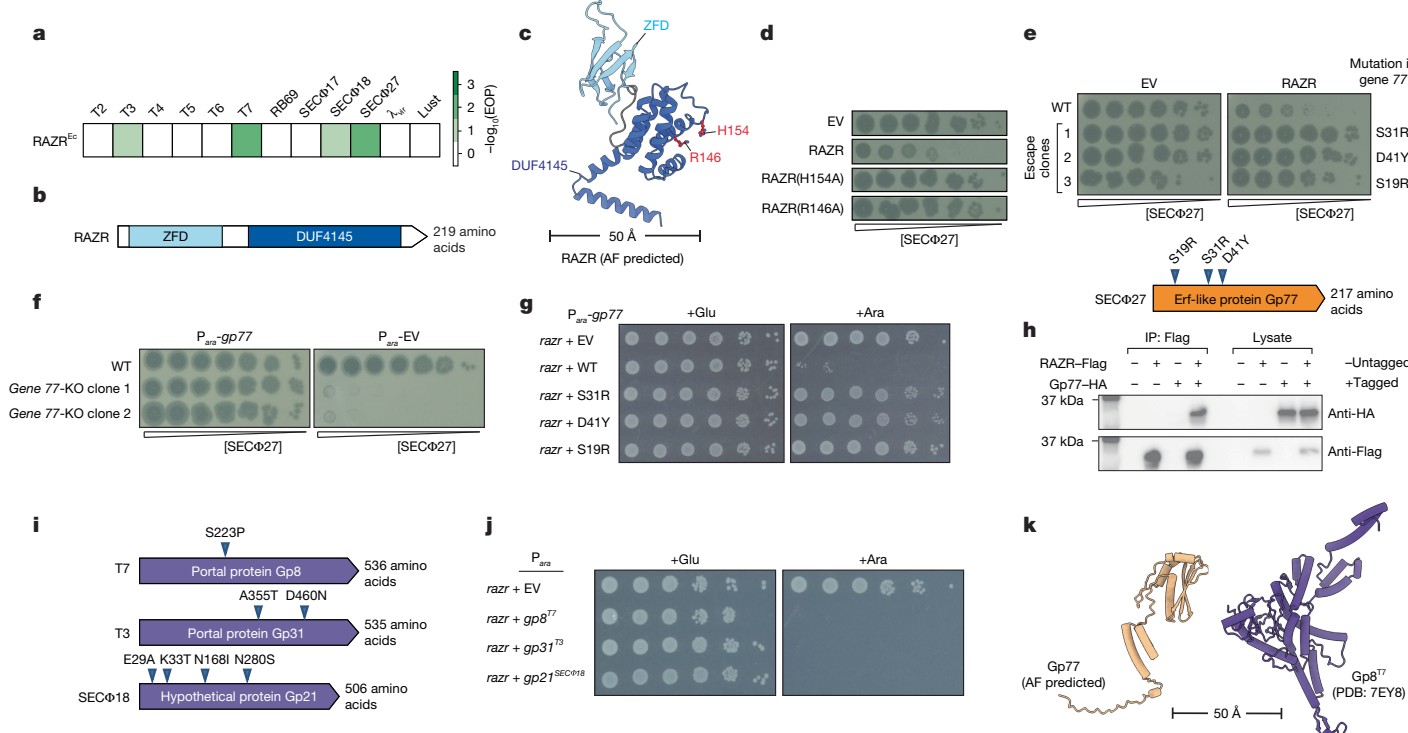

**Fig. 1 | RAZR is activated by multiple, unrelated phage proteins. a**, Efficiency of plaquing (EOP) data for the phages indicated when infecting cells producing RAZR[Ec]. **b**, Schematic of the domain organization of RAZR as predicted by HHpred. **c**, AlphaFold3 (AF3)-predicted structure of RAZR coloured by domains. The predicted active site residues R146 and H154 are highlighted. **d**, Serial tenfold dilutions of the SECΦ27 phage spotted on lawns of cells harbouring an empty vector (EV) or a plasmid producing RAZR (WT or the indicated variant). Relative phage concentration is indicated by the height of the wedge. **e**, Serial dilutions of three escape clones of SECΦ27 and the WT phage spotted on lawns of cells harbouring an empty vector or a RAZR expression vector (top). A schematic of Gp77 with the substitutions in the escape clones indicated is also shown (bottom). **f**, Serial dilutions of the SECΦ27 phage lacking gene 77 or the WT phage spotted on lawns of cells harbouring an empty vector or a plasmid expressing

Gp77 exogenously. KO, knockout. **g**, Cell viability assessed by serial dilutions of cells producing RAZR from its native promoter and the indicated variant of Gp77 from an arabinose-inducible promoter ($P_{ara}$) on media containing glucose (Glu) or arabinose (Ara). **h**, RAZR–Flag was immunoprecipitated from cells producing RAZR or RAZR–Flag and Gp77 or Gp77–HA and probed for the presence of Gp77 via the HA tag. Lysates used as input for the immunoprecipitation (IP) were probed as controls for expression levels. Representative of three biological replicates. **i**, Summary of the identified escape mutants of T7, T3 and SECΦ18. Substitutions map to the known or predicted portal proteins, respectively. **j**, Serial dilutions of cells producing RAZR from its native promoter and Gp8[T7], Gp31[T3] or Gp21[SECΦ18] from an arabinose-inducible promoter on media containing glucose or arabinose. **k**, AF3-predicted structure of Gp77 and the crystal structure of Gp8[T7].

(Fig. 1d and Extended Data Fig. 1e). On the basis of these results, and those below indicating that RAZR drives RNA cleavage, we concluded that the catalytic activity of the DUF4145 domain is important for the phage-defensive function of RAZR, and hereafter call it a HEPN domain.

Despite harbouring a HEPN domain and functioning through an abortive infection mechanism, RAZR was not toxic to *E. coli* when overproduced (Extended Data Fig. 1f). Thus, we hypothesized that RAZR must be activated by some phage factor (or factors) during infection. To identify these triggers, we isolated escape mutants of SECΦ27 that overcome RAZR defence (Fig. 1e and Extended Data Fig. 1g). All escape phage clones contained point mutations leading to a single amino acid substitution (S19R, S31R or D41Y) in Gp77, a hypothetical phage protein. Gp77 is conserved in phages related to SECΦ27 and encoded adjacent to a putative single-strand DNA-binding protein (Extended Data Fig. 1h). Although Gp77 has no annotated function in SECΦ27, we noted structural homology to the Erf (essential recombination function) protein from *Salmonella* phage P22 (Extended Data Fig. 1i), which is important for circularizing the phage genome upon entry into bacteria[23–25]. We hypothesized that Gp77 has a similar role in the phage SECΦ27 and is therefore essential to the phage. Consistent with this hypothesis, a variant of SECΦ27 lacking gene 77 cannot infect wild-type (WT) *E. coli* but can infect cells producing the Gp77 protein exogenously from a plasmid (Fig. 1f and Extended Data Fig. 1j).

To test whether Gp77 is an activator of RAZR, we co-produced WT or mutant variants of Gp77 with RAZR in the absence of phage infection. WT Gp77 rendered RAZR toxic, whereas each of the mutant variants had almost no effect on colony-forming units, although the S31R and D41Y variants each led to slightly smaller colonies (Fig. 1g and Extended Data Fig. 1k). As a control, we verified that neither the WT nor the mutant variants of Gp77 were toxic on their own or when co-produced with the catalytically inactive RAZR(H154A) (Extended Data Fig. 1l). To test whether Gp77 activates RAZR through a direct interaction, we immunoprecipitated RAZR–Flag from cells co-producing Gp77–HA and found that Gp77–HA co-precipitated with RAZR–Flag (Fig. 1h). These results indicate that Gp77 alone is sufficient to activate RAZR, probably through a direct interaction, and the escape phages overcome defence by preventing RAZR activation.

RAZR also defended against phages T3, T7 and SECΦ18 in a manner dependent on the catalytic activity of the HEPN domain (Fig. 1a and Extended Data Fig. 2a). T3, T7 and SECΦ18 belong to different phage families that are distantly related to SECΦ27 and do not encode homologues of Gp77. To identify triggers of RAZR in these phages, we isolated escape mutants of each phage that largely overcome RAZR defence (Extended Data Fig. 2b). In each case, the escape phages contained a missense mutation in the gene encoding the known portal protein of the phage (Gp8 in T7 and Gp31 in T3), or a hypothetical protein, Gp21, in SECΦ18 that has structural homology to the portal proteins

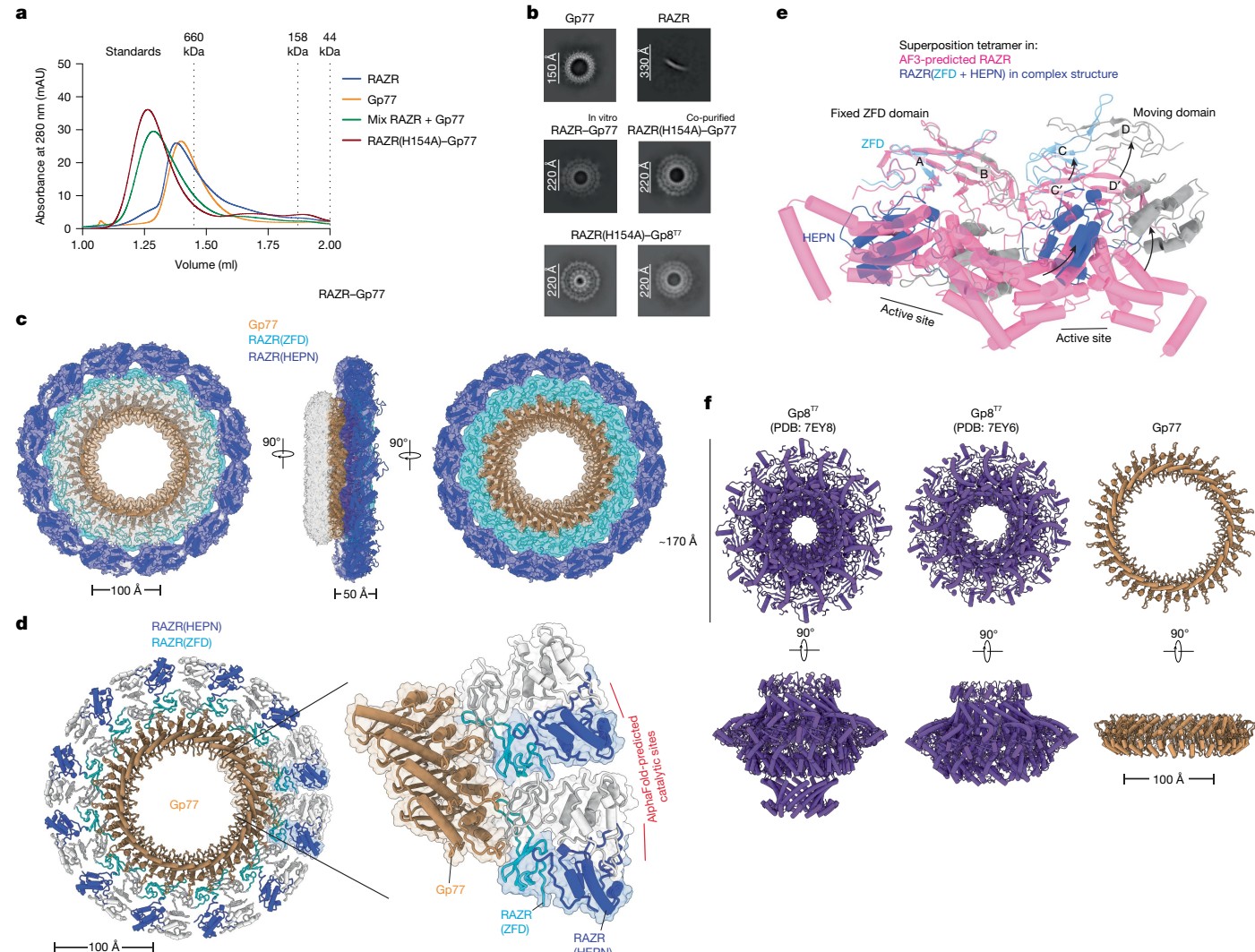

**Fig. 2 | RAZR assembles around a ring scaffold to be activated. a**, Analytical size-exclusion chromatography analysis of purified RAZR, Gp77, mixing RAZR and Gp77 at a 1:1 molar ratio, and a co-purified RAZR(H154A)–Gp77 complex. Molecular weight standards are labelled. **b**, Sample 2D class averages of RAZR, Gp77, RAZR–Gp77 reconstituted in vitro, the co-purified RAZR(H154A)–Gp77 complex and the RAZR(H154A)–Gp8$^{T7}$ complex. **c**, The cryo-EM density map of the in vitro-reconstituted RAZR–Gp77 complex coloured by corresponding protein domains, and the fitted atomic models are shown in cartoon representation. **d**, Cartoon representation of the 24-meric Gp77 ring in contact with 24 RAZR (left), and a detailed view of 4 protomers of Gp77 and 4 protomers

of RAZR, coloured by domains (right). The active sites of RAZR(HEPN) are labelled in red. **e**, Superposition of the AF3-predicted RAZR tetramer alone (pink) and the structure of four protomers of RAZR when in complex with Gp77 (coloured by domains) suggest a relative movement of the adjacent RAZR dimer—and consequently of the HEPN domains (protomers C and D)—when one ZFD dimer (protomers A and B) is overlaid. The arrows indicate the relative movement. **f**, Top and side views of the two structures of Gp8$^{T7}$ corresponding to the open (PDB ID: 7EY6) and closed (PDB ID: 7EY8) conformations, and the structure of Gp77.

(Fig. 1i and Extended Data Fig. 2c). Each of these proteins led to cellular toxicity when co-produced with RAZR (Fig. 1j) and was not toxic on its own or with the catalytically inactive RAZR(H154A) (Extended Data Fig. 2d), indicating that each is sufficient to activate RAZR. In addition, the portal proteins from T7, T3 and SECΦ18 each co-precipitated with RAZR(H154A)–Flag when both proteins were co-produced in cells (Extended Data Fig. 2e), suggesting a direct interaction. Portal proteins are essential and conserved structural elements of phages, having critical roles in genome packaging and injection during infection[26]. Despite substantial sequence variability, the portal proteins from these diverse phages have similar predicted structures (Extended Data Fig. 2c). Of note, phage SECΦ27 encodes a structurally similar portal protein, but it was unable to activate RAZR when both proteins were co-produced (Extended Data Fig. 2f). This observation is consistent with our finding above that mutations in gene *77* alone allowed SECΦ27 to fully overcome RAZR defence.

Our results demonstrate that Gp77 in SECΦ27 as well as the portal proteins in phages T3, T7 and SECΦ18 are each necessary and sufficient to activate RAZR. Gp77 and the portal proteins do not show any sequence homology, and they have significantly distinct monomeric structures (Fig. 1k). Together, our results demonstrate that RAZR can be activated by multiple, structurally unrelated proteins in different phages.

## RAZR assembles around a ring scaffold

To investigate how RAZR is triggered by different proteins, we expressed and purified RAZR and its activator Gp77 from SECΦ27. Size-exclusion chromatography showed that RAZR–His$_6$ and Gp77–His$_6$ independently formed large oligomeric assemblies approximately 660 kDa (Fig. 2a), whereas each monomer is only 25 kDa. To determine whether RAZR and Gp77 interact in vitro, we mixed the purified proteins at

a 1:1 ratio and observed the formation of a larger-molecular-weight species (Fig. 2a). We also co-produced Gp77–HA with the catalytically inactive RAZR(H154A)–His$_6$ in cells to prevent toxicity induced by active RAZR, and then purified RAZR(H154A)–His$_6$ and its associated proteins through Ni$^{2+}$-affinity chromatography. We observed a similar high-molecular-weight species containing both proteins (Fig. 2a and Extended Data Fig. 3a), and mass photometry analysis revealed a dominant species of approximately 1.4 MDa (Extended Data Fig. 3b), indicating a large supramolecular complex.

To gain insight into RAZR activation, we performed cryo-electron microscopy (cryo-EM) analysis of individually purified proteins (RAZR and Gp77), as well as the inactive RAZR(H154A)–Gp77 complex obtained by co-expression and the in vitro-reconstituted RAZR–Gp77 complex. 2D class averages revealed that Gp77 alone formed a 24-fold symmetric ring (Fig. 2b and Extended Data Fig. 3c). Mass photometry indicated a main species of approximately 600 kDa, consistent with 24 copies of Gp77 monomer (Extended Data Fig. 3d). This structure resembles the ring-like structure of Erf, a homologue from phage P22 implicated in phage genome circularization[27]. By contrast, the oligomeric RAZR adopted a linear conformation with variable lengths (Fig. 2b and Extended Data Fig. 3e,f). AlphaFold3 multimer prediction similarly suggested linear RAZR tetramers formed through inter-dimer contacts (Extended Data Fig. 3g,h). The co-purified complex of RAZR(H154A)–Gp77, or the complex formed by mixing separately purified proteins, also formed a ring-like structure (Fig. 2b and Extended Data Fig. 3i). However, unlike Gp77 alone, the complex contained three concentric rings (Fig. 2b). On the basis of these results, we hypothesized that RAZR forms linear oligomers that are inactive before infection. During SECΦ27 phage infection, RAZR assembles around the Gp77 ring scaffold to form a complex, leading to RAZR activation.

We next determined cryo-EM maps of both co-purified RAZR(H154A)–Gp77 and the in vitro-reconstituted RAZR–Gp77 complexes (Fig. 2c,d, Extended Data Table 1, Extended Data Figs. 4 and 5 and Supplementary Figs. 2 and 3). Both reconstructions revealed a symmetric, multi-layered ring approximately 270 Å in diameter, featuring a 24-fold inner ring surrounded by 12-fold outer rings (Fig. 2c,d).

Focused refinements improved the resolution of the inner and central rings (Extended Data Figs. 4 and 5 and Supplementary Fig. 3). The inner ring density was fitted with 24 copies of the AlphaFold-predicted Gp77 N-terminal domain (residues 1–125), supported by well-resolved density for bulky side chains (Extended Data Fig. 6a). The last modelled residue of Gp77, 125, pointed upwards at the complex apex, indicating that the C-terminal region (residues 126–217) corresponds to peripheral density lacking contacts with other subunits (Fig. 2c and Extended Data Fig. 6b). Deleting the Gp77 C-terminal domain did not affect RAZR activation (Extended Data Fig. 6c), demonstrating its peripheral role.

The central ring resolved 12 RAZR dimers, each containing the ZFD domain of the AlphaFold model (Fig. 2c,d and Extended Data Fig. 6d). These domains faced inwards to contact Gp77, implying that the HEPN domains extend outwards to occupy the outer ring. To test this orientation, we incubated anti-His antibody with the RAZR(H154A)–His$_6$–Gp77–HA complex and observed antibody density specifically at the outermost ring—where the C-terminal His tag of the HEPN domain should reside—whereas no such density appeared in controls (Extended Data Fig. 6e–g), supporting the proposed arrangement (Fig. 2c,d and Extended Data Fig. 6h). Although the HEPN domains were resolved at lower resolution, docking of the AlphaFold-predicted model revealed a canonical HEPN dimer-like arrangement (Extended Data Fig. 6i), where juxtaposition of monomers forms a composite RNase active site essential for cleavage activity[19,28]. However, unlike canonical HEPN RNases, RAZR further oligomerized through interactions between adjacent dimers and formed higher-order oligomers, both in the absence and in the presence of its activator (Fig. 2a,b).

Although the HEPN domain lacked sufficient resolution to unambiguously define the conformations of the active-site residues (R146

and H154), the map still revealed the relative orientation between the HEPN and ZFD domains. Comparison of the AlphaFold-predicted linear RAZR tetramers with the RAZR proteins in our complex suggested a relative movement of the adjacent ZFD dimer, and consequently the HEPN domains (protomers C and D), resulting in a linear-to-curved transition (Fig. 2e), which might be important for HEPN activation.

As RAZR can also recognize the portal proteins from T3, T7 and SECΦ18 phages, we co-purified RAZR(H154A) with the portal protein Gp8[T7] and analysed this complex by cryo-EM. 2D class averages revealed two species that adopted multi-layered 12-fold symmetric rings (Fig. 2b and Extended Data Fig. 6j). Gp8[T7] forms a dodecameric ring, and can adopt two conformations corresponding to the states before and after phage genome ejection[29] (Fig. 2f). Although a high-resolution complex structure could not be achieved, comparison with existing Gp8[T7] structures supports the model that RAZR assembles around these portal rings, as with Gp77 (Fig. 2f). Of note, the Gp8[T7] and Gp77 rings have nearly identical diameters of approximately 170 Å (Fig. 2f). Thus, we propose that RAZR activation is triggered by recognizing protein rings of a characteristic size, enabling it to sense structurally distinct phage proteins that share this geometric feature.

## Interface analysis of the RAZR–Gp77 complex

To further understand how RAZR senses its phage-encoded triggers, we examined the interaction interface between RAZR and Gp77. In our cryo-EM structure of the complex, the ZFDs from two adjacent RAZRs dimerize and interact with Gp77 (Fig. 3a and Extended Data Fig. 7a). Although the resolution was insufficient to resolve side-chain density for all interface residues, the structural model places R35 of one ZFD (protomer A) near D53 and E55 of Gp77, suggesting a potential salt bridge between proteins (Extended Data Fig. 7b). Residues R29, Y31, T41 and E43 of the other ZFD (protomer B) probably contribute to the interface via electrostatic interactions and hydrogen bonding with Gp77 (Fig. 3a and Extended Data Fig. 7a,b). To test the importance of these residues in RAZR ZFD in recognizing Gp77, we made single amino acid substitutions in RAZR and then co-produced with Gp77. The single substitutions R29A, R29E, Y31A, R35E, E43R and T41D each reduced or abolished RAZR activation (Extended Data Fig. 7c), and each substantially weakened RAZR defence against SECΦ27 (Fig. 3b and Extended Data Fig. 7d). In addition, these RAZR variants either no longer co-precipitated with Gp77 or exhibited reduced binding, further supporting the importance of these residues in interacting with Gp77 (Fig. 3c). Of note, three of the substitutions (Y31A, E43R and T41D) still allowed RAZR to defend against another phage, SECΦ18, for which the portal protein is the trigger, whereas the other substitutions (R29A, R29E and R35E) abolished defence (Fig. 3d). These results indicate that the binding interfaces between RAZR and Gp77 or the portal protein probably overlap, but also involve different residues.

Our escape mutants of phage SECΦ27 each contained a single amino acid substitution (S19R, S31R and D41Y) in Gp77 and fully escaped RAZR defence (Fig. 1e). To understand how these substitutions enabled phage escape, we mapped residues S19, S31 and D41 onto our complex structure (Fig. 3e). Residue S19 of Gp77 is buried in the interface, probably hydrogen bonding with T41 of RAZR (Extended Data Fig. 7b). The substitution S19R probably leads to a loss of the hydrogen bond and introduces an unfavourable charge that disrupts the binding interface. Consistent with this hypothesis, Gp77(S19R) almost completely lost toxicity when co-produced with RAZR (Fig. 1g and Extended Data Fig. 1k), and no longer co-precipitated RAZR(H154A) upon induction (Fig. 3f and Extended Data Fig. 7e). Residues S31 and D41 are located in a loop of Gp77 formed by residues 19–41 that faces towards the ZFD of RAZR and probably contacts the ZFD through residues S19, L20 and V22 (Fig. 3e and Extended Data Fig. 7f). Substitutions in S31 and D41 may restructure the loop such that it no longer

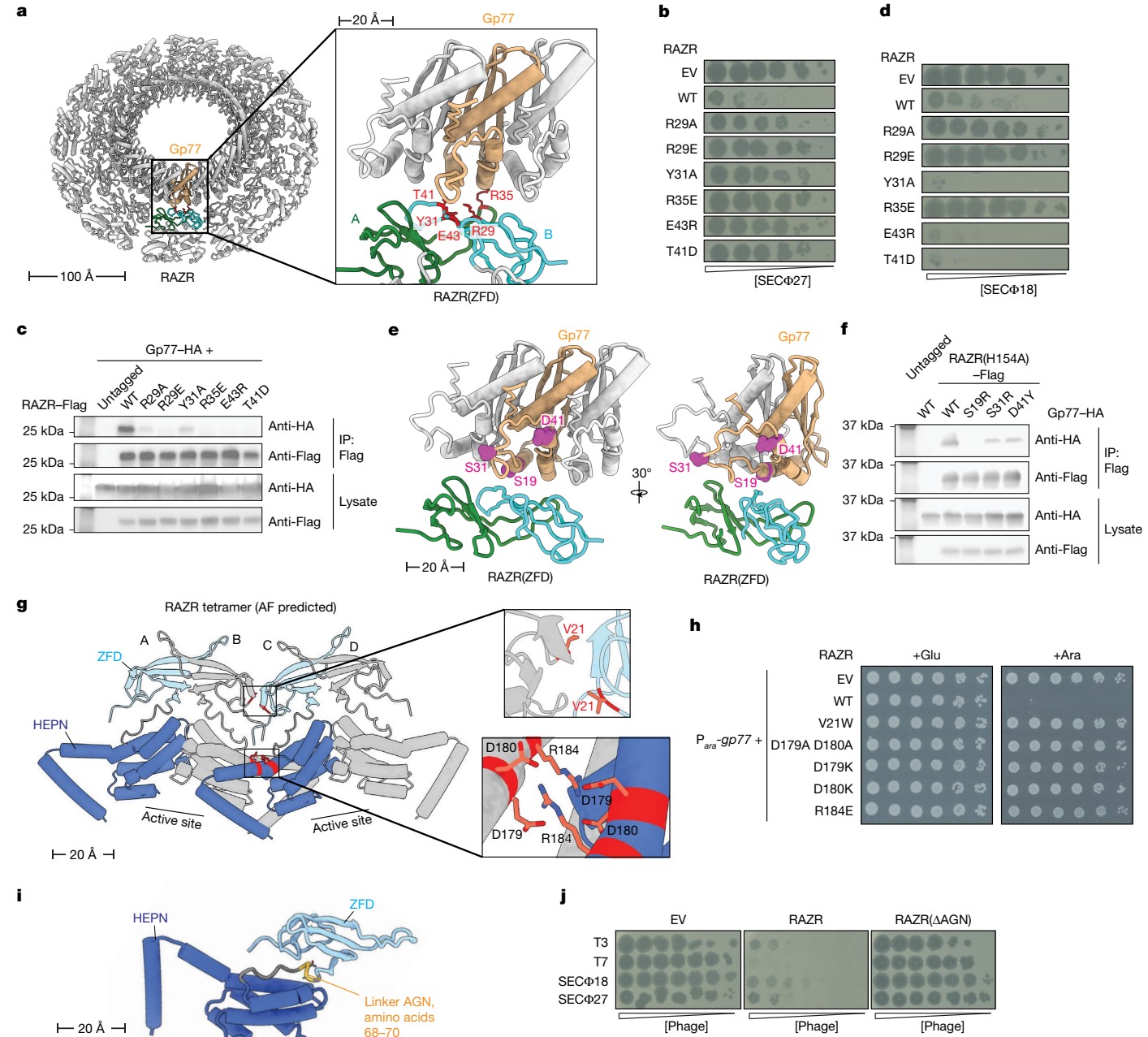

**Fig. 3 | Structural analysis of the RAZR–Gp77 complex. a**, Cartoon representation of Gp77 interacting with the ZFD of RAZR as in the structure of the RAZR–Gp77 complex (left), and a detailed interface of Gp77 (orange) and two protomers of RAZR(ZFD) (green and cyan; right). Substituted residues in RAZR(ZFD) are highlighted in red. **b**, Serial dilutions of the SECΦ27 phage spotted on lawns of cells harbouring an empty vector or a plasmid producing the WT or the indicated variant of RAZR. **c**, RAZR–Flag or the indicated variant was immunoprecipitated from cells producing RAZR–Flag and Gp77–HA and probed for the presence of Gp77 via the HA tag. Representative of two biological replicates. **d**, Serial dilutions of SECΦ18 phage spotted on lawns of cells harbouring an empty vector or a plasmid producing the WT or the indicated variant of RAZR. **e**, Interface of Gp77 (orange) and two protomers of RAZR(ZFD) (green and cyan) as in the complex structure of RAZR–Gp77.

Residues in Gp77 identified in escape phage mutants are in pink. **f**, RAZR(H154A)–Flag was immunoprecipitated from cells producing RAZR(H154A)–Flag and Gp77–HA (WT or the indicated variant) and probed for the presence of Gp77 via the HA tag. Representative of three biological replicates. **g**, AF3-predicted RAZR tetramer coloured by chains and by domains (left), and detailed interfaces of ZFDs (top) or HEPN domains (bottom) in between promoters B and C. Residues substituted are in red. **h**, Serial dilutions of cells producing the WT or the indicated variant of RAZR from its native promoter and Gp77 from an arabinose-inducible promoter on medium containing glucose or arabinose. **i**, AF3-predicted structure of RAZR coloured by domains. Residues AGN at positions 68–70 in the linker region are in orange. **j**, Serial dilutions of the indicated phage spotted on lawns of cells harbouring an empty vector or a plasmid producing the WT or the ΔAGN variant of RAZR.

binds to RAZR as efficiently, diminishing RAZR activation. Indeed, the S31R and D41Y escape variants of Gp77 had reduced binding to RAZR (Fig. 3f and Extended Data Fig. 7e), and co-production of these variants with RAZR led to the same colony-forming units as the empty vector control, although with slightly smaller colonies (Fig. 1g and Extended

Data Fig. 1k), indicating that these variants mostly abolished RAZR activation. In addition, we purified each of these Gp77 variants (S19R, S31R and D41Y) and observed by size-exclusion chromatography that they form large oligomers of similar size as the WT (Extended Data Fig. 7g).

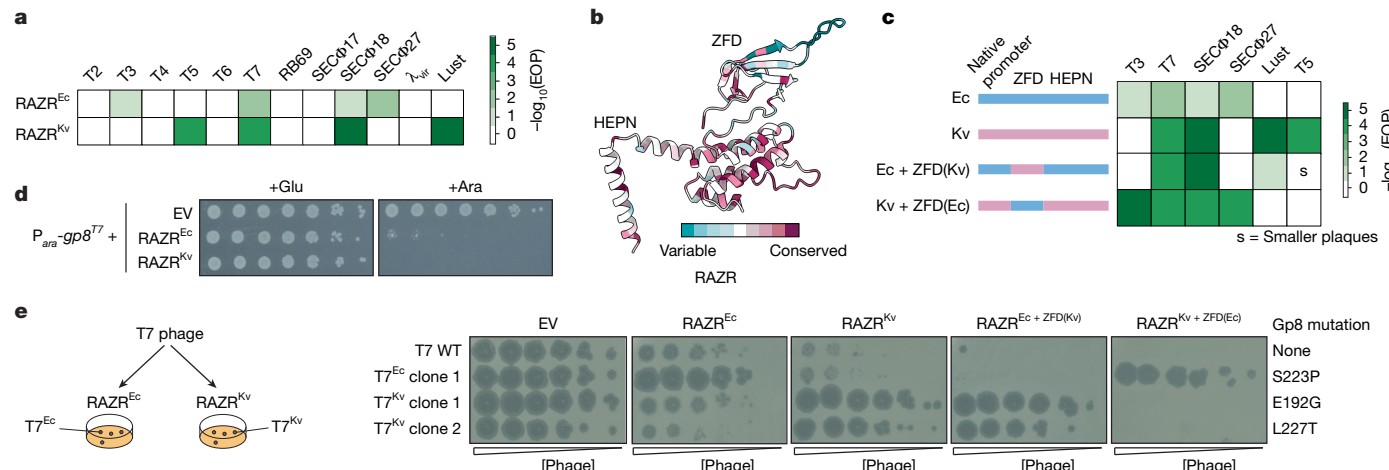

**Fig. 4 | The ZFD of RAZR confers specificity of antiphage defence. a**, EOP data for the phages indicated when infecting cells producing RAZR[Ec] or RAZR[Kv]. **b**, AF3-predicted structure of RAZR coloured by the conservation score of each amino acid calculated by ConSurf. **c**, Schematic of the RAZR constructs (left), and EOP data for the phages indicated when infecting cells producing RAZR[Ec], RAZR[Kv] or the corresponding RAZR chimeras. **d**, Serial dilutions of cells producing RAZR[Ec] or RAZR[Kv] from its native promoter and Gp8[T7] from an arabinose-inducible promoter on medium containing glucose or arabinose. **e**, Schematic of isolating T7 escape mutants of RAZR[Ec] or RAZR[Kv] (left), and serial dilutions of the indicated phage spotted on lawns of cells harbouring an empty vector or a plasmid producing RAZR[Ec], RAZR[Kv] or the corresponding RAZR chimeras (right). Substitutions in Gp8 of each escape mutant are indicated on the right.

To further validate our structure and test the importance of the loop of Gp77 formed by residues 19–41, we made substitutions in residues L20 and V22 that face outwards (Extended Data Fig. 7f). The single substitutions L20R and V22R in Gp77 mostly abolished RAZR activation (Extended Data Fig. 7h), and their expression allowed infection by a variant of SECΦ27 lacking gene *77* (Extended Data Fig. 7i), which indicates that these variants are not trivially unfolded and they support the phage-related function of Gp77, similar to the escape variants. Together, our interface analyses support a model in which the ZFD of RAZR serves as an infection sensor, and binding to the phage-encoded trigger Gp77 via the ZFD leads to RAZR activation.

## RAZR oligomerization is functionally important

Canonically, HEPN domain-containing RNases function as dimers, with each monomer contributing to a composite RNase active site[19]. Here we found that RAZR not only dimerized but also formed larger oligomers, before and during phage infection (Fig. 2a,b). To test whether the higher-order oligomerization is important for RAZR function, we identified residues that mediate interactions between adjacent dimers. According to the AlphaFold-predicted model of RAZR tetramers, the HEPN domains of adjacent dimers interacted extensively with each other in a linear manner (protomers B and C in Fig. 3g). Residues D179, D180 and R184 from each protomer probably form salt bridges and stabilize a helix–helix interaction (Fig. 3g). The ZFDs further contributed to the dimer–dimer interface via hydrophobic interactions involving residue V21 (Fig. 3g). We made substitutions of these residues that either changed the charges (D179A D180A, D179K, D180K and R184E) or introduced a bulky residue (V21W). We found that these substitutions each reduced or abolished the toxicity of RAZR in the presence of Gp77 (Fig. 3h), and mostly abolished RAZR defence against SECΦ27 (Extended Data Fig. 8a). As a control, we verified that these RAZR variants each accumulated in cells to levels similar to the WT protein, indicating that they were not trivially unfolded or degraded (Extended Data Fig. 8b). In addition, we purified RAZR(D179A D180A) and found by size-exclusion chromatography that it ran as a species with lower molecular weight than the WT and probably corresponds to smaller oligomers (Extended Data Fig. 8c). This result is consistent with our hypothesis that these substitutions disrupt the higher-order oligomerization seen in WT RAZR,

and that higher-order oligomerization of RAZR is important for its activation.

To explore how the ZFD and HEPN domains of RAZR are coordinated, we focused on the linker (residues 66–79) connecting these two domains. This linker is mostly conserved across diverse RAZR homologues (Extended Data Fig. 1d), and packs against the helices in the HEPN domain (Fig. 3i). To assess the importance of the linker, we deleted three non-conserved residues (amino acids AGN at positions 68–70) in the linker region and tested the function of this variant. RAZR(ΔAGN) failed to form a complex when mixed with purified Gp77 (Extended Data Fig. 8d), and it no longer defended against phages (Fig. 3j and Extended Data Fig. 8e). We verified that this variant was expressed at levels comparable with the WT protein in cells (Extended Data Fig. 8f) but probably formed smaller oligomeric species than the WT protein (Extended Data Fig. 8g), indicating that the linker region is also important for higher-order oligomerization of RAZR. Together, our analyses indicate that higher-order oligomerization of RAZR and the linker connecting its two domains are critical for activation and phage defence.

## The RAZR ZFD confers defence specificity

Our structural analyses indicated that the ZFD of RAZR is critical for binding its phage triggers; thus, we hypothesized that the ZFD determines the specificity of phage defence. To test this hypothesis, we identified another RAZR homologue from *Klebsiella variicola* (RAZR[Kv]), expressed it from its native promoter in *E. coli* MG1655, and challenged with various coliphages. RAZR[Kv] defended against phages T5, T7, SECΦ18 and Lust, which is overlapping but different than the protection profile of the *E. coli* homologue (RAZR[Ec]; Fig. 4a). The two RAZRs share 78% overall amino acid identity, but are substantially more different within their ZFDs (Extended Data Fig. 9a). The ZFD is the most variable region across diverse RAZR homologues (Fig. 4b and Extended Data Fig. 1d). We made chimeras of RAZR[Ec] and RAZR[Kv] by swapping their ZFDs and found that the chimeras largely inverted defence profiles (Fig. 4c and Extended Data Fig. 9b), demonstrating the importance of the ZFD in determining defence specificity.

Despite their differences in phage defence, both RAZR[Ec] and RAZR[Kv] defended against phage T7, and both were activated by its portal protein Gp8[T7] (Fig. 4a,d). To explore whether there exists any specificity while recognizing the same trigger, we looked for escape mutants of T7 that

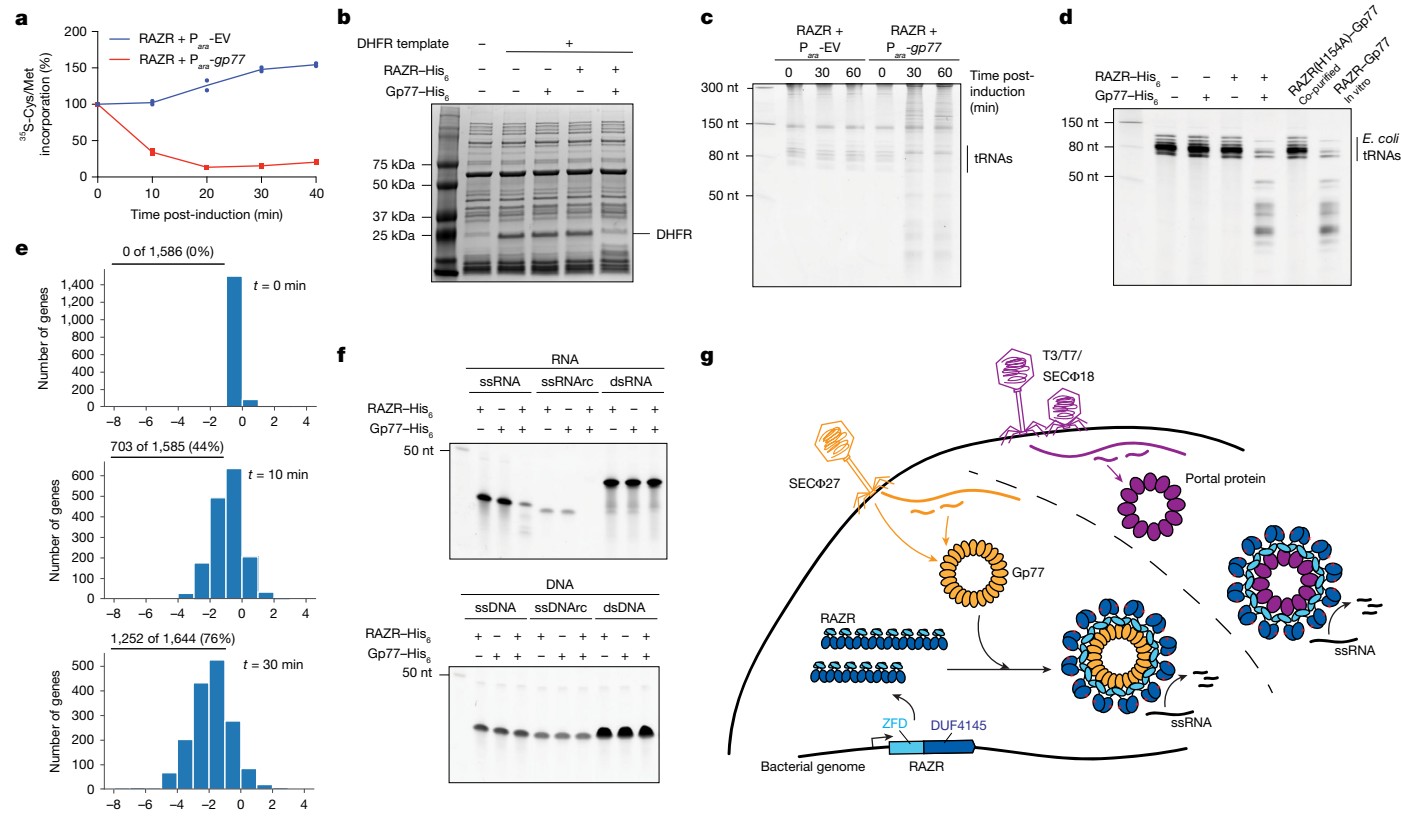

**Fig. 5 | RAZR inhibits translation by cleaving single-stranded RNA. a**, Cells producing RAZR from its native promoter and Gp77 from an arabinose-inducible promoter or an empty vector were pulse labelled with $^{35}$S-Cys/Met at the times indicated post-addition of arabinose. **b**, In vitro transcription–translation assays using DHFR production from a DNA template as the readout, with purified RAZR–His$_6$ and Gp77–His$_6$ added to the reactions. Representative of three biological replicates. **c**, RNAs were isolated from cells producing RAZR from its native promoter and Gp77 from an arabinose-inducible promoter or an empty vector at the times indicated post-addition of arabinose, and resolved on TBE–urea gels to visualize tRNAs. Representative of three biological replicates. **d**, Isolated bulk *E. coli* tRNAs were incubated with purified RAZR–His$_6$ and Gp77–His$_6$, RAZR–Gp77 reconstituted in vitro or the co-purified RAZR(H154A)–Gp77 complex, and visualized on TBE–urea gels. Representative of three biological replicates. **e**, Histograms showing the distributions of the minimum cleavage ratios within well-expressed coding regions in *E. coli*. RNA-seq was conducted for cells producing RAZR from its native promoter and Gp77 from an arabinose-inducible promoter or an empty vector at the times indicated post-addition of arabinose. The number of genes with a minimum cleavage ratio ≤ −1 and the total number of genes above the expression threshold at each time point are labelled. **f**, Nucleic acid substrates (single-stranded RNA (ssRNA) or DNA (ssDNA) or double-stranded RNA (dsRNA) or DNA (dsDNA)) were incubated with purified RAZR–His$_6$ and Gp77–His$_6$, and visualized on TBE–urea gels. Representative of two biological replicates. rc, reverse complement. **g**, Model for the activation of RAZR by Gp77 from SECΦ27 or the portal proteins from T3, T7 and SECΦ18.

either overcome RAZR$^{Ec}$ or RAZR$^{Kv}$ defence by selecting for escape from each system individually. Single amino acid substitutions in Gp8$^{T7}$ allowed T7 to escape each defence system: S223P allowed escape of RAZR$^{Ec}$, and E192G or L227T allowed escape of RAZR$^{Kv}$ (Fig. 4e, Extended Data Fig. 9c and Supplementary Table 1). These residues (S223, E192 and L227) were in the same region of Gp8$^{T7}$, consistent with this region being important for activating RAZR (Extended Data Fig. 9d). However, the escapers selected against one RAZR did not overcome defence by the other homologue, suggesting there is specificity to these phages by each RAZR (Fig. 4e and Extended Data Fig. 9c). This specificity depended on the ZFDs, as the chimeras with the swapped ZFDs had inverted defence phenotypes (Fig. 4e and Extended Data Fig. 9c). Thus, although RAZR$^{Ec}$ and RAZR$^{Kv}$ both defend against the same phage, the two proteins differ in how they recognize the phage trigger via their ZFDs, leading to additional specificity.

## RAZR cleaves single-stranded RNA to inhibit translation

Once activated during phage infection, RAZR functions as an abortive infection system, where infected cells die and prevent spread of phages through the bacterial population[20]. To examine the molecular basis of RAZR toxicity following activation, we co-produced RAZR with its trigger Gp77, and measured the effect on replication, transcription and translation by pulse labelling with radioisotopes specific for each process. Activated RAZR robustly inhibited translation in cells (Fig. 5a), but had no effect on transcription or DNA replication (Extended Data Fig. 10a). We then tested whether RAZR can inhibit translation in a reconstituted in vitro transcription–translation system. Incubating purified RAZR–His$_6$ with Gp77–His$_6$ strongly inhibited the synthesis of a model protein DHFR, whereas adding each individual protein had no effect (Fig. 5b).

Because RAZR contains a HEPN-like DUF4145 domain and mutating its predicted active site abolished phage defence (Fig. 1d), we hypothesized that RAZR inhibits translation by cleaving RNAs. To test this idea, we extracted total RNAs from cells co-producing RAZR and Gp77, and analysed tRNAs and rRNAs on denaturing gels. Upon Gp77 induction, there appeared to be smaller-sized products that probably correspond to cleaved products of tRNAs (Fig. 5c) and rRNAs (Extended Data Fig. 10b). Similar patterns were seen when RAZR-containing cells were infected with phage T7, indicating cleavage of tRNAs and rRNAs during infection (Extended Data Fig. 10c,d). To test whether RAZR cleaves tRNAs, we

incubated purified RAZR–His₆ and Gp77–His₆ with bulk *E. coli* tRNAs, and observed multiple small fragments less than 50 nt, probably indicating cleavage of multiple tRNA species (Fig. 5d). Similarly, we incubated purified proteins with *E. coli* 70S ribosomes and saw evidence of rRNA cleavage (Extended Data Fig. 10e). Cleavage of tRNAs or rRNAs was also observed with the in vitro-reconstituted RAZR–Gp77 complex, but not with the co-purified RAZR(H154A)–Gp77 complex that contains the catalytically inactive RAZR (Fig. 5d and Extended Data Fig. 10e).

To test whether RAZR can cleave mRNAs, we performed RNA sequencing (RNA-seq) on cells expressing RAZR from its native promoter and Gp77 from an inducible promoter or an empty vector as a control. We induced Gp77 for 10 or 30 min, and sequenced the RNAs extracted. We counted the number of RNA-seq reads across each nucleotide in the genome, and calculated a 'cleavage ratio' ($\log_2$ of read counts in Gp77 sample/empty vector)[30]. For all genes above our expression threshold, 44% (10 min) or 76% (30 min) had a minimum cleavage ratio ≤ −1 (Fig. 5e), indicating widespread cleavage of mRNAs following RAZR activation. To corroborate the RNA-seq data, we in vitro-transcribed eight mRNA targets and incubated them with purified RAZR and Gp77. We observed extensive cleavage of each mRNA substrate that led to a wide range of smaller products (Extended Data Fig. 10f). Together, these results indicate that RAZR cleaves diverse mRNAs upon activation.

To examine the substrates of RAZR during phage infection, we similarly performed RNA-seq on phage T7-infected cells either producing RAZR from its native promoter or harbouring an empty vector. At 20 min and 30 min post-infection, several T7 mRNAs had a minimum cleavage ratio ≤ −1, indicating cleavage of these transcripts during infection (Extended Data Fig. 10g,h). In addition, we also observed cleavage of 16S and 23S rRNAs at 30 min post-infection by phage T7 (Extended Data Fig. 10i). Combined with our denaturing gel analysis (Extended Data Fig. 10c,d), we conclude that RAZR cleaves mRNAs, tRNAs and rRNAs during phage infection.

Given that RAZR can cleave mRNAs, tRNAs and rRNAs, we further tested its specificity and preference towards different nucleic acid substrates (single-stranded or double-stranded RNA or DNA). Incubating purified RAZR–His₆ and Gp77–His₆ with a 24-nt single-stranded RNA oligo or its reverse complement both led to substantial cleavage, whereas no effect was seen for the annealed double-stranded RNA (Fig. 5f). No cleavage was observed for single-stranded DNA or double-stranded DNA oligos of the same sequence (Fig. 5f). Together, our results demonstrate that activated RAZR cleaves mRNAs, tRNAs and rRNAs to inhibit translation, probably through nonspecifically cleaving single-stranded RNA.

## Discussion

The formation of large, high-order macromolecular complexes is a common theme in eukaryotic immunity[13]. Such oligomeric structures may be crucial in enabling cells to rapidly respond to pathogens in an all-or-none manner. Our work demonstrates that RAZR, a bacterial immunity protein, forms a supramolecular protein complex of over 1 MDa upon recognition of its viral triggers, either the 24-meric putative recombination protein Gp77 of phage SECΦ27 or the dodecameric portal proteins of phages T3, T7 and SECΦ18 (Fig. 5g). The ability to oligomerize into high-order complexes may be a common feature of many bacterial defence systems. Avs3 and Avs4 bind to phage-encoded proteins as monomers and then tetramerize to form active complexes, although activation has not been examined during infection[17]. Avs5 forms large filaments that are critical for defence, but the phage signal regulating filamentation is unclear[31]. Other antiphage defence systems including DUF4297–HerA[8], SIR2–HerA[7,9], RADAR[10], Septu[11] and Gabija[12,32,33] also form large supramolecular complexes, but how these systems sense phage infection and how this regulates their oligomerization remains mostly unknown.

Of note, RAZR can sense two distinct phage proteins: Gp77 and the portal protein. The ability to recognize two completely unrelated phage proteins has been seen previously with other defence systems, including CapRel^SJ46[46] and PARIS[34,35], and enables protection against a broader set of phages. Of note, despite the lack of sequence or structural similarity between Gp77 and portal proteins at the monomeric level, each forms a large, similarly sized ring-shaped structure. Sensing a large ring of phage proteins may promote the specificity of RAZR activation and avoid autoimmunity, as our host strain (*E. coli* MG1655) only encodes one such ring before infection, the flagellar basal body, but it has a larger diameter of 260 Å (ref. 36). Although the geometry of the Gp77 and portal rings are probably crucial to the activation of RAZR by promoting the proper positioning of the HEPN domains, RAZR also makes specific amino acid contacts with each activator protein as we found individual substitutions that could disrupt binding and activation. Gp77 and portal proteins do not share sequence similarity, indicating that RAZR probably uses distinct sets of amino acids to recognize each trigger protein. The interaction of each RAZR monomer with a Gp77 or portal monomer may be relatively weak, with oligomerization promoting overall stability and activity.

Our results strongly suggest that the binding of RAZR to its phage-encoded ring-shaped scaffolds repositions the active site formed by adjacent monomers to promote RNase activity. Dimerization is a common property of HEPN RNases. Unlike canonical HEPNs, RAZR proteins further oligomerize into higher-order structures, and this high-order oligomerization is critical for RAZR activation (Fig. 3g–j). Following phage infection, RAZR probably transitions from linear to curved oligomers. The two unrelated phage triggers form rings with nearly identical diameters, which implies that the degree of curvature of the ring scaffolds is crucial for properly positioning and activating the RNase domains of RAZR, but additional biochemical and structural studies are needed to elucidate the precise activation mechanism.

In recent years, dozens of antiphage defence systems have been found, most of which function as abortive infection systems that must be tightly regulated to ensure they are only activated following phage infection. Our work demonstrates that the RAZR system can be specifically triggered by two disparate phage proteins that share the common property of assembling into large, approximately 170 Å rings. Such rings are not formed by host-encoded proteins, thereby helping to ensure that RAZR is only activated during phage infection. The polymerization of RAZR circumferentially around rings formed by phage-encoded proteins leads to a rapid activation of RNase activity. The assembly of these large oligomers of RAZR and its triggering proteins is reminiscent of the oligomerization that drives activation of mammalian immunity proteins. For instance, during viral infection in mammalian cells, RNase L dimerizes, leading to activation of its HEPN domains and consequent cleavage of viral and endogenous RNAs[19,37]. The restriction factor TRIM5α can form higher-order oligomers, templated by the geometry of retroviral cores, to block viral replication[38]. The use of oligomerization to drive viral restriction in both bacteria and eukaryotes underscores the common principles of immunity shared throughout biology.

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

## Methods

### Strains and growth conditions

All bacterial and phage strains used in this study are listed in Supplementary Table 2. *E. coli* strains were routinely grown at 37 °C in Luria–Bertani (LB) medium for cloning and maintenance. Phages were propagated by infecting a culture of *E. coli* MG1655 at an optical density at 600 nm ($OD_{600}$) of approximately 0.1–0.2 with a multiplicity of infection of 0.1. Cleared cultures were pelleted by centrifugation to remove residual bacteria and filtered through a 0.2-µm filter. Chloroform was then added to phage lysates to prevent bacterial growth. Antibiotics were used at the following concentrations (liquid; plates): carbenicillin (50 µg ml$^{-1}$; 100 µg ml$^{-1}$) and chloramphenicol (20 µg ml$^{-1}$; 30 µg ml$^{-1}$).

### Plasmid construction

All plasmids are listed in Supplementary Table 3. All primers and synthesized gene sequences are listed in Supplementary Table 4.

For the pEssential-*gp77* construct, the coding sequence of gene *77* from phage SECΦ27 was codon optimized for expression in *E. coli*, and commercially synthesized by Integrated DNA Technology as a gBlock (TZ-1) and assembled into pEssential vector amplified with TZ-2 and TZ-3 by Gibson assembly.

For the pBAD33-*gp77* constructs, *gp77* was PCR amplified from phage SECΦ27 using primers TZ-4 and TZ-5 and inserted into pBAD33 linearized with primers TZ-6 and TZ-7 by Gibson assembly. To add a C-terminal HA tag, primers TZ-10 and TZ-11 were used to PCR amplify pBAD33-*gp77* followed by Gibson assembly.

For the pBAD33-portal protein constructs, *gp8$^{T7}$*, *gp31$^{T3}$*, *gp21$^{SECΦ18}$* or *gp52$^{SECΦ27}$* was PCR amplified from the corresponding T7, T3, SECΦ18 or SECΦ27 phage using primers TZ-12 to TZ-17, or TZ-23 and TZ-24, respectively. Amplified fragments were inserted into pBAD33 linearized with TZ-6 and TZ-7 using Gibson assembly. To add a C-terminal HA tag, primers TZ-20 to TZ-22 were each used in combination with TZ-10 to PCR amplify the corresponding pBAD33-portal protein constructs followed by Gibson assembly.

For the pBAD33-*razr* construct, *razr* was PCR amplified from pLAND-*razr* using primers TZ-18 and TZ-19 and inserted into pBAD33 linearized with TZ-6 and TZ-7 by Gibson assembly.

For the pET-*razr*–His$_6$ and pET-*gp77*–His$_6$ constructs: *razr*–His$_6$ or *gp77*–His$_6$ was PCR amplified from pLAND-*razr* or pBAD33-*gp77* using primers TZ-27 to TZ-30, respectively. Amplified fragments were inserted into pET vector linearized with TZ-25 and TZ-26 by Gibson assembly.

For pACYC constructs, to construct pACYC-*gp77*–HA, *gp77*–HA was PCR amplified from pBAD33-*gp77*–HA using primers TZ-33 and TZ-34 and inserted into pACYC linearized with primers TZ-31 and TZ-32. To construct pACYC-*gp8$^{T7}$*–Flag, first *gp8$^{T7}$*–HA was PCR amplified from pBAD33-*gp8$^{T7}$*–HA using primers TZ-34 and TZ-35 and inserted into pACYC linearized with primers TZ-31 and TZ-32. To replace the C-terminal HA tag with a Flag tag, primers TZ-36 and TZ-37 were used to amplify pACYC-*gp8$^{T7}$*–HA followed by Gibson assembly.

For the pLAND-*razr* constructs, pLAND-*razr*–Flag or pLAND-*razr(ΔAGN)* was constructed by PCR amplifying pLAND-*razr* with primers TZ-8 and TZ-9, or TZ-38 and TZ39, followed by Gibson assembly. To construct pLAND-*razr$^{Kv}$*, DNA encoding the *razr$^{Kv}$* open reading frame was codon optimized for expression in *E. coli* and 200 bp of the upstream region from the source organism was added for native expression. DNA was commercially synthesized by Integrated DNA Technology as a gBlock (TZ-40) and assembled into a promoter-less backbone of pLAND amplified with TZ-41 and TZ-42 by Gibson assembly. To construct pLAND-*razr$^{Ec+ZFD(Kv)}$*, sequences for ZFD from RAZR$^{Kv}$ were amplified with primers TZ-43 and TZ-44 and inserted into pLAND-*razr$^{Ec}$* amplified with TZ-45 and TZ-46 by Gibson assembly. To construct pLAND-*razr$^{Kv+ZFD(Ec)}$*, sequences for ZFD from RAZR$^{Ec}$ were amplified with primers TZ-47 and TZ-48 and inserted into pLAND-*razr$^{Kv}$* amplified with TZ-49 and TZ-50 by Gibson assembly.

All mutants were constructed by site-directed mutagenesis using primers designed by Takara Bio In-Fusion design tool.

### Strain construction

Plasmids described above were introduced into *E. coli* MG1655 by Transformation and Storage Solution (TSS) transformation or electroporation.

SECΦ27-mutant phages with gene *77* deleted were generated using a CRISPR–Cas system for targeted mutagenesis as previously described[39]. In brief, sequences for RNA guides to target Cas9-mediated cleavage were designed using the toolbox in Geneious Prime (v2022.0.2) and selected for targeting of gene *77* but nowhere else in the SECΦ27 genome. The guides were inserted into the pCas9 plasmid and tested for their ability to restrict SECΦ27. An efficient guide was selected, and the pCas9-guide plasmid was co-transformed into *E. coli* MG1655 with a high copy-number repair plasmid with sequences flanking gene *77* for homologous recombination, and a pEssential plasmid encoding a recoded version of Gp77 under an arabinose-inducible control. Single colonies of *E. coli* MG1655 containing all three plasmids were grown overnight in LB medium. Overnight cultures were back-diluted 1:10 in 3 ml of LB + 0.2% arabinose and grown at 37 °C for 2 h to induce Gp77 expression. The WT SECΦ27 phage was mixed with 150 µl of induced culture and 4 ml LB + 0.5% agar + 0.2% arabinose and spread on an LB + 1.2% agar + antibiotic + 0.2% arabinose plate. Plates were incubated at 25 °C overnight. Single plaques were screened by Sanger sequencing, and two clones with gene *77* deleted were propagated on strains containing pCas9-guide and pEssential-*gp77*(recoded) with arabinose induction.

### Phage spotting assays and EOP measurements

Phage spotting assays were conducted similarly to that previously described[40]. For phage spotting assays, 80 µl of a bacterial strain of interest was mixed with 4 ml LB + 0.5% agar and spread on an LB + 1.2% agar + antibiotic plate. Phage stocks were then serially diluted in 1× FM buffer (20 mM Tris-HCl pH 7.4, 100 mM NaCl and 10 mM MgSO$_4$), and 2 µl of each dilution was spotted on the bacterial lawn. Plates were then incubated at 25 °C overnight before imaging. EOP was calculated by comparing the ability of the phage to form plaques on an experimental strain relative to the control strain. Experiments were replicated three times independently and representative images are shown.

For spotting phage SECΦ27 lacking gene *77*, *E. coli* MG1655 containing pEssential-*gp77*(recoded) or an empty vector were grown overnight in LB medium. Overnight cultures were back diluted 1:10 in 3 ml of LB + 0.2% arabinose and grown at 37 °C for 2 h to induce Gp77 expression. Of each induced culture, 300 µl was mixed with 4 ml LB + 0.5% agar + 0.2% arabinose and spread on an LB + 1.2% agar + antibiotic + 0.2% arabinose plate. Phages were serially diluted as above and spotted on the bacterial lawn. Plates were incubated at 37 °C overnight. For spotting SECΦ27 lacking gene *77* onto strains producing Gp77 variants (L20R or V22R), *E. coli* MG1655 containing pBAD33-EV or pBAD33-*gp77* (WT or each variant) were grown and processed as described above.

### Isolation of phage escape mutants to infect RAZR

SECΦ27, T3, T7 or SECΦ18 escape mutants of RAZR were isolated by plating a population of phage onto RAZR-containing cells. 20 µl of $10^9$–$10^{10}$ plaque-forming units (PFU) per millilitre phage was mixed with 80 µl overnight culture of *E. coli* MG1655 pLAND-*razr* and the mixture was added to 4 ml of LB + 0.5% agar and spread onto LB + 1.2% agar. Plates were incubated at 25 °C overnight. Single plaques were isolated and propagated using the same strain in LB at 25 °C. Amplified phage lysates were pelleted to remove bacteria, and then plated to single plaques and propagated similarly for a second round of isolation to improve purity. Escape phages were then sequenced by Illumina sequencing as described below to identify mutations.

## Phage DNA extraction and Illumina sequencing

Phage DNA extraction and sequencing were conducted as previously described[40]. To extract phage DNA, high titre phage lysates (more than $10^6$ PFU $\mu l^{-1}$) were treated with DNase I (0.001 U $\mu l^{-1}$) and RNase A (0.05 mg $ml^{-1}$) at 37 °C for 30 min. 10 mM EDTA was used to inactivate the nucleases. Lysates were then incubated with proteinase K at 50 °C for 30 min to disrupt capsids and release phage DNA. Phage DNA was isolated by ethanol precipitation. In brief, NaOAc pH 5.2 was added to 300 mM followed by 100% ethanol to a final volume fraction of 70%. Samples were incubated at −80 °C overnight, pelleted at 21,000$g$ for 20 min and supernatant removed. Pellets were washed with 100 µl isopropanol and 200 µl 70% (v/v) ethanol, and then air dried at room temperature and resuspended in 25 µl 1× TE buffer (10 mM Tris-HCl and 0.1 mM EDTA pH 8).

To prepare Illumina sequencing libraries, 100–200 ng of genomic DNA was sheared in a Diagenode Bioruptor 300 sonicator water bath for 20 × 30 s cycles at maximum intensity. Sheared genomic DNA was purified using AmpureXP beads, followed by end repair, 3′ adenylation and adaptor ligation. Barcodes were added to both 5′ and 3′ ends by PCR with primers that anneal to the Illumina adaptors. The libraries were cleaned by Ampure XP beads using a double cut to elute fragment sizes matching the read lengths of the sequencing run. Libraries were sequenced on an Illumina MiSeq at the MIT BioMicro Center. Illumina reads were assembled to the corresponding reference genomes using Geneious Prime (v2022.0.2).

## Toxicity assays on solid media

Bacterial toxicity assays were conducted similarly to that previously described[40]. For co-producing RAZR with Gp77 or portal proteins (Gp8[T7], Gp31[T3] and Gp21[SECΦ18]), single colonies of *E. coli* MG1655 harbouring pLAND-*razr* and pBAD33-*gp77* or pBAD33-portal protein (WT or the corresponding variants) were grown for 6 h at 37 °C in LB–glucose to saturation. Of each saturated culture, 200 µl was then pelleted by centrifugation at 4,000$g$ for 10 min, washed once in 1× PBS, and resuspended in 400 µl 1× PBS. Cultures were then serially diluted tenfold in 1× PBS and spotted on M9L plates supplemented with 0.4% glucose or 0.2% arabinose. M9L plates contain M9 medium (6.4 g $l^{-1}$ $Na_2HPO_4$–$7H_2O$, 1.5 g $l^{-1}$ $KH_2PO_4$, 0.25 g $l^{-1}$ NaCl and 0.5 g $l^{-1}$ $NH_4Cl$ medium supplemented with 0.1% casamino acids, 0.4% glycerol, 2 mM $MgSO_4$ and 0.1 mM $CaCl_2$) supplemented with 5% LB (v/v). Plates were then incubated at 37 °C overnight before imaging.

To quantify colony sizes, cultures were processed as described above and plated on M9L plates supplemented with 0.4% glucose or 0.2% arabinose to single colonies. Colony sizes of 30–100 colonies on each plate were quantified by Fiji, and the ratio of colony sizes on arabinose and glucose plates were calculated. Data reported are three independent biological replicates.

For producing full-length RAZR, *E. coli* MG1655 containing pBAD33-*razr* were grown to saturation and processed as above. Cultures were plated onto 0.4% glucose and 0.2% arabinose and incubated at 37 °C overnight.

## Co-immunoprecipitation analysis

Co-immunoprecipitation experiments were conducted similar to those previously described[40]. For co-producing RAZR with Gp77 or portal proteins (Gp8[T7], Gp31[T3] and Gp21[SECΦ18]), *E. coli* MG1655 containing pLAND-*razr*–*Flag* (WT or mutant variants) and pBAD33-*gp77*–*HA* (WT or mutant variants) or pBAD33-portal protein–*HA* were grown overnight in M9–glucose. Overnight cultures were back diluted to $OD_{600}$ = 0.05 in 50 ml of M9 (no glucose) and grown to $OD_{600}$ of approximately 0.3 at 37 °C. Cells were induced with 0.2% arabinose for 30 min at 37 °C, then $OD_{600}$ was measured and cells were pelleted at 4,000$g$ for 10 min at 4 °C. Supernatant was removed and cells were resuspended in 800 µl lysis buffer (25 mM Tris-HCl pH 8.0, 150 mM NaCl, 1 mM EDTA,

1% Triton X-100 and 5% glycerol) supplemented with protease inhibitor (Roche), 1 µl $ml^{-1}$ Ready-Lyse Lysozyme Solution (Lucigen) and 1 µl $ml^{-1}$ benzonase nuclease (Sigma). Samples were lysed by two freeze–thaw cycles, and lysates were normalized by $OD_{600}$. Lysates were pelleted at 21,000$g$ for 10 min at 4 °C, and 750 µl of supernatant were incubated with pre-washed anti-Flag magnetic agarose beads (Pierce) for 1 h at 4 °C with end-over-end rotation. Beads were then washed three times with 500 µl lysis buffer. Laemmli sample buffer (1×; Bio-Rad) supplemented with 2-mercaptoethanol was added to beads directly to elute proteins. Samples were boiled at 95 °C and analysed by 4–20% SDS–PAGE and transferred to a 0.2-µm PVDF membrane. Anti-Flag and anti-HA antibodies (#14793 and #3724, Cell Signaling Technology) were used at a final concentration of 1:1,000, and SuperSignal West Femto Maximum Sensitivity Substrate (Thermo Fisher) was used to develop the blots. Blots were imaged by a ChemiDoc Imaging system (Bio-Rad). Images shown are representatives of two or three independent biological replicates.

## Protein expression and purifications

To produce $His_6$-tagged RAZR (WT or mutant variants) and $His_6$-tagged Gp77 (WT or mutant variants), *E. coli* BL21(DE3) cells were transformed with pET-*razr*–$His_6$ or pET-*gp77*–$His_6$ and grown in LB medium to $OD_{600}$ of 0.5. Protein expression was induced by addition of 0.2 mM IPTG, and cells were grown overnight at 16 °C. The culture was centrifuged at 4,000$g$ for 10 min at 4 °C, and cell pellet was resuspended in lysis buffer (50 mM Tris-HCl pH 8.0, 150 mM NaCl, 2 mM $MgCl_2$, 10 µM $ZnCl_2$ and 1 mM dithiothreitol) supplemented with 0.4 mM PMSF and 10 µg $ml^{-1}$ lysozyme. Cells were disrupted using sonication with amplitude 50 and 3 min total process time (10 s on and 20 s off; Qsonica) and glycerol was added to the lysate at final 10% concentration after sonication. The supernatant was separated from the pellet by centrifugation (15,000 rpm for 30 min, JA-25.50 rotor (Beckman Coulter)). The clarified supernatant was loaded onto a gravity-flow chromatography column (Bio-Rad) packed with 2 ml Ni-NTA agarose resin (Qiagen) pre-equilibrated with 15 ml lysis buffer. The resin was washed with 10 column volumes of wash buffer 1 (50 mM Tris-HCl pH 8.0, 500 mM NaCl, 2 mM $MgCl_2$, 10 µM $ZnCl_2$, 10 mM imidazole, 10% glycerol and 1 mM dithiothreitol), and then with 10 column volumes of wash buffer 2 (50 mM Tris-HCl pH 8.0, 150 mM NaCl, 2 mM $MgCl_2$, 10 µM $ZnCl_2$, 50 mM imidazole, 10% glycerol and 1 mM dithiothreitol). The proteins were eluted in 4 ml elution buffer (50 mM Tris-HCl pH 8.0, 150 mM NaCl, 10 µM $ZnCl_2$, 250 mM imidazole, 10% glycerol and 1 mM dithiothreitol). To remove remaining contaminants, the eluted protein sample was loaded onto a size-exclusion chromatography Superose 6 Increase 10/300 GL column (Cytiva) pre-equilibrated in the size-exclusion chromatography buffer (50 mM Tris-HCl pH 8.0, 150 mM NaCl, 10 µM $ZnCl_2$, 5% glycerol and 1 mM dithiothreitol). The purity of the protein samples were assessed spectrophotometrically and by SDS–PAGE. To reconstitute the WT complex in vitro, purified RAZR–$His_6$ and Gp77–$His_6$ were mixed at a 1:1 molar ratio using concentrations calculated by Bradford assay and loaded onto a Superose 6 Increase 10/300 GL column (Cytiva). The fractions containing the complex of interest were pooled, concentrated and used for structural determination.

For the RAZR(H154A)–Gp77 complex, *E. coli* BL21(DE3) cells were transformed with pET-*razr(H154A)*–$His_6$ and pACYC-*gp77*–*HA*. Protein expression and purification were conducted as described above.

For the RAZR(H154A)–Gp8[T7] complex, *E. coli* BL21(DE3) cells were transformed with pET-*razr(H154A)*–$His_6$ and pACYC-*gp8[T7]*–*Flag*. Protein expression and purification with Ni-NTA agarose resin were conducted as described above. Eluted protein sample was mixed with 2 ml anti-Flag M2 affinity gel (Millipore Sigma) pre-equilibrated with the size-exclusion chromatography buffer, and incubated overnight at 4 °C. The mixture was loaded onto a gravity-flow chromatography column (Bio-Rad) and washed three times with 3 ml size-exclusion chromatography buffer. One bed volume (2 ml) of 1 mg $ml^{-1}$ Flag peptide (APExBIO) was added to the resin, incubated for 30 min at 4 °C,

and proteins were eluted by gravity. Elution was repeated three times, pooled and concentrated, and then loaded onto a Superose 6 Increase 10/300 GL column (Cytiva) for further purification.

For analytical size-exclusion chromatography, purified RAZR–His$_6$, Gp77–His$_6$ or the co-purified RAZR(H154A)–His$_6$–Gp77–HA complex was loaded onto a Superose 6 Increase 3.2/300 analytical size-exclusion chromatography column (Cytiva) pre-equilibrated in the size-exclusion chromatography buffer. Each fraction corresponding to the peak of RAZR(H154A)–His$_6$–Gp77–HA was analysed by SDS–PAGE, transferred to a PVDF membrane and blotted with anti-HA (Cell Signaling Technology) and anti-His (Invitrogen) antibody as described below.

## Cryo-EM sample preparation

For the co-purified RAZR(H154A)–His$_6$–Gp77–HA complex, before vitrification, 2.5 µl of the complex (2.6 mg ml$^{-1}$) was placed on 200-mesh Quantifoil 2/1 copper grids (0.5-s incubation time), which had been glow-discharged for 60 s in an easiGlow glow discharger (Pelco) at 25 mA and were blotted using a FEI Vitrobot Mark IV instrument for 4 s with a blot force of +4 (6 °C; 95% relative humidity).

For the in vitro-reconstituted complex of RAZR–His$_6$ and Gp77–His$_6$, 2.5 µl of the sample (0.5 mg ml$^{-1}$) was placed on 200-mesh Quantifoil 2/1 copper grids using the same parameters as mentioned above.

For the RAZR–His$_6$, 2.5 µl of the sample (0.75 mg ml$^{-1}$) was placed on 200-mesh Quantifoil 2/1 copper grids using the same parameters as mentioned above.

For the Gp77–His$_6$, 2.5 µl of the sample (1 mg ml$^{-1}$) was placed on 200-mesh Quantifoil 2/1 copper grids using the same parameters as mentioned above, except with a 5-s incubation time of the sample on the grid.

For the co-purified complex of Gp8$^{T7}$–Flag and RAZR(H154A)–His$_6$, 2.5 µl of the complex (0.6 mg ml$^{-1}$) was applied to 200-mesh Quantifoil 2/1 copper grids with a 2-nm carbon support. The grids were glow discharged for 20 s in an easiGlow glow discharger (Pelco) at 25 mA. The sample was incubated on the grid for 10 s, then blotted using a FEI Vitrobot Mark IV instrument for 3.5 s with a blot force of +4 (6 °C; 95% relative humidity).

The protein–antibody complex was formed by incubating the co-purified RAZR(H154A)–His$_6$–Gp77–HA complex (2.6 mg ml$^{-1}$) with either 0.5 mg ml$^{-1}$ His-tag antibody (MA1-2135, Invitrogen) or 0.25 mg ml$^{-1}$ Strep-tag antibody (MA5-37747, Invitrogen) for 1 h at 4 °C before grid freezing, or with buffer only as a control. All concentrations are reported as final. Samples were prepared by applying 2.5 µl of the mixture onto 200-mesh Quantifoil 2/1 copper grids. The grids were glow-discharged using a GloQube Plus (MiTeGen) at 25 mA for 60 s. Sample-loaded grids were then blotted for 4 s with a blot force of +4 at 6 °C and 100% relative humidity using a FEI Vitrobot Mark IV instrument (Thermo Scientific).

## Cryo-EM data collection

For the co-purified complex of Gp77 and RAZR(H154A), a total of 12,965 movies without stage tilt (0° tilt) and 8,911 movies with a 30° stage tilt were collected using EPU (v2.12.1; Thermo Fisher Scientific) on a Titan Krios G3i microscope (Thermo Fisher Scientific). The microscope was operated at an acceleration voltage of 300 kV with a magnification of ×130,000, and data were recorded in super-resolution mode on a K3 detector (Gatan) at a pixel size of 0.65 Å (pre-binned by 2). Each movie consisted of 40 frames and was collected within a defocus range of −0.25 to −1.75 µm for the 0° tilt data and −0.75 to −2.5 µm for the 30° tilt data. The total electron dose per specimen was 47.62 e$^-$ Å$^{-2}$ for the 0° tilt data and 47.96 e$^-$ Å$^{-2}$ for the 30° tilt data.

For in vitro-reconstituted complex of RAZR–His$_6$ and Gp77–His$_6$, a total of 11,152 movies were collected using EPU (v2.12.1; Thermo Fisher Scientific) on a Titan Krios G3i microscope (Thermo Fisher Scientific). The microscope was operated at an acceleration voltage of 300 kV,

with a magnification of ×130,000. Data were recorded in super-resolution mode on a K3 detector (Gatan) at a pixel size of 0.65 Å (pre-binned by 2). Each movie consisted of 40 frames and was collected within a defocus range of −0.5 to −2.0 µm. The total electron dose per specimen was 45.7 e$^-$ Å$^{-2}$.

For Gp77–His$_6$, 16,605 movies were collected using EPU (v2.12.1; Thermo Fisher Scientific) on a Titan Krios G3i microscope (Thermo Fisher Scientific). The microscope was operated at an acceleration voltage of 300 kV, with a magnification of ×130,000. Data were recorded in super-resolution mode on a K3 detector (Gatan) at a pixel size of 0.65 Å (pre-binned by 2). Each movie consisted of 40 frames and was collected within a defocus range of −0.75 to −2.25 µm. The total electron dose per specimen was 46.2 e$^-$ Å$^{-2}$.

For RAZR–His$_6$, 19 micrographs were collected using EPU (v2.12.1; Thermo Fisher Scientific) on a Talos Arctica G2 microscope (Thermo Fisher Scientific). The microscope was operated at an acceleration voltage of 200 kV, with a magnification of ×73,000. Data were recorded on a Falcon 3EC detector at a pixel size of 1.95 Å and a defocus of −5 µm during sample screening.

For the co-purified complex of Gp8$^{T7}$–Flag and RAZR(H154A)–His$_6$, a total of 4,139 movies were collected using EPU (v2.12.1; Thermo Fisher Scientific) on a Titan Krios G3i microscope (Thermo Fisher Scientific). The microscope was operated at an acceleration voltage of 300 kV, with a magnification of ×130,000. Data were recorded in super-resolution mode on a K3 detector (Gatan) at a pixel size of 0.65 Å (pre-binned by 2). Each movie consisted of 40 frames and was collected within a defocus range of −0.75 to −2.25 µm. The total electron dose per specimen was 48.53 e$^-$ Å$^{-2}$.

For RAZR(H154A)–His$_6$–Gp77 samples incubated with either anti-His antibody or buffer-only control, 2,000 movies were collected for each condition using EPU (v.3.11.0) on a Titan Krios G3 operated at 300 kV. For the RAZR(H154A)–His$_6$–Gp77 samples incubated with anti-Strep tag II antibody, 1,000 movies were collected under otherwise identical conditions. Data were acquired at a nominal magnification of ×130,000 with multiple images per hole and recorded on a Falcon 4 detector, with an effective pixel size of 0.776 Å. Each movie consisted of 50 frames, with a defocus range of −0.4 to −2 µm and a total exposure of 52.48 e$^-$ Å$^{-2}$.

## Cryo-EM pre-processing and particle picking

Data processing was performed in cryoSPARC[41] (v4.5.3 and v4.7.0) using default parameters unless otherwise noted. For data #1 (0° tilt data) of the co-purified complex of Gp77–RAZR(H154A) (Supplementary Fig. 2), 12,965 raw movies were pre-processed using 'Patch Motion Correction' and 'Patch CTF Estimation'. Visual inspection revealed that the particles predominantly adopted 'top' views. In the cryoSPARC live session, 14 2D classes were selected and used for 'Template Picker' (particle diameter of 170 Å). Particles were extracted (box size of 520 × 520 pixels, Fourier cropped to 256). After two rounds of 2D classification, 1,015,530 particles were selected and extracted (box size of 720 × 720 pixels, Fourier cropped to 360), followed by three additional rounds of 2D classification. A final stack of 793,658 particles was selected.

For data #2 (30° tilt data) of the co-purified complex of Gp77 and RAZR(H154A) (Supplementary Fig. 2), 8,911 raw movies were pre-processed using Patch Motion Correction and Patch CTF Estimation. Two 2D classes from the 0° tilt data (representing particles with one and three rings) were selected for use in the 'Template Picker' (particle diameter of 250 Å). Particles were extracted (box size of 800 × 800 pixels, Fourier cropped to 360). After two rounds of 2D classification, 33,400 particles were selected as a preliminary stack.

For the in vitro-reconstituted Gp77–RAZR complex (Supplementary Fig. 3), 11,152 raw movies were pre-processed with Patch Motion Correction and Patch CTF Estimation. A total of 1,796 particles were manually picked using the Manual Picker utility. After 2D classification, two classes corresponding to top and side views (1,470 particles) were

selected and used for training with the 'Topaz Train'[42] in cryoSPARC, followed by 'Topaz Extract'. Particles ($n$ = 195,741) were extracted (box size of 720 × 720 pixels, Fourier cropped to 360), followed by three rounds of 2D classification. A final stack of 42,191 particles was selected as a preliminary stack.

### Ab initio reconstruction, global refinement and model building

For the co-purified complex of Gp77 and RAZR(H154A), ab initio reconstruction was performed using two classes and $C_1$ symmetry. On the basis of 2D class averages, the presence of higher-order symmetry ($C_{12}$ or $C_{24}$) was evident in both the inner and the outer rings. Homogeneous refinement was carried out using $C_{24}$ symmetry, followed by heterogeneous refinement ($C_{24}$) using two classes. Particles ($n$ = 238,489) were selected and further refined using homogeneous refinement ($C_{24}$).

We suspected that there might be a discrepancy in symmetry within the inner and outer ring, therefore homogenous refinement was tested with $C_{12}$ symmetry. Duplicate particles were removed using the 'Remove Duplicates' utility, and the data were subjected to non-uniform refinement ($C_6$ symmetry).

Particles ($n$ = 33,400) from the 30° tilt dataset were used for ab initio reconstruction ($C_1$), followed by homogeneous refinement ($C_1$), and another round of homogeneous refinement using $C_{12}$ symmetry. These particles were then extracted (box size of 720 × 720 pixels, Fourier cropped to 360) and combined with the 0° tilt data for further homogeneous refinement ($C_1$). After heterogeneous refinement ($C_1$), 234,639 particles were selected for non-uniform refinement with $C_{12}$ symmetry.

A mask containing both the inner and the outer rings was generated using ChimeraX[43] (threshold of 0.2, soft pad 15) for local refinement. The particles were then subjected to global and local CTF refinement, followed by another round of local refinement and the application of the 'remove duplicates' utility. Subsequently, another round of local refinement and remove duplicates was performed, followed by both global and local CTF refinement. Finally, the particles underwent a final round of local refinement ($C_{12}$ symmetry) using the previous mask, resulting in a GSFSC (gold-standard Fourier shell correlation) of approximately 3.4 Å map with a sphericity score of 0.79 (out of 1; Extended Data Fig. 4a–c).

For the in vitro-reconstituted Gp77–RAZR complex, ab initio reconstruction was first performed using a single class with $C_{12}$ symmetry, followed by non-uniform refinement ($C_{12}$). Particles were then re-extracted (box size of 1,000 × 1,000 pixels, Fourier cropped to 440) and subjected to an additional round of 2D classification. A second ab initio reconstruction was carried out using two classes with the 'ab initio reconstruction, high symmetry' utility ($C_{12}$ symmetry). Particles corresponding to the volume representing the intact complex were selected for non-uniform refinement ($C_{12}$). After global and local CTF refinement, another round of non-uniform refinement ($C_{12}$) was performed. To further improve the model, symmetry expansion ($C_{12}$) was applied, followed by 3D classification with a mask containing one copy of Gp77 and two copies of RAZR, without pose refinement (10 classes). A class exhibiting well-defined density for the full complex was selected and used as the initial model for a subsequent non-uniform refinement of the original particle stack (before symmetry expansion), following local CTF refinement. This procedure yielded a final map at approximately 3.4 Å resolution (GSFSC) (Extended Data Fig. 4d–f and Supplementary Fig. 3a) with a sphericity score of 0.94 (out of 1). To improve the resolution of Gp77, signal subtraction was performed using two masks: one containing only the inner ring and another containing only the outer rings. To prepare particles for signal subtraction, local refinement was carried out with a mask applied to the outer ring to improve alignment of particles on the outer ring features. Signal subtraction was then performed using the outer ring mask. The resulting signal-subtracted particle images were used for homogeneous reconstruction with a mask applied to the central ring, followed by local refinement with $C_{24}$ symmetry. This yielded a Gp77-only map at a 3.1 Å GSFSC resolution (Extended Data Fig. 5a–c and Supplementary Fig. 3b) with a sphericity score of 0.94 (out of 1).

To further improve the resolution of the RAZR ZFD (central ring), focused refinement was performed on the final particle stack (28,190) using a mask that included only the central ring, yielding a focused-refined map at a GSFSC resolution of 3.5 Å (Extended Data Fig. 5d–f and Supplementary Fig. 3c) with a sphericity score of 0.75 (out of 1). Applying the same procedure to the outermost ring did not result in improved resolution.

For atomic model generation, only the in vitro-reconstituted complex maps were used, as these showed reduced anisotropy compared with the co-purified map. The Gp77-focused map enabled docking of residues 1–125 into the inner ring using 'phenix.local_em_fitting' in ChimeraX, followed by manual adjustments in Coot and subsequent refinement in Phenix. The final modelled residue indicated that the remaining C-terminal domain—predicted to form three α-helices followed by a long unstructured region—is flexibly positioned at the top of the assembly, corresponding to a low-resolution density forming an additional ring-like feature (Extended Data Fig. 6b). The accuracy of the Gp77 model (1–125) was supported by well-resolved side-chain densities for bulky residues (Extended Data Fig. 6a; map–model correlation coefficient (mask) of 0.71; $Q$ score[44] (global/expected) of 0.48/0.56).

The central ring was modelled using an AlphaFold prediction of the RAZR ZFD, fitted with phenix.local_em_fitting[45] and refined through manual rebuilding in Coot and Phenix. A continuous density bridging the ZFD and HEPN domains clearly delineated the interdomain loop, suggesting that the HEPN domain occupies the outermost ring, which was resolved at substantially lower resolution relative to the Gp77 core and the RAZR ZFD domain (Extended Data Fig. 4). Docking of the AlphaFold-predicted HEPN domain into this region enabled modelling of RAZR residues 3–179. The remaining portion of the HEPN domain, which could not be confidently placed, is predicted to adopt a helix–loop–helix motif that may flexibly position between subunits. After placement of the AlphaFold-predicted HEPN domain, the entire model was further refined in Phenix to reduce steric clashes and improve overall geometry (map–model CC (mask) of 0.75; overall quality control score (global/expected) of 0.41/0.48; RAZR(ZFD) $Q$ score of 0.42; and RAZR(HEPN) $Q$ score of 0.13).

The 270 Å diameter of the ring reported for the in vitro-reconstituted RAZR–Gp77 complex was a distance measured between the Cα atoms of Q140 in chains OA and OM, and the diameter of approximately 170 Å in Gp8[17] was measured between Cα Q201 of chains A and G (PDB ID: 7EY8) that is almost identical to that of the Gp77 ring in our cryo-EM structure (measured between Cα S31 of chains IA and IM).

Model building was performed using ChimeraX[43] (v1.6), Coot[46] (v0.9.4) and Phenix[47] (v1.21.2-5419). Final maps were sharpened using cryoSPARC. Local resolution was estimated using MonoRes[48] within cryoSPARC; angular Fourier shell correlations were calculated using the 3DFSC server; and $Q$ scores were calculated using a ChimeraX $Q$ score plugin[44].

### Mass photometry

Purified RAZR(H154A)–His$_6$–Gp77–HA or Gp77–His$_6$ samples were diluted to 50 nM (as a complex) in a buffer containing 50 mM Tris-HCl pH 8.0, 150 mM NaCl, 10 μM ZnCl$_2$ and 1 mM dithiothreitol. Of each protein, 2 μl was added to 18 μl of buffer in the well, and measured using a Refeyn TwoMP mass photometer (Refeyn) with a data acquisition time of 60 s. Data were acquired by AcquireMP and analysed by DiscoverMP software. The recorded events were fitted to Gaussian distributions, and masses were calculated by applying calibrations performed with BSA (66 kDa) and thyroglobulin (660 kDa) in the same buffer. Each sample was measured independently two times as replicates.

### Western blot of RAZR expression levels

Single colonies of *E. coli* MG1655 pLAND-*razr–Flag* (WT or mutant variants) were grown overnight in LB. Overnight cultures were back

diluted to $OD_{600} = 0.05$ in 5 ml fresh LB and grown to $OD_{600} = 0.3$ at 37 °C. $OD_{600}$ was measured, and 3 ml of cells was pelleted at $6,000g$ for 10 min with $OD_{600}$ normalized. Supernatant was removed and pellets were resuspended in 1× Laemmli sample buffer (Bio-Rad) supplemented with 2-mercaptoethanol. Samples were then boiled at 95 °C for 15 min and analysed by 4–20% SDS–PAGE and transferred to a 0.2-μm PVDF membrane. Anti-Flag antibody (#14793, Cell Signaling Technology) was used at a final concentration of 1:1,000, and SuperSignal West Femto Maximum Sensitivity Substrate (Thermo Fisher) was used to develop the blots. Blots were imaged by a ChemiDoc Imaging system (Bio-Rad). Blots were stained with Coomassie stain and imaged as loading control. The image shown is a representative of two independent biological replicates.

## Incorporation assays

Incorporation assays were performed similarly to those previously described[40]. For co-producing RAZR and Gp77, single colonies of *E. coli* MG1655 containing pLAND-*razr* and pBAD33-*gp77* or corresponding empty vectors were grown overnight in M9–glucose. Overnight cultures were back diluted to $OD_{600} = 0.05$ in 25 ml M9–glucose and grown to $OD_{600}$ of approximately 0.3 at 37 °C. Cells were pelleted at $4,000g$ for 5 min at 4 °C and washed once with M9 (no glucose), and then back diluted to $OD_{600} = 0.1$ in 15 ml M9 (no glucose) and recovered for 45 min at 37 °C. At the beginning of the experiment, cells were induced with 0.2% arabinose. At the indicated time points (0, 10, 20, 30 and 40 min), $OD_{600}$ was measured and an aliquot of 250 μl of cells was transferred to microcentrifuge tube containing [methyl-$^3$H]-thymidine (Revvity; 40 μCi ml$^{-1}$) for replication measurements, [5,6-$^3$H]-uridine (Revvity; 4 μCi ml$^{-1}$) for transcription measurements or EasyTag EXPRESS-$^{35}$S Protein Labeling Mix, [$^{35}$S] (Revvity; 22 μCi ml$^{-1}$) for translation measurements. Tubes were incubated at 37 °C for 2 min, then quenched by addition of non-radioactive thymidine (1.5 mM), uridine (1.5 mM) or cysteine and methionine (15 mM each) and incubated for an additional 2 min. Samples were then added to ice-cold trichloroacetic acid (TCA) (10% w/v) and incubated at least 30 min on ice to allow for precipitation. Resulting samples were vacuum filtered onto a glass microfibre filter (1820-024, Whatman) that had been pre-wetted with 5% w/v TCA. Filters were washed with 35× volume of 5% w/v TCA, then with 5× volume of 100% ethanol. Air-dried filters were placed in tubes with scintillation fluid and measured in a scintillation counter (PerkinElmer). Counts per million was normalized to $OD_{600}$ and percent incorporation at each time point was calculated by normalizing to $T = 0$. Data reported are the individual data points from two (transcription or translation) or four (replication) independent biological replicates.

## Cell-free translation

Experiments with PURExpress in vitro protein synthesis kit (E6800, NEB) were performed as per the manufacturer's instructions. All reactions were supplemented with 0.8 U μl$^{-1}$ RNase Inhibitor Murine (M0314S, NEB). Purified His$_6$-tagged RAZR protein and purified His$_6$-tagged Gp77 protein were added to the 15 μl reaction at a final concentration of 1 μM each (as monomers). A template plasmid encoding the control protein DHFR was used at 5 ng μl$^{-1}$. The reactions were incubated at 37 °C for 2 h, and 2 μl of each reaction was mixed with 10 μl of 1× Laemmli sample buffer (Bio-Rad) supplemented with 2-mercaptoethanol. The mixtures were boiled for 5 min at 95 °C and analysed by 12% SDS–PAGE. The gels were stained with Coomassie stain and imaged by a ChemiDoc Imaging system (Bio-Rad). Images shown are representatives of three independent biological replicates.

## RNA extraction from cells expressing RAZR and Gp77

Single colonies of *E. coli* MG1655 containing pLAND-*razr* and pBAD33-*gp77* or the corresponding empty vector were grown overnight in M9–glucose. Overnight cultures were back diluted to $OD_{600} = 0.05$ in 20 ml M9–glucose and grown to $OD_{600}$ of approximately 0.3 at 37 °C. Cells

were pelleted at $4,000g$ for 5 min at 4 °C and washed once with M9 (no glucose), and then back diluted to $OD_{600} = 0.1$ in 15 ml M9 (no glucose) and recovered for 45 min at 37 °C. At the beginning of the experiment, cells were induced with 0.2% arabinose. At the indicated time points (0, 30 and 60 min), cells were harvested by adding 900 μl of culture to 100 μl of stop solution (95% ethanol and 5% acid-buffered phenol) on ice and spinning at $13,000g$ for 30 s. Supernatants were removed, and pellets were flash frozen in liquid nitrogen. To extract RNAs, 400 μl of TRIzol (Invitrogen) preheated to 65 °C was added to each pellet. Resuspended pellets were incubated at 65 °C for 10 min at 2,000 rpm in a thermomixer, flash frozen in liquid nitrogen for 10 min and thawed to room temperature. Samples were spun at $21,000g$ for 5 min at 4 °C, and supernatants were mixed with 400 μl of ethanol. RNAs were purified using the Direct-zol RNA Miniprep kit (Zymo Research) per the manufacturer's instructions with on-column DNase treatment. RNA yield was measured by a Nanodrop spectrophotometer. Of each purified RNA, 80 ng was analysed by a Novex 15% TBE–urea gel (Invitrogen) in 1× TBE buffer and stained with SYBR Gold (Invitrogen). For visualizing rRNAs, 1 μg of purified RNAs were analysed by 1% agarose gel in 1× TAE buffer supplemented with 1% bleach and ethidium bromide. The gels were imaged by a ChemiDoc Imaging system (Bio-Rad). Images shown are representatives of two independent biological replicates.

## RNA extraction following phage infection

Single colonies of *E. coli* MG1655 containing pLAND-*razr* or an empty vector were grown overnight in LB medium. Overnight cultures were back diluted to $OD_{600} = 0.05$ in 25 ml LB and grown to $OD_{600}$ of approximately 0.2 at 25 °C. At the beginning of the experiment, cells were infected with phage T7 at multiplicity of infection = 10. At the indicated time points (0, 10, 20, 30 and 40 min), 500 μl of cells were mixed with 500 μl of preheated lysis buffer containing 2% SDS and 4 mM ETDA pH 8.0. Samples were incubated at 100 °C for 5 min, flash frozen in liquid nitrogen and thawed to room temperature. Of acid-buffered phenol (pH 4.5, Sigma) preheated to 67 °C, 1 ml was added to each sample and incubated at 67 °C for 2 min. Samples were pelleted at $21,000g$ for 10 min, and the aqueous layer was collected and repeated with another hot phenol extraction. A third round of extraction was done with 1 ml acid-buffered phenol chloroform (pH 4.5, Ambion). RNA was precipitated with 1× volume isopropanol, 0.1× volume 3 M NaOAc pH 5.5 and 0.01× volume GlycoBlue on ice for 1 h. RNAs were pelleted at $21,000g$ for 30 min at 4 °C. Pellets were washed twice with 500 μl of ice-cold 70% ethanol, air dried and resuspended in 90 μl nuclease-free water. To remove DNA, samples were treated with 10 μl of 10× Turbo DNase buffer (Invitrogen) and 2 μl Turbo DNase I (Invitrogen) for 20 min at 37 °C. An additional 2 μl of Turbo DNase I was added, and samples were incubated for another 20 min at 37 °C. Samples were then mixed with 96 μl of water, extracted with 200 μl of acid-buffered phenol chloroform (pH 4.5, Ambion) and precipitated for 1 h at −20 °C with 3× volume ice-cold ethanol, 0.1× volume 3 M NaOAc pH 5.5 and 0.01× volume GlycoBlue. RNAs were pelleted, washed and resuspended as described above. RNAs were analysed by a Novex 15% TBE–urea gel and a 1% agarose gel with 1% bleach as described above.

## In vitro cleavage assays

For tRNA cleavage assays, purified RAZR–His$_6$ and Gp77–His$_6$ were added to a 5 μl reaction at a final concentration of 1.2 μM each (as monomers) and mixed with 180 ng of extracted bulk *E. coli* tRNAs in cleavage buffer (50 mM Tris-HCl pH 8.0, 150 mM NaCl, 30 mM KCl, 7 mM MgCl$_2$, 10 μM ZnCl$_2$ and 1 mM dithiothreitol). The co-purified RAZR(H154A)–His$_6$–Gp77–HA complex or the in vitro-reconstituted complex RAZR–His$_6$–Gp77–His$_6$ were each added at a final concentration of 2.5 μM (as monomers). After incubation at 37 °C for 1 h, 2.5 μl of each reaction was mixed with 2.5 μl of Novex TBE–urea sample buffer (Invitrogen) and analysed by a Novex 15% TBE–urea gel (Invitrogen) in 1× TBE buffer and stained with SYBR Gold stain.

For rRNA cleavage assays, purified RAZR–His$_6$ and Gp77–His$_6$ were added to a 15 µl reaction at a final concentration of 1.2 µM each and mixed with *E. coli* 70S ribosomes (NEB) at a final concentration of 0.44 µM in cleavage buffer. The co-purified RAZR(H154A)–His$_6$–Gp77–HA complex or the in vitro-reconstituted complex RAZR–His$_6$–Gp77–His$_6$ were each added at a final concentration of 2.5 µM (as monomers). After incubation at 37 °C for 1 h, RNAs were extracted by acid-buffered phenol chloroform and precipitated as described above. Of purified RNAs, 1 µg were analysed by 1% agarose gel in 1× TAE buffer supplemented with 1% bleach and ethidium bromide.

For cleavage assays of RNA or DNA oligos (single stranded or double stranded), purified RAZR–His$_6$ and Gp77–His$_6$ were added to a 15 µl reaction at a final concentration of 1.2 µM each (as monomers) and mixed with 500 ng each corresponding oligo in cleavage buffer. After incubation at 37 °C for 1 h, 3 µl of each reaction was analysed by a Novex 15% TBE–urea gel (Invitrogen) as described above. Oligo sequences are listed in Supplementary Table 4 (TZ-51 to TZ-54).

For mRNA cleavage assays, mRNA substrates were in vitro transcribed using MEGAscript T7 transcription kit (Invitrogen) per the manufacturer's instructions. In brief, each coding sequence was PCR amplified from genomic DNA of *E. coli* MG1655 with a T7 promoter introduced directly upstream of it using primers TZ-55 to TZ-70. Amplified DNA was purified with DNA clean & concentrator kit (Zymo Research) and used as template. In vitro transcription reactions were incubated at 37 °C for 4 h, and treated with 1 µl Turbo DNase at 37 °C for 15 min. Transcribed mRNAs were purified by phenol-chloroform extraction and ethanol precipitation as described above. For cleavage assays, purified RAZR–His$_6$ and Gp77–His$_6$ were added to a 15 µl reaction at a final concentration of 1.2 µM each (as monomers) and mixed with 2 µg of each mRNA substrate in cleavage buffer. After incubation at 37 °C for 1 h, 0.75 µl of each reaction was analysed by a Novex 15% TBE–urea gel (Invitrogen) as described above.

### RNA-seq sample preparation and analysis

For co-producing Gp77 and RAZR, RNAs were extracted as described above from *E. coli* MG1655 cells containing pLAND-*razr* and pBAD33-*gp77* or the corresponding empty vector, after inducing with arabinose for 0, 10 and 30 min. RNAs were purified using the Direct-zol RNA Miniprep kit (Zymo Research) per the manufacturer's instructions and eluted in 90 µl nuclease-free water. To remove DNA, samples were treated with 10 µl of 10× Turbo DNase buffer (Invitrogen) and 2 µl Turbo DNase I (Invitrogen) for 20 min at 37 °C. An additional 2 µl of Turbo DNase I was added, and samples were incubated for another 20 min at 37 °C. Samples were then mixed with 96 µl of water, extracted with 200 µl of acid-buffered phenol chloroform (pH 4.5, Ambion), and precipitated for 1 h at −20 °C with 3× volume ice-cold ethanol, 0.1× volume 3 M NaOAc pH 5.5 and 0.01× volume GlycoBlue. RNAs were pelleted, washed and resuspended as described above. RNA quality was assessed by a Novex 6% TBE–urea gel, and yield was measured by a Nanodrop spectrophotometer. For RNA-seq during T7 infection, RNAs were extracted and processed as described above.

To prepare libraries for RNA-seq, rRNA was removed using a developed *E. coli* rRNA depletion kit[49], with 1.7 µg of total RNAs as input. rRNA-depleted samples were further purified using RNA Clean and Concentrator kit (Zymo Research). Libraries were generated using NEBNext Ultra II RNA Library Prep Kit for Illumina (NEB) following the manufacturer's instructions for purified mRNA or rRNA-depleted RNA. Libraries were sequenced on an Illumina NextSeq at the MIT BioMicro Center. Two biological replicates were harvested and sequenced independently.

RNA-seq data were analysed similarly to that previously described[30]. FASTQ files for each sample were trimmed using cutadapt (v1.15)[50] and then mapped to the MG1655 genome (NC_00913.2) and the T7 genome (V01146), or the consensus map of rRNA loci as previously described[30] using bowtie2 (v2.3.4.1)[51] with the following arguments: −D 20, −I 40, −X 300, −R 3, −N 0, −L 20, −i S,1,0.50. Sam files generated from bowtie2 mapping were converted to bam files using samtools (v1.7)[52], and then converted to numpy arrays using the genomearray3 Python library[30]. Gene names and coding regions were extracted from NCBI annotations. For the T7 transcriptome, mRNA positions were extracted based on the T7 promoter and terminator positions from NCBI annotations, similar to that previously described[53]. For analysis of fragment density across the transcriptome, one count was added for all positions between and including the 5' and 3' ends of the reads. To correct for variability in sequencing depth, counts at each position were divided by a sample size factor as previously described for normalization[30]. In brief, counts in each coding region were summed for each sample, and the geometric mean of these sums was calculated to yield a reference sample. The total counts in each coding region were then normalized by the reference sample, and the median of these ratios was taken as the size factor for that sample. The cleavage ratio at each nucleotide position was calculated as the log$_2$ transformed + Gp77:empty vector ratio (for co-producing Gp77 and RAZR) or the log$_2$ transformed + RAZR:empty vector ratio (for phage infection), and the average of two replicates were taken. Any regions in the empty vector sample that had fewer counts than the expression cut-off of 64 counts (for co-producing Gp77 and RAZR) or 32 counts (for phage infection) were discarded, and minimum cleavage ratio was taken for each coding region.

### Homology search and conservation analysis

Homologues of RAZR were identified by ConSurf[54] with default settings to search the UniRef90 database, using AlphaFold-predicted RAZR structure as input. Homologues (*n* = 150) were used to generate the multiple sequence alignment by MAFFT. Conservation scores were calculated using the Bayesian method and default settings.

### Reporting summary

Further information on research design is available in the Nature Portfolio Reporting Summary linked to this article.

### Code availability

This paper does not report any original code.

### Data availability

Cryo-EM maps and associated atomic models are available with the following PDB and Electron Microscopy Data Bank codes: 9Y9C and 9Y6U (PDB); and 72693, 72631, 72692 and 72691 (Electron Microscopy Data Bank). Sequencing data are available in the Sequence Read Archive under BioProject PRJNA1207560. All other data are available in the article or the Supplementary Information. Other previously published structures are available in the PDB. The UniRef90 database is publicly available. Reference bacteria and phage genomes are publicly available: MG1655 (NC_00913.2), SECΦ27 (NC_047938.1), T3 (NC_047864.1), T7 (NC_001604.1) and SECΦ18 (NC_073071.1). Materials including strains and plasmids are available on reasonable request.

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

**Acknowledgements** Samples were prepared and collected at the Washington University in St. Louis Center for Cellular Imaging (WUCCI) and Cryogenic Electron Microscopy Facility at MIT.nano, which was a gift from the Arnold and Mabel Beckman Foundation. The work was supported by startup funds from the Department of Biochemistry and Molecular Biophysics at Washington University School of Medicine (R35GM160328 and R35-GM141517). We thank R. T. Sauer and J. H. Davis for their valuable discussions and comments; the MIT BioMicro Center and its staff for their support in sequencing; the MIT Biophysical Instrumentation Facility and its staff for help with mass photometry experiments; A. Mohamed and the Vos laboratory for help with the analytical size-exclusion chromatography; and all members of the Ghanbarpour and Laub laboratory for discussions. We thank SBGrid for providing access to structural biology software used in this study[55]. M.T.L. is an Investigator of the Howard Hughes Medical Institute.

**Author contributions** Experiments were conceived and designed by T.Z., A.G. and M.T.L. All biochemical and phenotypical experiments were done by T.Z., with help from C.R.B. (RNA-seq during phage infection) and R.B. (phage escaper isolation in Fig. 4e). All protein purifications were done by T.Z., except the antibody interaction studies (Extended Data Fig. 6e–g), which were done by Y.L. Cryo-EM sample preparation and data collection were done by A.G. High-resolution reconstruction and model building were performed by A.G. and Y.L. N.I prepared the cryo-EM grids for Extended Data Fig. 6e–g and collected datasets. Figure design, manuscript writing and editing were done by T.Z., A.G. and M.T.L. Project supervision was provided by A.G. and M.T.L. Funding was provided by A.G. and M.T.L.

**Competing interests** The authors declare no competing interests.

**Additional information**
**Correspondence and requests for materials** should be addressed to Alireza Ghanbarpour or Michael T. Laub.

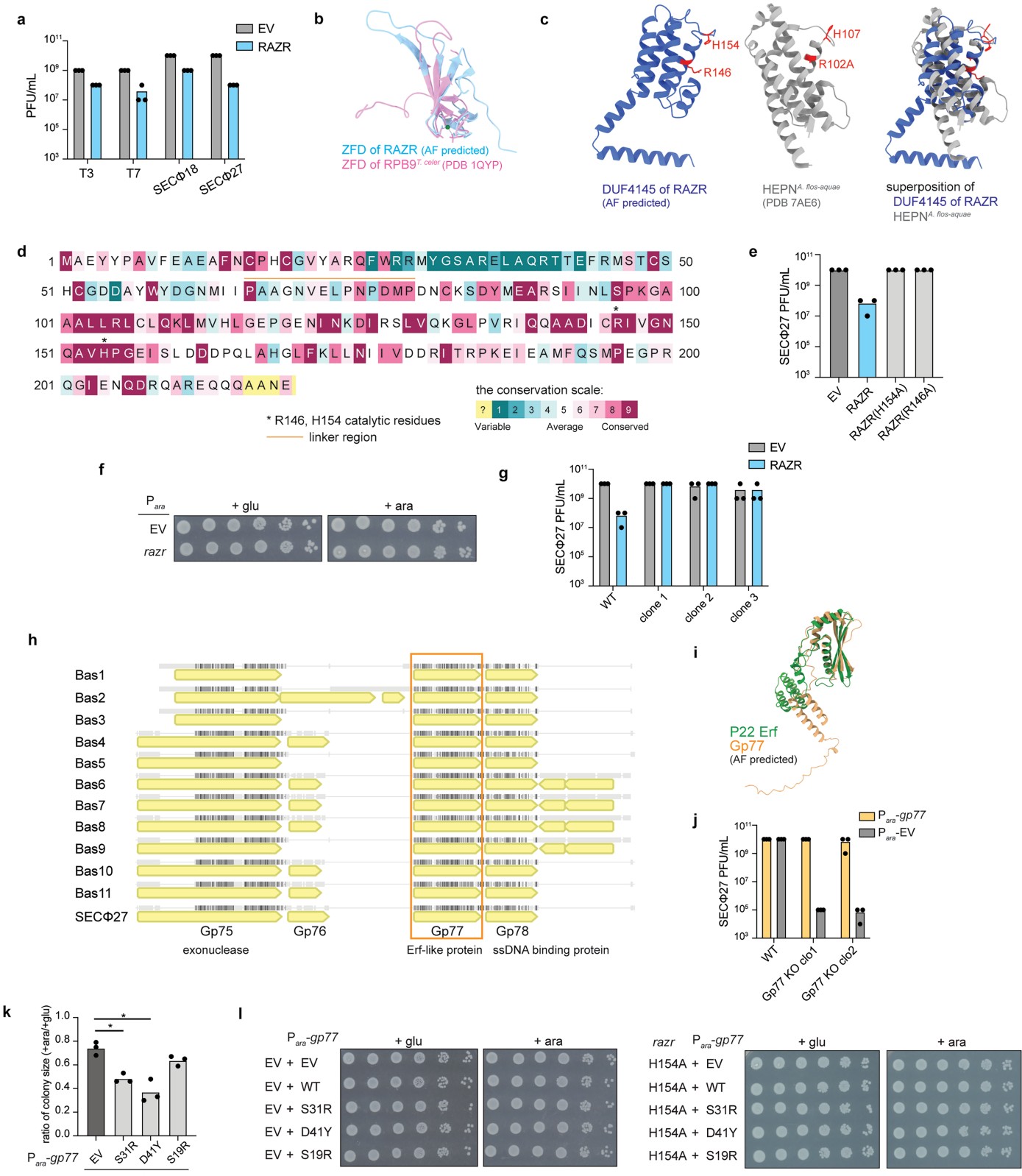

**Extended Data Fig. 1** | See next page for caption.

**Extended Data Fig. 1 | Analysis of RAZR and its trigger in phage SECΦ27, Gp77.** (**a**) Summary of 3 independent replicates of phage spotting assay in Fig. 1a. (**b**) Superposition of the ZFD of RAZR (AlphaFold3-predicted, colored in light blue) onto the ZFD of RPB9 from *Thermococcus celer* (PDB ID: 1QYP, colored in pink). (**c**) Cartoon representation and superposition of the AlphaFold3-predicted DUF4145 domain of RAZR (colored in dark blue) and the structure of HEPN toxin from *Aphanizomenon flos-aquae* (PDB ID: 7AE6, colored in grey). Active site arginines and histidines are colored in red. (**d**) Amino acid sequences of RAZR$^{Ec}$ from *E. coli* UMB1727 strain colored by the conservation score of each amino acid calculated by ConSurf. (**e**) Summary of 3 independent replicates of phage spotting assay in Fig. 1d. (**f**) Serial dilutions of cells producing RAZR from an arabinose-inducible promoter or an empty vector on media containing glucose or arabinose. (**g**) Summary of 3 independent replicates of phage spotting assay in Fig. 1e. (**h**) Genome alignment of phage SECΦ27 and its related phages Bas1-Bas11, with protein annotations in yellow and tracks indicating nucleotide similarities (black means identical). Gp75-Gp78 from SECΦ27 are labeled correspondingly. (**i**) Superposition of the AlphaFold3-predicted Gp77 from phage SECΦ27 (orange) or the Erf protein from phage P22 (green). (**j**) Summary of 3 independent replicates of phage spotting assay in Fig. 1f. (**k**) Ratio of colony size of the indicated strain plated onto arabinose (+ara) or glucose (+glu) plates. Asterisks indicate P = 0.003 (S31R) or P = 0.005 (D41Y) (unpaired two-tailed Student's t-test). (**l**) Serial dilutions of cells producing an empty vector (*left*) or RAZR(H154A) (*right*) from its native promoter and Gp77 (wild-type or the indicated variant) from an arabinose-inducible promoter on media containing glucose or arabinose.

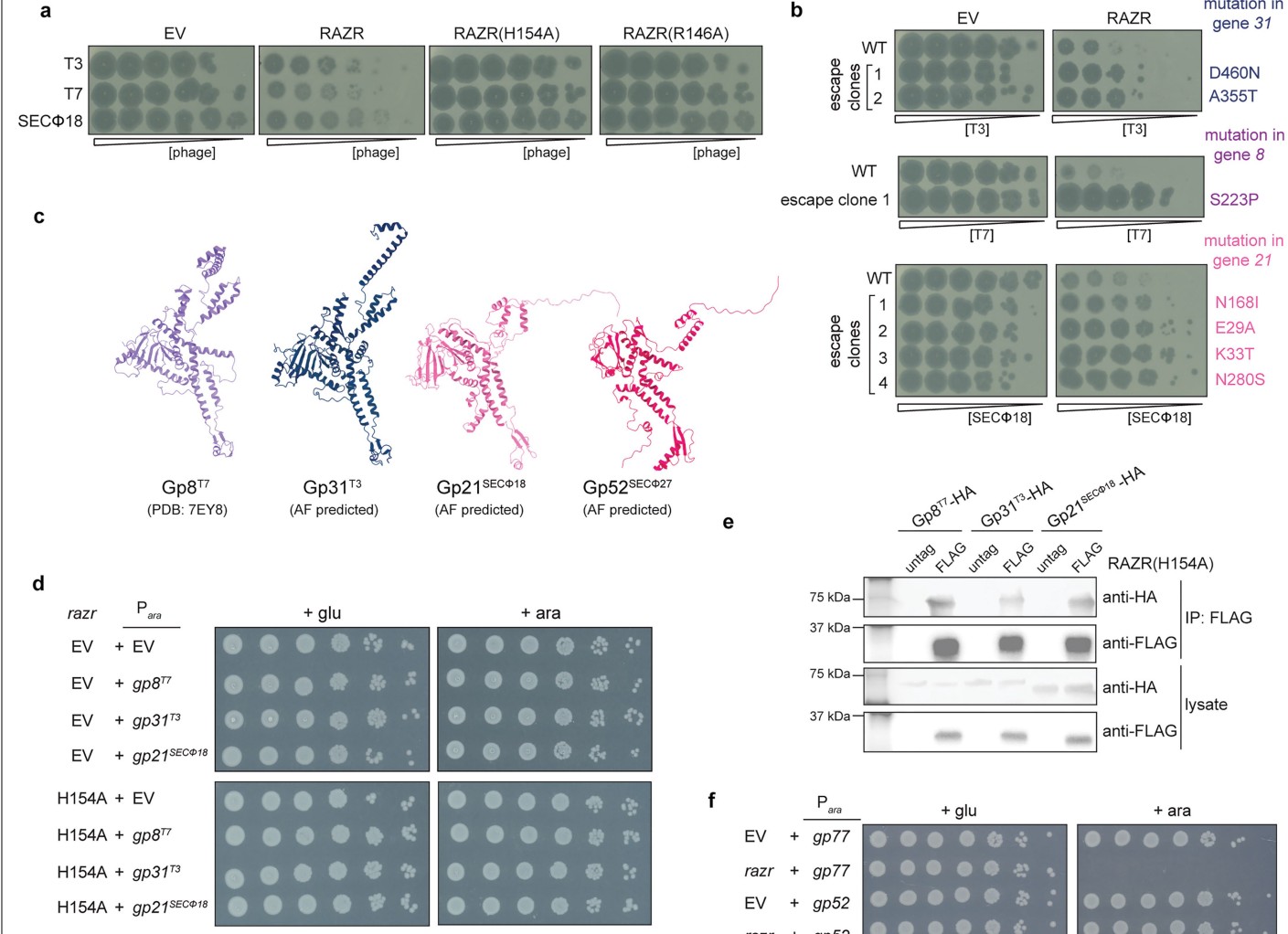

**Extended Data Fig. 2 | RAZR is activated by the portal proteins from T3, T7, and SECΦ18.** (**a**) Serial dilutions of the indicated phage spotted on lawns of cells harboring an empty vector or producing the wild-type or the indicated variant of RAZR. (**b**) Serial dilutions of escape clones of T3, T7, or SECΦ18 and the corresponding wild-type phage spotted on lawns of cells harboring an empty vector or a RAZR expression vector, with the corresponding mutations labeled on the right. (**c**) Structure of Gp8^T7 (PDB ID: 7EY8) and the AlphaFold3-predicted structures of portal proteins Gp31^T3, Gp21^SECΦ18, or Gp52^SECΦ27. (**d**) Serial dilutions of cells producing an empty vector or RAZR(H154A) from its native promoter

and Gp8^T7, Gp31^T3, or Gp21^SECΦ18 from an arabinose-inducible promoter on media containing glucose or arabinose. (**e**) RAZR(H154A)-FLAG was immunoprecipitated from cells producing RAZR(H154A)-FLAG and Gp8^T7-HA, Gp31^T3-HA, or Gp21^SECΦ18-HA and probed for the presence of each protein via the HA tag. (**f**) Serial dilutions of cells producing an empty vector or RAZR from its native promoter and Gp77 (putative recombination protein) or Gp52 (portal protein) of phage SECΦ27 from an arabinose-inducible promoter on media containing glucose or arabinose.

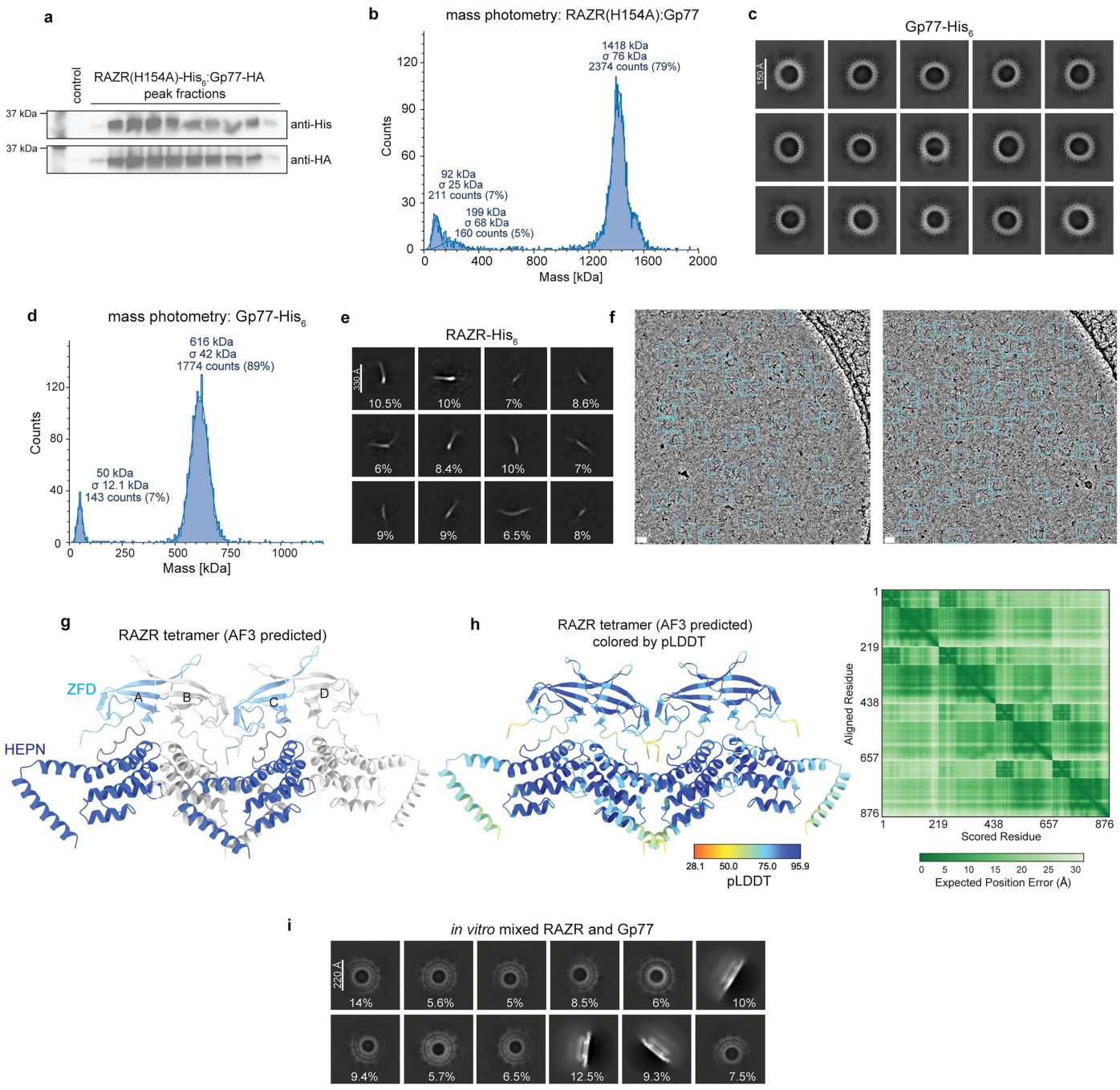

**Extended Data Fig. 3 | Structural analysis of RAZR and Gp77.** (**a**) Immunoblots for RAZR(H154A)-His₆ or Gp77-HA in the analytical size-exclusion chromatography fractions corresponding to the co-purified RAZR(H154A)-His₆:Gp77-HA complex. (**b**) Mass photometry analysis of RAZR(H154A):Gp77 complex. (**c**) Sample 2D class averages of Gp77-His₆ calculated using 410,040 particles. (**d**) Mass photometry analysis of Gp77-His₆ proteins. (**e**) Sample 2D class averages of RAZR-His₆ calculated using 4,306 particles. (**f**) The sample micrographs of RAZR-His₆ alone revealing the heterogeneity in the conformation of RAZR particles (in blue boxes) in the absence of phage protein. (**g**) AlphaFold3-predicted RAZR tetramer colored by chains and by domains. (**h**) *Left*, AlphaFold3-predicted RAZR tetramer colored by pLDDT. *Right*, predicted aligned error (PAE) plot. (**i**) Sample 2D class averages showing the top and side views of the RAZR and Gp77 complex reconstituted in vitro calculated using 75,351 particles.

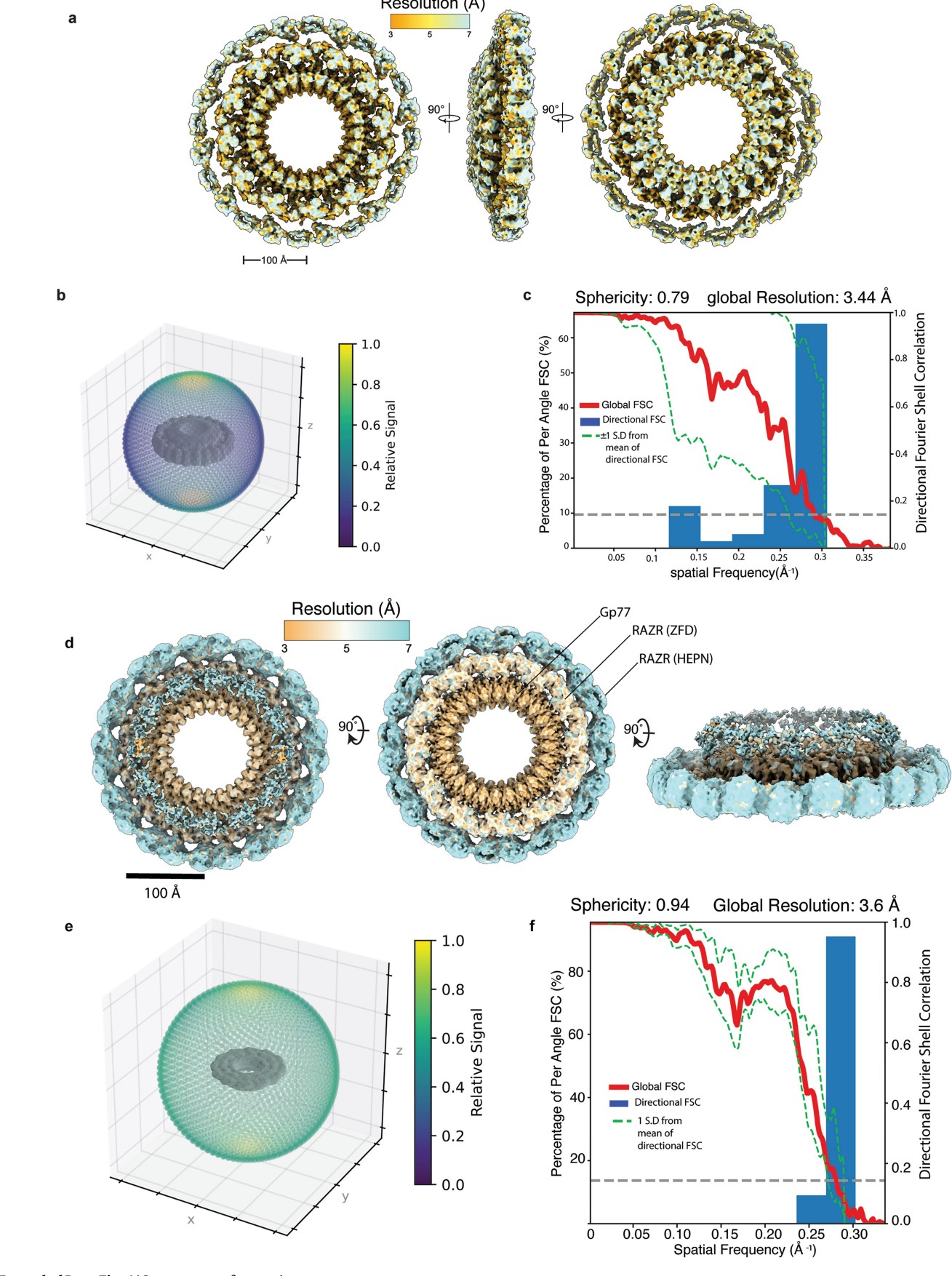

**Extended Data Fig. 4** | See next page for caption.

Extended Data Fig. 4 | Estimates of resolution and angular sampling for the co-purified RAZR(H154A):Gp77 complex and in vitro reconstituted RAZR:Gp77 cryo-EM structures. (**a**) The map of the co-purified RAZR(H154A):Gp77 complex colored by local resolution, as estimated by the cryoSPARC MonoRes implementation. (**b**) Viewing distribution plot of the co-purified complex. (**c**) Directional resolution plot of the co-purified complex. (**d**) The map of the in vitro reconstituted RAZR:Gp77 complex colored by local resolution, as estimated by the cryoSPARC MonoRes implementation. (**e**) Viewing distribution plot of the in vitro reconstituted complex. (**f**) Directional resolution plot of the in vitro reconstituted complex.

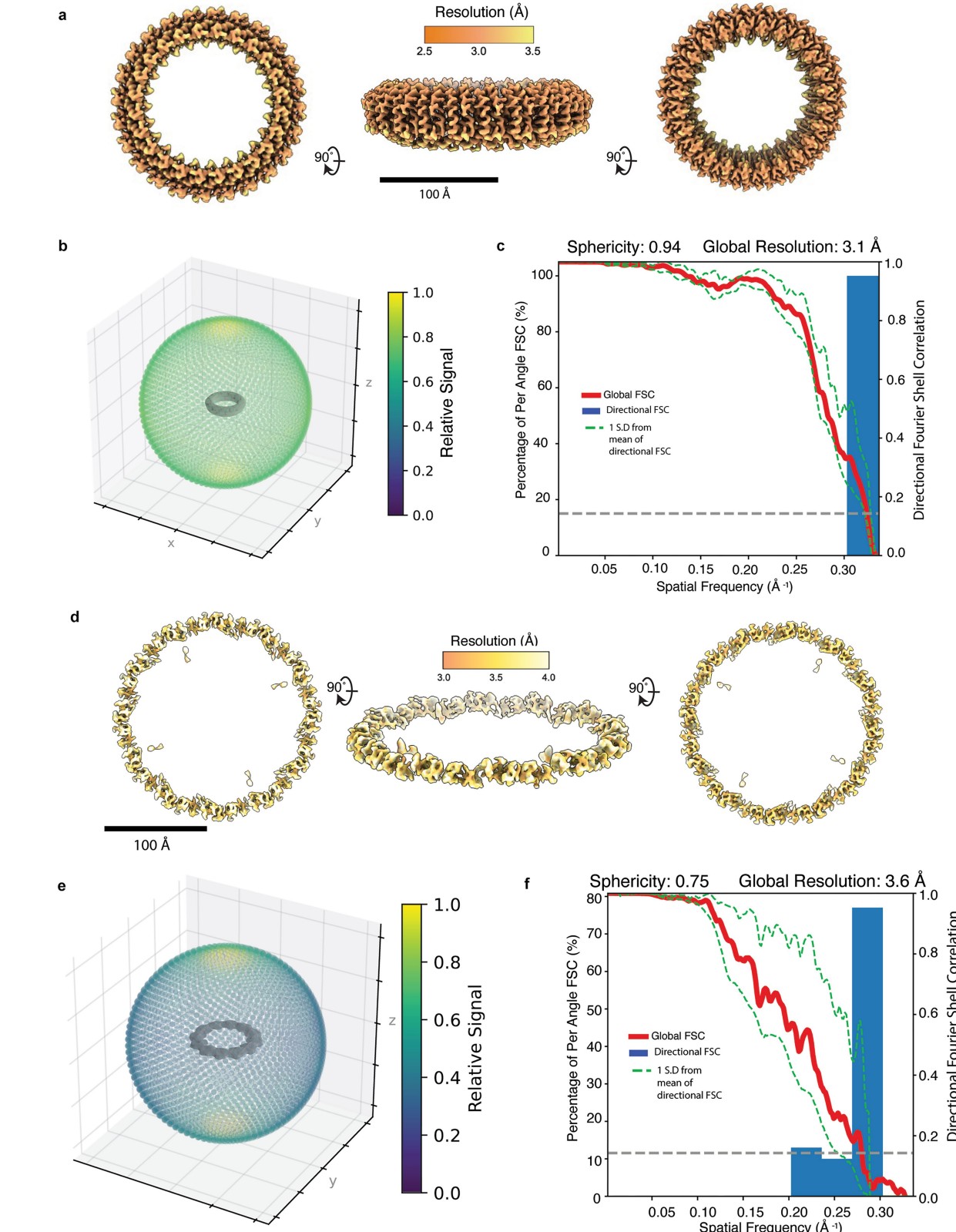

**Extended Data Fig. 5 | Estimates of resolution and angular sampling for the focused-refined maps of Gp77 and RAZR ZFD within the in vitro reconstituted RAZR:Gp77 complex.** (**a**) The map of Gp77 within the complex colored by local resolution, as estimated by the cryoSPARC MonoRes implementation.

(**b**) Viewing distribution plot of Gp77. (**c**) Directional resolution plot of Gp77. (**d**) The map of RAZR ZFD within the complex colored by local resolution, as estimated by the cryoSPARC MonoRes implementation. (**e**) Viewing distribution plot of RAZR ZFD. (**f**) Directional resolution plot of RAZR ZFD.

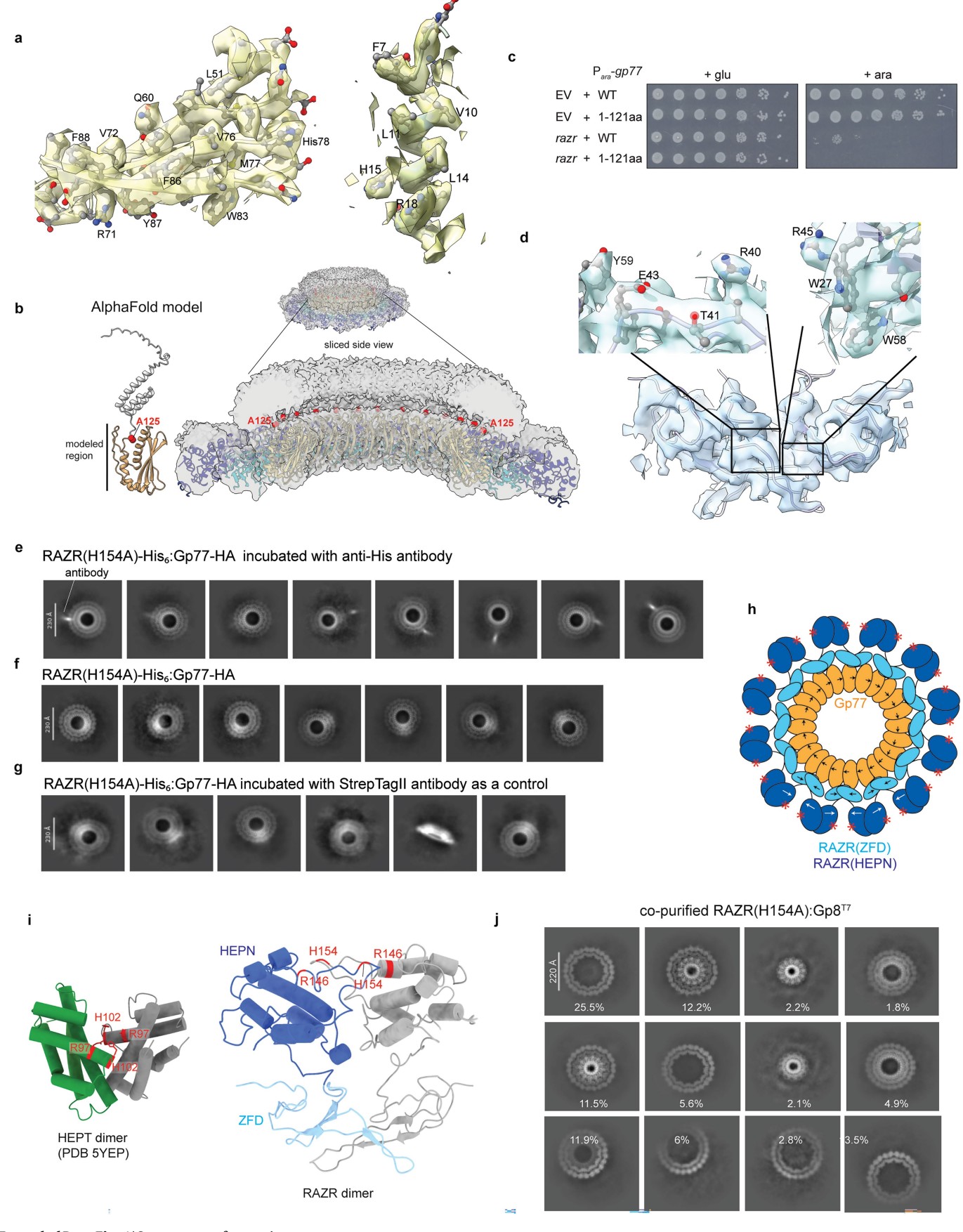

**Extended Data Fig. 6** | See next page for caption.

**Extended Data Fig. 6 | Structural analysis of RAZR in complex with Gp77 or Gp8$^{T7}$.** (**a**) Exemplary sharpened cryo-EM density map and atomic model of Gp77, showing residues 50–90 (chain IJ) and residues 5–20 (chain IA). (**b**) *Left*, AlphaFold model of Gp77, color-coded according to the region resolved in the cryo-EM structure. The last modeled residue (A125) and the C-terminal domain of Gp77 are highlighted in red and grey, respectively. *Right*, sliced side view of the complex (inset) showing the position of the last modeled Gp77 residue (A125), which points toward the apex of the complex at the ring-like density. (**c**) Serial dilutions of cells producing wild-type RAZR from its native promoter and Gp77 (full length or 1-121aa N-terminal domain) from an arabinose-inducible promoter on media containing glucose or arabinose. (**d**) Exemplary sharpened cryo-EM density map and atomic model of RAZR (ZFD), chains OD and OE. Insets show representative residues with resolved side-chain densities. (**e-g**) 2D class averages of RAZR(H154A)-His$_6$:Gp77-HA complexes incubated with (**e**) anti-His antibody, (**f**) buffer, and (**g**) anti-StrepTagII antibody. Antibody signal was observed only in panel (**e**), localized to the outermost ring where RAZR HEPN His-tags are positioned. (**h**) Schematic of RAZR:Gp77 complex, with arrows indicating orientation of subunits, and stars indicating activate sites of HEPN domain. (**i**) The crystal structure of HEPT dimer (*left*, PDB ID: 5YEP), and the structure of RAZR dimer colored by domains (*right*). The active site arginines and histidines are colored in red. (**j**) 2D class averages of co-purified RAZR(H154A) and Gp8$^{T7}$ complex calculated using 37,350 particles.

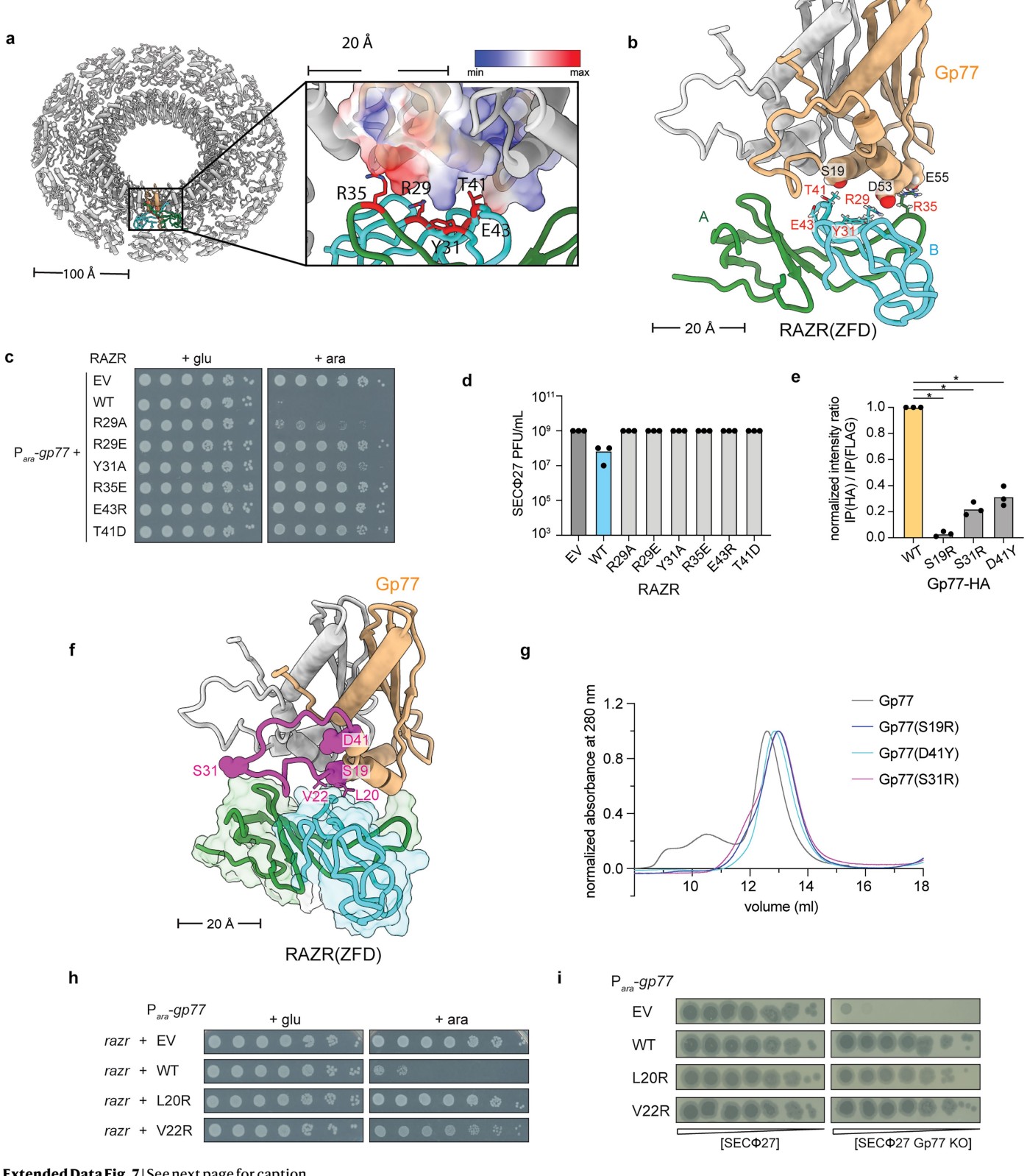

**Extended Data Fig. 7** | See next page for caption.

**Extended Data Fig. 7 | Interface analysis of RAZR and Gp77. (a)** *Left*, cartoon representation of Gp77 interacting with the ZFD of RAZR as in the structure of RAZR:Gp77 complex. *Right*, detailed interface of Gp77 and two protomers of RAZR(ZFD). Substituted residues in RAZR(ZFD) are highlighted in stick representation. Gp77 monomers are displayed as electrostatic surface representations, with red and blue indicating negatively and positively charged residues, respectively. **(b)** Detailed interface of Gp77 (orange) and two protomers of RAZR(ZFD) (green and cyan) as in the complex structure of RAZR:Gp77. Residues in Gp77 and RAZR involved in the interface are labelled. **(c)** Serial dilutions of cells producing wild-type or the indicated variant of RAZR from its native promoter and Gp77 from an arabinose-inducible promoter on media containing glucose or arabinose. **(d)** Summary of 3 independent replicates of phage spotting assay in Fig. 3b. **(e)** Quantification of the band intensity ratios of Gp77-HA (wild-type or variant) to RAZR(H154A)-FLAG in IP samples normalized to the wild-type sample. Data are from 3 independent replicates. Asterisks indicate $P = 10^{-7}$ (S19R), $10^{-5}$ (S31R), or $10^{-4}$ (D41Y) (unpaired two-tailed Student's t-test). Representative image shown in Fig. 3f. **(f)** Detailed interface of Gp77 (orange) and two protomers of RAZR(ZFD) (green and cyan) as in the complex structure of RAZR:Gp77. The loop of Gp77 formed by residues 19-41 are colored in pink with substituted residues labeled. **(g)** Size-exclusion chromatography analysis of the wild-type or the indicated variant of Gp77-His$_6$. **(h)** Serial dilutions of cells producing RAZR from its native promoter and Gp77 (wild-type or the indicated variant) from an arabinose-inducible promoter on media containing glucose or arabinose. **(i)** Serial dilutions of SECΦ27 phage lacking gene *77* or the wild-type phage spotted on lawns of cells harboring an empty vector or a plasmid expressing Gp77 (wild-type or the indicated variant) exogenously.

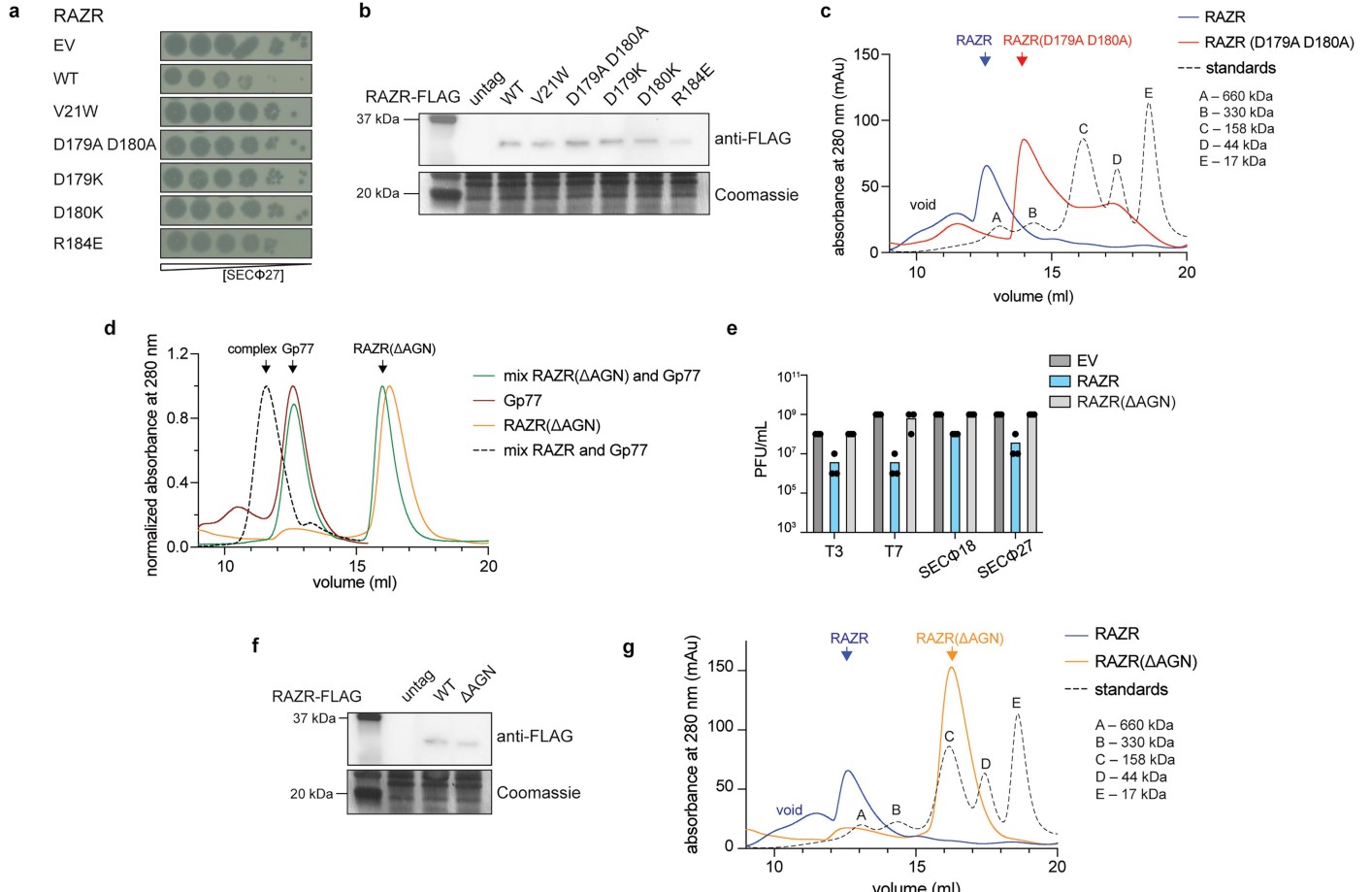

**Extended Data Fig. 8 | Oligomerization is important for RAZR activation.**
(**a**) Serial dilutions of SECΦ27 phage spotted on lawns of cells harboring an empty vector or a plasmid producing wild-type or the indicated variant of RAZR. (**b**) Immunoblot of FLAG-tagged RAZR expressed from its native promoter (wild-type or the indicated variant). Coomassie stain of the blot is included as a loading control. (**c**) Size-exclusion chromatography analysis of the wild-type or the D179A D180A variant of RAZR-His$_6$, with the protein standards included. (**d**) Size-exclusion chromatography analysis of RAZR(ΔAGN) or wild-type) and Gp77 mixed at a 1:1 molar ratio. (**e**) Summary of 3 independent replicates of phage spotting assay in Fig. 3j. (**f**) Same as in (**b**). (**g**) Size-exclusion chromatography analysis of the wild-type or the ΔAGN variant of RAZR-His$_6$.

**a**

RAZR^Ec 1 MAEYYPAVFEAEAFNCPHCGVYARQFWRRMYGSARELAQRTTEFRMSTCSHCGDDAYWYDGNMIIIPAAGNVELPNPDMPDNCKSDYM 87
RAZR^Kv 1 MAEYYPAVFEAEAFNCPYCNVYARQHWREMYGLANHSPSMKTDFCLSRCLHCEEKAYWYKGSMMIIPVSANIIEMPNSDMPEDCKSDYM 87

RAZR^Ec 88 EARSIINLSPKGAAALLRLCLQKLMVHLGEPGENINKDIRSLVQKGLPVRIQQAADICRIVGNQAVHPGEISLDDDPQLAHGLFKLL 174
RAZR^Kv 88 EARSIVNLSPKGAAALLRLCLQKLMIHLGEPGNNINADIKSLVEKGLPPRIQQAADICRIVGNQAVHPGEISLDDDPQLTHGLFKLL 174

RAZR^Ec 175 NIIVDDRITRPKEIIEAMFQSMPEGPRQGIENQDRQAREQQQAANE 219
RAZR^Kv 175 NIIVDDRITRPKEVEAMFNSMPERARKGIENRDK 208

■ identical ▨ similar ☐ not similar

**b**

**c**

**d**

Gp8^T7
(PDB: 7EY8)

**Extended Data Fig. 9 | Comparison of the RAZR homologs from *E. coli* and *K. variicola*.** (**a**) Sequence alignment of RAZR^Ec and RAZR^Kv with the domains labeled. (**b**) Serial dilutions of the indicated phage spotted on lawns of cells harboring an empty vector or a plasmid producing RAZR^Ec, RAZR^Kv, or the corresponding RAZR chimeras. (**c**) *Left*, schematic of the RAZR constructs.

*Right*, EOP data for the phages indicated when infecting cells producing RAZR^Ec, RAZR^Kv, or the corresponding RAZR chimeras. (**d**) Cartoon representation of the structure of Gp8^T7 (PDB ID: 7EY8) with residues substituted in the escape phage mutants labeled in red.

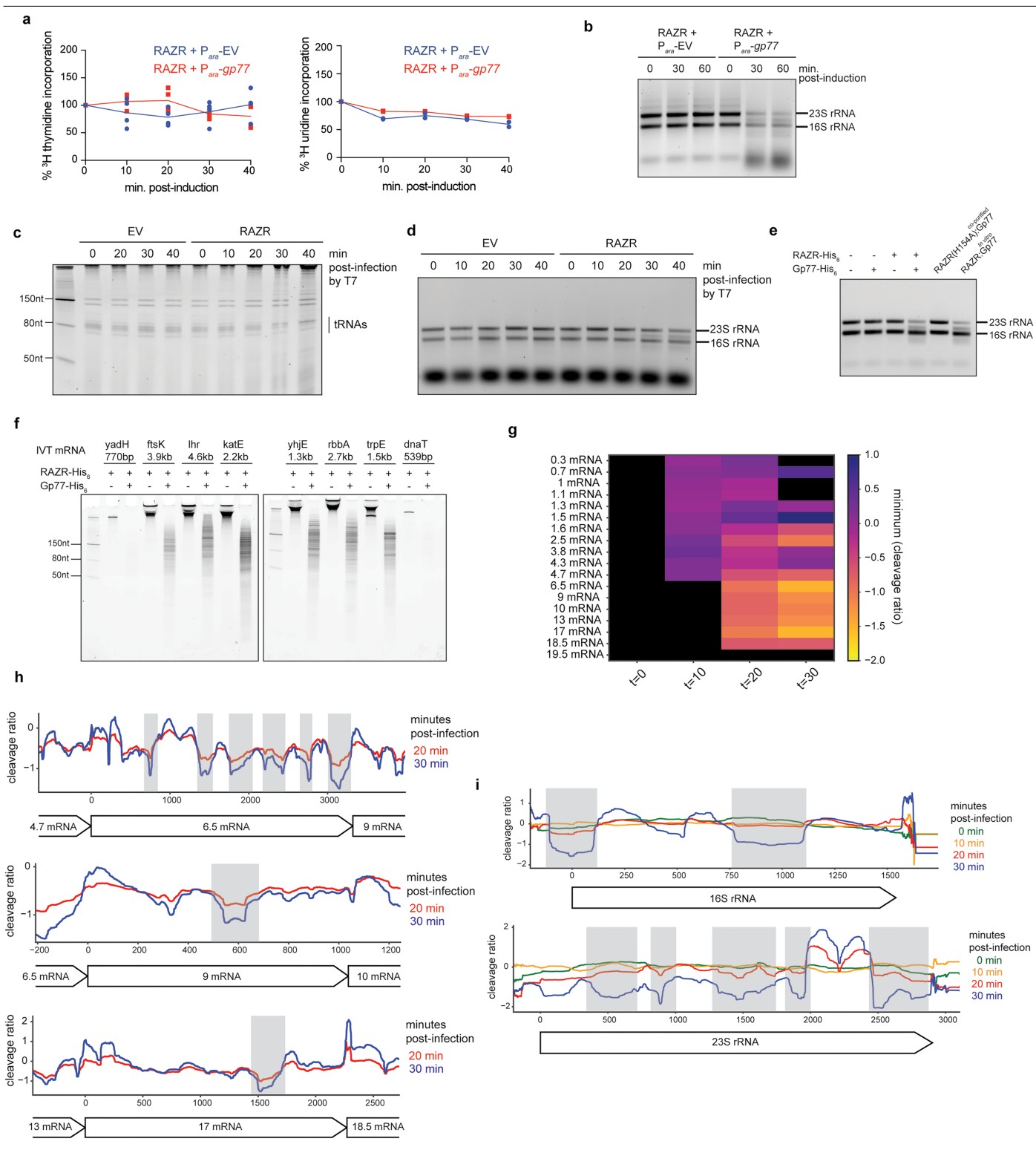

**Extended Data Fig. 10 | RAZR cleaves tRNAs, rRNAs, and mRNAs.** (**a**) Cells producing RAZR from its native promoter and Gp77 from an arabinose-inducible promoter or an empty vector were pulse-labeled with $^3$H-thymidine (*left*) or $^3$H-uridine (*right*) at the times indicated post-addition of arabinose. (**b**) RNAs were isolated from cells producing RAZR from its native promoter and Gp77 from an arabinose-inducible promoter or an empty vector at the times indicated post-addition of arabinose, and resolved on agarose gels to visualize rRNAs. (**c**) RNAs were isolated from cells producing RAZR from its native promoter or an empty vector following infection with T7 and resolved on TBE-urea gels to visualize tRNAs. (**d**) Same as in (**c**) except RNAs were resolved on agarose gels for visualization of rRNAs. (**e**) *E. coli* 70S ribosomes were incubated with purified RAZR-His$_6$ and Gp77-His$_6$, RAZR:Gp77 reconstituted in vitro, or the co-purified RAZR(H154A):Gp77 complex, and visualized on agarose gels. (**f**) In vitro transcribed mRNAs were incubated with purified RAZR-His$_6$ and Gp77-His$_6$ and visualized on TBE-urea gels. (**g**) Heatmap showing the minimum cleavage ratio of T7 transcript at 0, 10, 20, 30 min post-infection. Transcripts not well-expressed at a given time point are indicated in black. (**h**) Cleavage ratio of T7 transcripts at 20 min or 30 min post-infection, with gray boxes indicating potential cleavage regions. (**i**) Cleavage ratio of 16S and 23S rRNAs at 0, 10, 20, 30 min post-infection of T7, with gray boxes indicating potential cleavage regions.

**Extended Data Table 1 | Cryo-EM data collection, refinement, and validation statistics**

| | #1 Co-purified RAZR(H154A):Gp77 (EMD-72691) | #2 *In vitro* RAZR:Gp77 complex (EMD-72693) (PDB 9Y9C) | #3 Gp77 from *in vitro* RAZR:Gp77 complex (EMD-72631) (PDB 9Y6U) | #4 Focused refined RAZR ZFD map from *in vitro* RAZR:Gp77 complex (EMD-72692) |
|---|---|---|---|---|
| **Data collection and processing** | | | | |
| Magnification | 130,000 X | 130,000 X | 130,000 X | 130,000 X |
| Voltage (kV) | 300 | 300 | 300 | 300 |
| Electron exposure (e–/Å$^2$) | 47.62 for 0°-tilt data and 47.96 for 30°-tilt | 45.66 | 45.66 | 45.66 |
| Defocus range (μm) | -0.25 to -1.75 for 0°-tilt data and -0.75 to -2.5 for 30°-tilt data | -0.5 to -2.0 | -0.5 to -2.0 | -0.5 to -2.0 |
| Pixel size (Å) | 0.65 Å (binned by 2) | 0.65 Å (binned by 2) | 0.65 Å (binned by 2) | 0.65 Å (binned by 2) |
| Symmetry imposed | C$_{12}$ | C$_{12}$ | C$_{24}$ | C$_{12}$ |
| Initial particle images (no.) | 827,058 | 42,191 | 42,191 | 42,191 |
| Final particle images (no.) | 234,639 | 28,191 | 28,191 | 28,191 |
| Map resolution (Å) | 3.4 | 3.4 | 3.0 | 3.5 |
| FSC threshold | 0.143 | 0.143 | 0.143 | 0.143 |
| Map resolution range (Å) | 3.4-4.2 | 3.4-6.0 | 3.0-4.2 | 3.5-7.5 |
| | | | | |
| **Refinement** | | | | |
| Initial model used (PDB code) | NA | AlphaFold model (RAZR: AF-A0A0F3V1L6-F1) | AlphaFold model | NA |
| Model resolution (Å) | NA | 3.5 | 3.1 | NA |
| FSC threshold | NA | 0.143 | 0.143 | NA |
| Model resolution range (Å) | NA | 3.3-7.5 | 3.0-3.8 | NA |
| Map sharpening *B* factor (Å$^2$) | NA | -80 | -80 | NA |
| Model composition | | | | |
| Non-hydrogen atoms | NA | 56,830 | 23,664 | NA |
| Protein residues | NA | 7248 | 3000 | NA |
| Ligands | NA | 24 (Zn) | 0 | NA |
| *B* factors (Å$^2$) | | | | |
| Protein | NA | 221.4 | 88.2 | NA |
| Ligand | NA | 160.5 | NA | NA |
| R.m.s. deviations | | | | |
| Bond lengths (Å) | NA | 0.004 | 0.004 | NA |
| Bond angles (°) | NA | 1.086 | 0.939 | NA |
| Validation | | | | |
| MolProbity score | NA | 1.09 | 1.18 | NA |
| Clashscore | NA | 2.95 | 3.95 | NA |
| Poor rotamers (%) | NA | 0.57 | 0.93 | NA |
| Ramachandran plot | | | | |
| Favored (%) | NA | 98.76 | 99.19 | NA |
| Allowed (%) | NA | 1.22 | 0.81 | NA |
| Disallowed (%) | NA | 0.03 | 0.00 | NA |

# Reporting Summary

## Statistics

For all statistical analyses, confirm that the following items are present in the figure legend, table legend, main text, or Methods section.

| n/a | Confirmed | |
|---|---|---|
| ☐ | ☒ | The exact sample size (*n*) for each experimental group/condition, given as a discrete number and unit of measurement |
| ☐ | ☒ | A statement on whether measurements were taken from distinct samples or whether the same sample was measured repeatedly |
| ☐ | ☒ | The statistical test(s) used AND whether they are one- or two-sided *Only common tests should be described solely by name; describe more complex techniques in the Methods section.* |
| ☒ | ☐ | A description of all covariates tested |
| ☒ | ☐ | A description of any assumptions or corrections, such as tests of normality and adjustment for multiple comparisons |
| ☐ | ☒ | A full description of the statistical parameters including central tendency (e.g. means) or other basic estimates (e.g. regression coefficient) AND variation (e.g. standard deviation) or associated estimates of uncertainty (e.g. confidence intervals) |
| ☐ | ☒ | For null hypothesis testing, the test statistic (e.g. *F*, *t*, *r*) with confidence intervals, effect sizes, degrees of freedom and *P* value noted *Give P values as exact values whenever suitable.* |
| ☒ | ☐ | For Bayesian analysis, information on the choice of priors and Markov chain Monte Carlo settings |
| ☒ | ☐ | For hierarchical and complex designs, identification of the appropriate level for tests and full reporting of outcomes |
| ☒ | ☐ | Estimates of effect sizes (e.g. Cohen's *d*, Pearson's *r*), indicating how they were calculated |

*Our web collection on statistics for biologists contains articles on many of the points above.*

## Software and code

Policy information about availability of computer code

| Data collection | Consurf 2016 web server for identifying homologs EPU (Thermo Fisher Scientific, v2.12.1, v.3.11.0 ) for cryo-EM data collection |
|---|---|
| Data analysis | Geneious 2022.0.2 for genome and sequence alignment cryoSPARC (v.4.5.3 and v.4.7.0) for cryo-EM data processing UCSF ChimeraX(v.1.6) for cryoEM data processing and structural visualization Coot (v.0.9.4) and Phenix (v.1.21.2-5419) for structural model building 3DFSC server for FSC calculation Fiji (v 2.1.0/1.53c) for colony size quantification |

For manuscripts utilizing custom algorithms or software that are central to the research but not yet described in published literature, software must be made available to editors and reviewers. We strongly encourage code deposition in a community repository (e.g. GitHub). See the Nature Portfolio guidelines for submitting code & software for further information.

## Data

Policy information about availability of data

All manuscripts must include a data availability statement. This statement should provide the following information, where applicable:
- Accession codes, unique identifiers, or web links for publicly available datasets
- A description of any restrictions on data availability
- For clinical datasets or third party data, please ensure that the statement adheres to our policy

Cryo-EM maps and associated atomic models are available with the following PDB and EMDB codes: 9Y9C, 9Y6U (PDB); and 72693, 72631, 72692, 72691 (EMDB). Sequencing data are available in the Sequence Read Archive (SRA) under BioProject PRJNA1207560. All other data are available in the manuscript or the supplementary materials. Other previously published structures are available in PDB. UniRef90 database is publicly available. Reference bacteria and phage genomes are publicly available: MG1655 (NC_00913.2), SECΦ27 (NC_047938.1), T3 (NC_047864.1), T7 (NC_001604.1), and SECΦ18 (NC_073071.1). Materials including strains and plasmids are available upon reasonable request.

## Research involving human participants, their data, or biological material

Policy information about studies with human participants or human data. See also policy information about sex, gender (identity/presentation), and sexual orientation and race, ethnicity and racism.

| | |
|---|---|
| Reporting on sex and gender | N/A |
| Reporting on race, ethnicity, or other socially relevant groupings | N/A |
| Population characteristics | N/A |
| Recruitment | N/A |
| Ethics oversight | N/A |

Note that full information on the approval of the study protocol must also be provided in the manuscript.

# Field-specific reporting

Please select the one below that is the best fit for your research. If you are not sure, read the appropriate sections before making your selection.

☒ Life sciences          ☐ Behavioural & social sciences          ☐ Ecological, evolutionary & environmental sciences

For a reference copy of the document with all sections, see nature.com/documents/nr-reporting-summary-flat.pdf

# Life sciences study design

All studies must disclose on these points even when the disclosure is negative.

| | |
|---|---|
| Sample size | No sample size calculation was performed. Sample sizes were chosen based on the number needed to reliably determine differences between groups, and most bacterial experiments were performed with isogenic strains. Given large effect sizes, we chose to replicate experiments 2-3 times as is routine to simply indicate reproducibility. |
| Data exclusions | No data were excluded from analysis. |
| Replication | All experimental findings were repeated at least twice. All reported results were successfully reproduced. |
| Randomization | No experimental groups or control groups were subjectively chosen and there are no covariates to control for as experiments were done in isogenic strains. No randomization is required. |
| Blinding | Blinding was not required because all data were obtained objectively and had strong effect sizes over multiple independent replicates. |

# Reporting for specific materials, systems and methods

We require information from authors about some types of materials, experimental systems and methods used in many studies. Here, indicate whether each material, system or method listed is relevant to your study. If you are not sure if a list item applies to your research, read the appropriate section before selecting a response.

## Materials & experimental systems

| n/a | Involved in the study |
|---|---|
| ☐ | ☒ Antibodies |
| ☒ | ☐ Eukaryotic cell lines |
| ☒ | ☐ Palaeontology and archaeology |
| ☒ | ☐ Animals and other organisms |
| ☒ | ☐ Clinical data |
| ☒ | ☐ Dual use research of concern |
| ☒ | ☐ Plants |

## Methods

| n/a | Involved in the study |
|---|---|
| ☒ | ☐ ChIP-seq |
| ☒ | ☐ Flow cytometry |
| ☒ | ☐ MRI-based neuroimaging |

# Antibodies

| Antibodies used | HA-tag (C29F4) rabbit mAb (Cell Signaling Technology #3724)<br>(FLAG) DYKDDDDK Tag (D6W5B) Rabbit mAb (Cell Signaling Technology, #14793)<br>6x-His Tag Monoclonal Antibody (HIS.H8) (Invitrogen, catalog #MA1-21315)<br>Anti-Strep, Catalog # MA5-37747, Invitrogen, Lot # XC3529122 |
|---|---|
| Validation | Antibodies have been validated based on manufacturer's website:<br>anti-HA: validated by "western blot analysis of extracts from HeLa cells, untransfected or transfected with either HA-FoxO4 or HAAkt3.<br>anti-FLAG: validated by "western blot analysis of extracts from 293T cells, mock transfected (-) or transfected with DYKDDDDK-GFP".<br>In addition, all western blotting experiments include controls with untagged proteins as internal validation for antibodies.<br>anti-His: validated by 'western blot analysis of His-tagged protein in E.coli lysates."<br>anti-Strep: The Anti-Strep antibody was validated by Western blot on a recombinant STREP II–tagged protein. Ref: Mossman, D., et al., Cell (2023)<br>All antibodies were used according to the manufacturer's guidelines. |

# Plants

| Seed stocks | N/A |
|---|---|
| Novel plant genotypes | N/A |
| Authentication | N/A |

