## [Peer Review File · Nature]

Bacterial immune activation via supramolecular assembly with phage triggers

Corresponding Author: Professor Michael Laub

Version 0:

Reviewer comments:

Referee #1

(Remarks to the Author)

Zhang et al describe the mechanism of a bacterial antiphage defense system they name RAZR. The authors demonstrate that RAZR is a promiscuous RNase that degrades single-stranded RNA in response to direct binding between RAZR and surprisingly diverse phage proteins. To explain activation, the authors present a combination of structural modeling and experimental cryo-EM analyses (including 2D class averages of RAZR in complex with distinct phage proteins, and a partial low-resolution 3D model of RAZR in complex with the phage SECphi27 protein Gp77) supporting oligomerization of RAZR around viral ring-like assemblies. Overall, the authors' genetic, biochemical, and virology results are stunning, providing an incredibly interesting model for how an immune protein can use shape-based recognition to discriminate disparate viral protein that form ring-like structures. The 2D cryo-EM data add considerably support for the model, but the experimental 3D cryo-EM data are comparatively weak. The manuscript's structural biology conclusions are mostly reliant on AlphaFold3 predicted models that do not appear to agree with the partial experimental density. Overall, the manuscript represents an incredibly important advance for the field, but the authors may want to consider reframing how the structural biology data are presented given limitations in the current data.

Major points

1) The 3D cryo-EM data of RAZR in complex with the phage SECφ27 protein Gp77 are insufficient to support the claims as written in the paper.

The map contains clear density for an inner ring of Gp77, but only partial density that is not fully interpretable for the middle ring putatively assigned as the RAZR ZFN domain and the outer ring putatively assigned as the RAZR HEPN domain. Particularly concerning is that there is limited apparent density for the contacting interface between the phage protein and RAZR, and the overall proposed domain orientations do not appear to be unambiguously supported from the current experimental structural information alone.

The authors state that the structural data are ambiguous (for example Lines 182 "the structural resolution of the HEPN domains was low" and 213 "Although the resolution is insufficient to resolve side-chain density"). However, long sections of the manuscript results rely on the cryo-EM data as the main support for statements that are written in definitive language including "RAZR dimerized through extensive interactions between adjacent ZFDs, with additional contributions from the HEPN domains" (Line 181), "comparison of the AlphaFold-predicted RAZR tetramers and the RAZR proteins in our complex structure suggested a relative movement of the HEPN domain and the adjacent RAZR dimer" (Line 188), and "Residues R29, T41, and E43 of the other ZFD (protomer B) contribute to the interface via electrostatic interactions and hydrogen bonding with Gp77". The mutagenesis studies are then therefore difficult to interpret in the absence of strong supporting experimental structural data.

Overall, my sense is that the authors' interpretation of domain placement and orientation is likely correct, but I don't understand how the data can allow one to be confident that other orientations and mechanisms of recognition can be excluded. The strongest support for domain orientations, interfacing residues, and potential mechanism of interaction appears to be derived from the AF3 structure predictions and not from the cryo-EM data themselves. It may be more accurate

to describe the structure as a hybrid model derived from AF3 prediction and partial 3D cryo-EM data.

The manuscript would be significantly improved with further experimental validation of domain orientations within the RAZR-Gp77 model. The authors appear to have already invested significant effort to improve the cryo-EM maps, and it is understandable that further direct structural information may not be possible to obtain. The authors could consider cross-linking mass spectrometry or other biophysical methods used to support hybrid models, or alternatively re-write the structure section to more clearly state limitations of the current data.

Minor points

2) The data demonstrating RAZR catalyzes degradation of mRNA, tRNA, and rRNA are convincing. Although not necessary, it would be interesting if the authors could perform their RNA-seq analysis during phage infection as a possible experiment to help determine if one of the RNA targets may be more important in vivo for phage inhibition.

3) Related to point 2, in addition to the nicely presented reactions with and without Gp77 addition, the authors should consider testing RAZR active-site mutations in one of the RNA degradation assays in Figure 5 or Figure S10.

4) Please include DALI z-score or another measurement of structural similarity to help compare RAZR with well-characterized HEPN domains.

5) Fig. 1E - It would be helpful to include more information about genomic location or conservation of gp77 from SECphi27. Also, does SECphi27 have a separate portal protein that forms a similar ring structure? Is it possible that gp77 is the portal protein for SECphi27? Do phages ever encode both of these RAZR triggers?

6) It is not clear from the model figure how the authors envision RAZR transitions from a linear-like filament to a ring-like assembly upon phage recognition. No further experiments are necessary here, but do the authors envision this process is through direct transition of filaments wrapping around a phage target, or is it that inactive filaments undergo spontaneous disassembly and subunits of RAZR can reassemble as rings around a target?

I hope the authors will find my comments useful, thank you for the opportunity to read this exciting manuscript.

Philip Kranzusch

Referee #2

(Remarks to the Author)

Zhang et al. here study the previously identified (now renamed) RAZR system of phage immunity in bacteria. They here show that this system activates in response to two distinct phage components. The first, GP77 from phage SECP27 is a poorly characterized protein that might be involved in recombination. The second class are portal proteins that inject phage DNA into the phage capsid; three examples are shown, GP8 from phage T7, GP31 from phage T3, and GP21 from phage SECP18. Both of these types of proteins are known to form roughly 170 angstrom rings. RAZR forms an oligomer linear curved arc in its inactive state, but remarkably it circularizes around its target protein rings with its zinc finger domain on the interior interacting with the target. This large oligomer now correctly orients the RAZR exterior HEPN domain such that it activates RNase enzymatic activity. This degrades tRNA, rRNA, and mRNA, causing bacterial death that prevent phage replication, and does not degrade DNA. Point mutations are shown in multiple different interfaces and the linker to disrupt the function. They show specificity of this system by examining RAZR from *E. coli* compared to *Klebsiella* and perform chimeric swaps that show that each RAZR is able to identify specific phages.

Overall, the authors present a remarkable discovery and their conclusion are definitively proven by the experiments. From my knowledge, the specific structures or molecules or events that activate anti phage defense are often uncertain. The authors definitively identify two phage proteins that form structures that activate RAZR. One of the biggest surprises for me is that RAZR detects two distinct phage proteins that have no amino acid similarity. This detection is accomplished by looking for the size and shape of these ring forming proteins as the "pattern" that is recognized. I am not aware of mammalian systems that detect broad classes of non-self molecules by their size and shape, so this is a novel discovery. On the other hand, the RNase enzymatic activity of RAZR is quite reminiscent of many nucleic acid degrading enzymes in mammalian systems, perhaps most notably RNase L that similarly degrades RNA during viral infection. This RAZR system is also unique in that it can detect active phage replication and causes bacterial cell death and has exquisite specificity in contrast to other anti phage systems that may detect bacterial stress (e.g. loss of ATP activating OLD nucleases) as a less specific marker. I have a few comments to possibly improve the paper.

Major comments

1. The one confusing aspect of the model is that RAZR appears to have little specificity for the exact amino acid sequence because it detects both GP77 and the portal rings (that have no sequence identity). Yet RAZR simultaneously is sensitive to point mutations in its targets, arguing that it is very specific for the amino acids. The data seems very definitive that both of these are true, but how can this be the case? My thought is that the RAZR interface with GP77 or portal proteins is very weak in each individual monomer. If so then RAZR could have many weak interactions with many proteins. Then the interaction only becomes strong if multiple targets oligomerize, creating a large interaction interface. Then once the circle of RAZR is complete, the last interaction to link the first RAZR to the last to complete the circle provides another strong interaction interface (assuming that the RAZR:RAZR interface is strong). The authors may or may not agree with those ideas, but

regardless they need to address the point of simultaneous nonspecificity and amino acid substitution sensitivity in the discussion.

2. The authors are concluding that RAZR attacks tRNA, rRNA, and mRNA. However, a single attack on one of these species would be sufficient to halt phage replication and kill the bacteria, so it does not have to be all three equally. In particular, rRNA are generally folded into ribosomes and might be quite inaccessible. The more physiologic experiments are placed in the supplement in Figure S10b-c, where bacteria are infected with phage. However, the degradation of tRNA (Figure S10b) and rRNA (Figure S10c) is unimpressive. This leads me to speculate that the true target are mRNAs, which would be the best antiviral target in my opinion. Figure S10d shows impressive degradation of in vitro transcribed mRNA. Figure 5G shows impressive effects in *E. coli* bacteria upon the mRNA pool. I think this can be resolved by one additional experiment, which is to repeat Figure S10b-c and probe additionally for a few specific mRNA species by northern blot, such as those shown to be potential candidates in Figure S10d.

3. Figure 3g-h and S8a-c. The experiments showing that the RAZR dimer/dimer interface is important carry a caveat that these mutations could cause folding problems in the active site, resulting in an enzymatically dead protein. I cannot think of a counter screen that would demonstrate that the point mutants are simply not at all capable of folding. If these proteins could still interact with GP77 or portal proteins by immunoprecipitation, this would show some functionality to the mutant protein. This should work for the D179/180A mutant because it still appears to form a half sized oligomer (Figure S8c).

4. The authors state that RAZR does not recognize the portal protein of the SECP27 (while simultaneously recognizing GP77 of this phage). It seems odd that a phage where the portal was not detected would simultaneously have another protein that is detected. Are there GP77 homologs in the other three phages, and have these been tested for a RAZR response? This should be discussed.

Minor comments

1. I think the authors did not comment much about the fact that GP77 is a 24-mer, whereas the portal proteins are 12-mers. Is 12-fold symmetry the key to RAZR function? That its targets must have 12-fold (or multiples thereof) symmetry in order for RAZR to oligomerize around them?

2. The authors claim that RAZR targets are of a discreet size and that no endogenous complexes would be the same size and shape. With this in mind, there are many polymeric structures in bacteria that form circular structures. It should be easy to look at the known sizes of some of these to determine if they are potential RAZR targets that would need to not be detected, or whether all the potential targets are not encoded by bacteria that encode RAZR. I can think of several ring-like bacterial structures that would be worth considering: the cytosolic components of T3SS, flagellar basal body, T4SS, T6SS. In fact, T6SS has a whole circular complex that crosses the width of the bacterial cytosol. I am sure there are many more that are known as well. It may be worth commenting upon.

3. It seems fairly easy to evade RAZR with point mutations. Can the authors comment upon why these phages have not already evolved to evade RAZR? Are there RAZR negative *E. coli* that are the natural hosts for these phages? Also, are there phages that infect the strain of *E. coli* that encodes RAZR? Such phages should have evolved to evade RAZR detection. Did the authors survey of phages include others that have portal proteins or GP77 homologs but against which RAZR fails? If so, these would be worth mentioning.

4. It should be possible to generate mutations in GP77 or a portal protein that prevents ring formation. This should prevent RAZR activation, and would support the conclusion that the ring is detected. I mention this because it would be very uncomfortable for the authors if a monomeric GP77 or portal was subsequently found to activate RAZR. That said, I think that is very unlikely that monomeric proteins would interact with RAZR, and given that it is not essential for the current paper.

5. Figure 2E, is it possible to annotate the active site residues so that the reader can see where they are and how they move (instead of a bar) that is only showing the AF3 predicted active site. It is hard to see precisely.

6. Figure 4E. I assume that the whole phage was sequenced and that only one mutation was found, and it was the one described for Figure 4E. If so, please state that.

7. Based on the P22 portal structure and its predicted position with the phage capsule published for example by Cingolani (doi: 10.1038/ncomms14310), would the authors predict that RAZR is incorporated into forming phage heads? Or if the portal complex assembled first and then the capsid assembles around it or dock onto it, would RAZR prevent the assembly of the capsid and portal into a phage head?

8. I am confused about what Gp77 does, and why it forms a ring. From Line 161 the authors mention that the ring structure of Gp77 is similar to another phage, and that this ring may be important for circularizing the genome. Could the authors provide some speculation on why a ring structure might be needed to circularize the genome? No reason for this comes to my mind as a reader immediately. And this property seems to be the key to RAZR function. The explanation hypothesized by reference 28 published in 1983 could be incorrect. In contrast, for the portal proteins it is very apparent from prior work why they form the ring structure as part of their structure in phage assembly so that is a very satisfying rationale for their detection by RAZR.

9. Figure 2B. I suggest using the same magnification for each panel, thus allowing the reader to directly compare the size, for example, of the rings of Gp77 to the size of those seen in the RAZR-Gp77 structure. This would be a more intuitive presentation (sacrificing resolution of the Gp77 alone structure, but to the same extent as is lost by shrinking the co-structure).

10. Figure S7D is remarkable in contrast to Figure 3C. Is there space to move it to the main figure? I wonder if Figure 3B could be moved to the supplement to make space, as the take home for me from 3B and 3C was the same in that the mutations seem to all have a big effect. In contrast, 3C vs S7D was quite impressive showing specificity.

11. In Figure 4e, it was intriguing to see that the chimeric protein on the far right is stronger in defense than the *E. coli* RAZR against T7-Kv clones 1 and 2. Can the authors speculate as to why the chimera is superior in a way that appears independent of the ZFD?

12. You may want to mention the similarities to RNAse L in the mammalian system in your discussion.

- Ed Miao

Referee #3

(Remarks to the Author)

I co-reviewed this manuscript with one of the reviewers who provided the listed reports.

Referee #4

(Remarks to the Author)

In this study authors provide biochemical and structural characterization of the RAZR antiviral defense system. Most notably, they show a novel activation mechanism of antiphage defense that relies on the RAZR recognition of circular assemblies of phage proteins through the RAZR ZFD domains triggering the transition of linear RAZR filaments into ring-shaped structures and activation of HEPN RNase. Activation of antiviral defense system and discrimination of the self vs non-self through the shape recognition certainly adds a new dimension to the currently known mechanisms for sensing phage infection by antiphage defense proteins.

Overall, the study is well thought out, technically sound and well presented, providing compelling evidence for the proposed mechanism.

Points that should be addressed by the authors are listed below.

1. An important question that remains open is how the formation of the circular structure on the circumference of the gp77 disc leads to the activation of the RNase activity of the HEPN domains. The overlay of the linear-like AlphaFold3 prediction on a fragment of the circular structure is presented in Fig. 2E. However, the overlay is based on the ZFD domains, not the HEPN domains, making it difficult to judge the overall similarity between the AF3-predicted HEPN structure and the experimentally modeled HEPN domain. It is also unclear whether the transition from the linear AF3-predicted filament to the experimentally observed ring-shaped structure results in the rearrangement of RAZR (HEPN) homodimers relative to each other or to RAZR (HEPN) domains within the dimers, leading to the rearrangement/assembly of the catalytic centers at the HEPN domain interfaces.
2. In addition to the expected HEPN-HEPN domain interactions within the RAZR primary dimers observed in the structure, the primary HEPN domains seem to form additional interaction surfaces between the primary dimers. How large are these interaction surface areas? Which structural elements that form these interfaces could be reliably modeled into the density, and how does this overlap with the AlphaFold3 model shown in Fig. 3G, which was used to design interface-disrupting point mutations?
3. Authors show that activated HEPN RNase cleave various RNA molecules including tRNA. Where tRNA cleavage occurs?
4. Would the authors predict that all potential activators of the RAZR system would form rings/discs with 12-fold symmetry? The RAZR system is activated by unrelated proteins from different phages (Gp77/Erf-like protein from SECphi27, Gp8/Gp31 portal proteins from T7/T3). However, can the authors explain, based on structural predictions, why the SECphi27 portal protein and T3/T7 Gp77 homologs (if present) are not activators for the RAZR system? Can some other structural requirements (beyond the 170Å outer diameter) for the activator protein be detected based on these observations?
5. The cryo-EM processing figure (Fig. S11) does not provide the local resolution for the final C1 reconstruction obtained after symmetry expansion/3D classification. Since the obtained C1 map was used to build the atomic model of the HEPN domain (Fig. S5g), it should be deposited to the EMDB.
6. The ConSurf-colored RAZR sequence and AF3 model are provided in Fig. S1d and Fig. 4d. Please provide more information on the RAZR sequences that were used to generate the respective data in the Methods section.
7. The conserved linker (aa 66-79), which is subjected to deletion analysis (lines 278-281), should be indicated in Fig. S1d. Does the 3aa linker deletion characterized in your study alter the AF3 prediction of the RAZR filament? Does AF3 co-fold Gp77 with RAZR (say, in a 4:4 subfragment)?
8. Lines 243-244: The "slightly smaller colonies" observation, referring to data in Fig. 1g, could be supported by quantification of actual colony sizes in the +glu/+ara experiments.
9. Showing denoised micrographs in Fig. S3f and Fig. S4a could provide a better representation of the actual particles observed in the respective samples.
10. Throughout the Methods section, the oligomeric states of the proteins/complexes to which the listed concentrations refer should be indicated.

Referee #5

(Remarks to the Author)

I co-reviewed this manuscript with one of the reviewers who provided the listed reports.

Version 1:

Reviewer comments:

Referee #1

(Remarks to the Author)

The authors' revised manuscript is greatly improved, with new experimental data addressing each reviewer critique. Notably, the primary structural data and description of the cryo-EM models are each significantly improved. I recommend publication and congratulate the authors again on an incredible discovery.

Referee #2

(Remarks to the Author)

The authors have addressed all my comments and questions. I have one additional suggestion, which the authors can take or not. I suggest adding discussion of a conceptual comparison between RAZR and the mammal TRIM5 defense against retroviral infection. One of the big concepts with RAZR is that the defense is recognizing viruses based upon their shape and symmetry. Similarly, TRIM5 detects viral capsids, I think also dependent upon their shape and symmetry. See a very nice review: doi 10.1038/s41579-019-0225-2. This may be worth discussing the conceptual similarities. Certainly TRIM5 is not looking for circular shapes, but rather capsid shapes. The aforementioned review sums it up nicely: "TRIM5 α can match both the symmetry and the spacing of the capsid lattice and generate powerful avidity effects".

Referee #3

(Remarks to the Author)

I co-reviewed this manuscript with one of the reviewers who provided the listed reports.

Referee #4

(Remarks to the Author)

The authors have adequately addressed our concerns. Given the limited resolution of the outer ring in the cryo-EM maps, accurate modeling of the catalytic HEPN domains was not possible. Nevertheless, the more conservative interpretation of the cryo-EM data introduced in the revised manuscript, presenting the structure as a hybrid model derived from AlphaFold predictions and cryo-EM density, still provides compelling evidence supporting the proposed model of RAZR activation.

Referee #5

(Remarks to the Author)

I co-reviewed this manuscript with one of the reviewers who provided the listed reports.

We thank all three reviewers for their constructive comments and questions. Below we respond to each query and indicate, when appropriate, how the text and figures have been updated.

Referee #1 (Remarks to the Author):

Zhang et al describe the mechanism of a bacterial antiphage defense system they name RAZR. The authors demonstrate that RAZR is a promiscuous RNase that degrades single-stranded RNA in response to direct binding between RAZR and surprisingly diverse phage proteins. To explain activation, the authors present a combination of structural modeling and experimental cryo-EM analyses (including 2D class averages of RAZR in complex with distinct phage proteins, and a partial low-resolution 3D model of RAZR in complex with the phage SECphi27 protein Gp77) supporting oligomerization of RAZR around viral ring-like assemblies. Overall, the authors' genetic, biochemical, and virology results are stunning, providing an incredibly interesting model for how an immune protein can use shape-based recognition to discriminate disparate viral protein that form ring-like structures. The 2D cryo-EM data add considerably support for the model, but the experimental 3D cryo-EM data are comparatively weak. The manuscript's structural biology conclusions are mostly reliant on AlphaFold3 predicted models that do not appear to agree with the partial experimental density. Overall, the manuscript represents an incredibly important advance for the field, but the authors may want to consider reframing how the structural biology data are presented given limitations in the current data.

Major points

1) The 3D cryo-EM data of RAZR in complex with the phage SECΦ27 protein Gp77 are insufficient to support the claims as written in the paper.

The map contains clear density for an inner ring of Gp77, but only partial density that is not fully interpretable for the middle ring putatively assigned as the RAZR ZFN domain and the outer ring putatively assigned as the RAZR HEPN domain. Particularly concerning is that there is limited apparent density for the contacting interface between the phage protein and RAZR, and the overall proposed domain orientations do not appear to be unambiguously supported from the current experimental structural information alone.

The authors state that the structural data are ambiguous (for example Lines 182 “the structural resolution of the HEPN domains was low” and 213 “Although the resolution is insufficient to resolve side-chain density”). However, long sections of the manuscript results rely on the cryo-EM data as the main support for statements that are written in definitive language including “RAZR dimerized through extensive interactions between adjacent ZFDs, with additional contributions from the HEPN domains” (Line 181), “comparison of the AlphaFold-predicted RAZR tetramers and the RAZR proteins in our complex structure suggested a relative movement of the HEPN domain and the adjacent RAZR dimer” (Line 188), and “Residues R29, T41, and E43 of the other ZFD (protomer B)

contribute to the interface via electrostatic interactions and hydrogen bonding with Gp77". The mutagenesis studies are then therefore difficult to interpret in the absence of strong supporting experimental structural data.

Overall, my sense is that the authors' interpretation of domain placement and orientation is likely correct, but I don't understand how the data can allow one to be confident that other orientations and mechanisms of recognition can be excluded. The strongest support for domain orientations, interfacing residues, and potential mechanism of interaction appears to be derived from the AF3 structure predictions and not from the cryo-EM data themselves. It may be more accurate to describe the structure as a hybrid model derived from AF3 prediction and partial 3D cryo-EM data.

The manuscript would be significantly improved with further experimental validation of domain orientations within the RAZR–Gp77 model. The authors appear to have already invested significant effort to improve the cryo-EM maps, and it is understandable that further direct structural information may not be possible to obtain. The authors could consider cross-linking mass spectrometry or other biophysical methods used to support hybrid models, or alternatively re-write the structure section to more clearly state limitations of the current data.

We thank the reviewer for his thoughtful assessment of our structural data. To address the points raised, we have now combined improved cryo-EM maps and additional structural studies with pull-down and phenotypic assays to further confirm our domain assignments.

In addition to our previous structure of co-purified RAZR(H154A):Gp77, we have now determined cryo-EM maps of *in vitro* reconstituted RAZR:Gp77 complexes and performed focused refinements of different regions (Fig. S5-S7). Overall, the new maps showed reduced anisotropy (sphericity score 0.94) compared with our previous dataset and improved resolution. Particle picking with the neural-network-based program Topaz recovered both top- and side-view orientations, enabling higher-quality 3D reconstructions. Focused refinements improved the resolution of inner and central rings but not the outermost layer. The inner ring density was clearly assigned to 24 copies of the Gp77 N-terminal domain (residues 1–125), supported by well-resolved side-chain densities for bulky residues (Fig. S8A). The last modeled residue of Gp77 suggested that the remainder of it occupies the ring-like density at the apex of the assembly that is not involved in interactions with the rest of the complex (Fig. S8B). Consistent with this interpretation, deleting this region did not disrupt RAZR activation, confirming that it is dispensable (Fig. S8C).

For the central ring, focused refinement yielded a 3.5 Å map into which the RAZR zinc-finger domain (ZFD) domain could be confidently fitted. Local-EM fitting in Phenix (Reed, R., et al., *Acta Crystallogr D Struct Biol.* 2024), which uses a Fourier-space likelihood-based target, confirmed this assignment: when both the ZFD and HEPN domains were included in the fitting, only the ZFD domain converged to the central ring density, whereas

the HEPN domain did not. Continuous density connecting the central and outer rings further supported the assignment of the outer ring to HEPN domains. Although this region was at lower resolution, three independent validation strategies confirmed our interpretation: (i) docking of the AlphaFold-predicted HEPN domain revealed an arrangement consistent with canonical HEPN RNase-active dimers (Fig. S8I, Jia et al., 2018); (ii) incubating an anti-His₆ antibody with RAZR-His₆:Gp77 complexes revealed antibody binding to the outermost ring in 2D class averages, consistent with the His₆-tag location at the C-terminus of the HEPN domain (see new Fig. S8E-G); and (iii) mutational analysis at the RAZR:Gp77 interface demonstrated that structurally defined residues are critical for complex formation and interaction (Fig. 3A-C, and 3F). Further, experiments with chimeric RAZRs demonstrated that the ZFD determines phage defense specificity (Fig. 4C), providing strong functional support for our assignment of this domain as the central ring that contacts Gp77 directly. Finally, mass spectrometry confirmed that no proteins other than Gp77 and RAZR were present, excluding the possibility of contamination. Taken all together, multiple lines of evidence are consistent with a model in which the ZFD of RAZR faces the Gp77 ring with the HEPN domain occupying the outer ring.

We have, as suggested by the reviewer, revised the manuscript to present the structure as a hybrid model derived from AlphaFold predictions combined with cryo-EM density, as the map of this massive assembly shows a gradient of resolution—a feature commonly observed in multi-subunit complexes. We now use more cautious language when discussing domain orientations and interfacial residues, and we explicitly note the resolution limits of the HEPN density. Finally, while we attempted cross-linking mass spectrometry, the complex dissociated at lower concentrations, and at higher concentrations intra- (Gp77:RAZR) versus inter-complex (Gp77:RAZR with uncomplexed Gp77) cross-links could not be distinguished. For these technical reasons, we did not pursue this approach further.

Minor points

2) The data demonstrating RAZR catalyzes degradation of mRNA, tRNA, and rRNA are convincing. Although not necessary, it would be interesting if the authors could perform their RNA-seq analysis during phage infection as a possible experiment to help determine if one of the RNA targets may be more important in vivo for phage inhibition.

We have now performed RNA-seq analysis during phage T7 infection - the new data have been added as Fig. 5H and Fig. S12E-F, and are discussed on lines 377-383. In this experiment, we observed cleavage of T7 mRNAs at 20 and 30 min post-infection in cells producing RAZR, and we also observed cleavage of 16S and 23S rRNAs on a similar time-scale. Combined with the RNA gel analyses presented in Fig. S12B-C, our results indicate that upon activation RAZR likely cleaves mRNAs, tRNAs, and rRNAs during phage infection, rather than targeting any one of these classes of RNA.

3) Related to point 2, in addition to the nicely presented reactions with and without Gp77 addition, the authors should consider testing RAZR active-site mutations in one of the RNA degradation assays in Figure 5 or Figure S10.

Figures 5E and 5F have been updated to show that RAZR(H154A):Gp77 complex that contains the RAZR active site mutation does not cleave rRNAs or tRNAs, whereas the wild-type complex generated by mixing the purified proteins does cleave rRNAs and tRNAs.

4) Please include DALI z-score or another measurement of structural similarity to help compare RAZR with well-characterized HEPN domains.

RAZR has a DALI Z-score of 8.7 when compared to a known HEPN toxin (PDB 7AE8). This has now been added to the manuscript (see line 90).

5) Fig. 1E - It would be helpful to include more information about genomic location or conservation of gp77 from SECphi27. Also, does SECphi27 have a separate portal protein that forms a similar ring structure? Is it possible that gp77 is the portal protein for SECphi27? Do phages ever encode both of these RAZR triggers?

Data on the genomic location and conservation of Gp77 from SEC ϕ 27 have been added as Fig. S1H and discussed on lines 106-108. As noted in the revised manuscript, SEC ϕ 27 encodes a portal protein Gp52 that is structurally similar to the portal proteins from T3, T7, and SEC ϕ 18 (Fig. S2C). However, this portal protein Gp52 was unable to activate RAZR (Fig. S2F), likely because the regions that are important for interacting with Gp77 differ significantly in this portal protein. Although we showed that SEC ϕ 27 only uses Gp77 as the activator for RAZR, it is possible that some other phages could harbor variants of both proteins that can function as triggers.

6) It is not clear from the model figure how the authors envision RAZR transitions from a linear-like filament to a ring-like assembly upon phage recognition. No further experiments are necessary here, but do the authors envision this process is through direct transition of filaments wrapping around a phage target, or is it that inactive filaments undergo spontaneous disassembly and subunits of RAZR can reassemble as rings around a target?

We thank the reviewer for this excellent question. Our 2D class averages of RAZR alone revealed linear filament-like assemblies with noticeable curvature, and similar curved arrangements were also observed in several 2D class averages of the RAZR:Gp77 dataset. These findings suggest that a direct transition to curved filaments wrapping around a phage target could represent a plausible mechanism, as RAZR mutations that reduce the oligomeric state abolish function. However, because this aspect of RAZR activation remains speculative, we have not attempted to distinguish this model from alternative mechanisms of activation in the manuscript. This is an exciting area for future investigation.

I hope the authors will find my comments useful, thank you for the opportunity to read this exciting manuscript.

Philip Kranzusch

Referee #2 (Remarks to the Author):

Zhang et al. here study the previously identified (now renamed) RAZR system of phage immunity in bacteria. They here show that this system activates in response to two distinct phage components. The first, GP77 from phage SECP27 is a poorly characterized protein that might be involved in recombination. The second class are portal proteins that inject phage DNA into the phage capsid; three examples are shown, GP8 from phage T7, GP31 from phage T3, and GP21 from phage SECP18. Both of these types of proteins are known to form roughly 170 angstrom rings. RAZR forms an oligomer linear curved arc in its inactive state, but remarkably it circularizes around its target protein rings with its zinc finger domain on the interior interacting with the target. This large oligomer now correctly orients the RAZR exterior HEPN domain such that it activates RNase enzymatic activity. This degrades tRNA, rRNA, and mRNA, causing bacterial death that prevents phage replication, and does not degrade DNA. Point mutations are shown in multiple different interfaces and the linker to disrupt the function. They show specificity of this system by examining RAZR from *E. coli* compared to *Klebsiella* and perform chimeric swaps that show that each RAZR is able to identify specific phages.

Overall, the authors present a remarkable discovery and their conclusion is definitively proven by the experiments. From my knowledge, the specific structures or molecules or events that activate anti phage defense are often uncertain. The authors definitively identify two phage proteins that form structures that activate RAZR. One of the biggest surprises for me is that RAZR detects two distinct phage proteins that have no amino acid similarity. This detection is accomplished by looking for the size and shape of these ring forming proteins as the “pattern” that is recognized. I am not aware of mammalian systems that detect broad classes of non-self molecules by their size and shape, so this is a novel discovery. On the other hand, the RNase enzymatic activity of RAZR is quite reminiscent of many nucleic acid degrading enzymes in mammalian systems, perhaps most notably RNase L that similarly degrades RNA during viral infection. This RAZR system is also unique in that it can detect active phage replication and causes bacterial cell death and has exquisite specificity in contrast to other anti phage systems that may detect bacterial stress (e.g. loss of ATP activating OLD nucleases) as a less specific marker. I have a few comments to possibly improve the paper.

Major comments

1. The one confusing aspect of the model is that RAZR appears to have little specificity for the exact amino acid sequence because it detects both GP77 and the portal rings (that

have no sequence identity). Yet RAZR simultaneously is sensitive to point mutations in its targets, arguing that it is very specific for the amino acids. The data seems very definitive that both of these are true, but how can this be the case? My thought is that the RAZR interface with GP77 or portal proteins is very weak in each individual monomer. If so then RAZR could have many weak interactions with many proteins. Then the interaction only becomes strong if multiple targets oligomerize, creating a large interaction interface. Then once the circle of RAZR is complete, the last interaction to link the first RAZR to the last to complete the circle provides another strong interaction interface (assuming that the RAZR:RAZR interface is strong). The authors may or may not agree with those ideas, but regardless they need to address the point of simultaneous nonspecificity and amino acid substitution sensitivity in the discussion.

RAZR likely relies on distinct (but potentially overlapping) sets of residues to interact with Gp77 and the portal protein with some level of specificity. Supporting this idea, we have identified residues that are specific for recognizing Gp77 but not the portal protein (Fig. 3B, 3D). This would explain how RAZR can recognize two different proteins but with individual substitutions still capable of disrupting the interactions. Additionally, we agree with the reviewer's idea that each monomer of RAZR may not have a particularly strong interaction with a monomer of Gp77 or portal protein, but that oligomerization creates a large and strong overall interaction interface. We have expanded on these ideas in the Discussion (see lines 428-434 of the revised manuscript).

2. The authors are concluding that RAZR attacks tRNA, rRNA, and mRNA. However, a single attack on one of these species would be sufficient to halt phage replication and kill the bacteria, so it does not have to be all three equally. In particular, rRNA are generally folded into ribosomes and might be quite inaccessible. The more physiologic experiments are placed in the supplement in Figure S10b-c, where bacteria are infected with phage. However, the degradation of tRNA (Figure S10b) and rRNA (Figure S10c) is unimpressive. This leads me to speculate that the true target are mRNAs, which would be the best antiviral target in my opinion. Figure S10d shows impressive degradation of *in vitro* transcribed mRNA. Figure 5G shows impressive effects in *E. coli* bacteria upon the mRNA pool. I think this can be resolved by one additional experiment, which is to repeat Figure S10b-c and probe additionally for a few specific mRNA species by northern blot, such as those shown to be potential candidates in Figure S10d.

As suggested by this reviewer and noted above as response to review 1, we have now performed RNA-seq analysis during phage T7 infection with the data added as Fig. 5H and Fig. S12E-F, and discussed on lines 377-383. In this experiment, we observed cleavage of several T7 mRNAs at 20 min or 30 min post-infection in cells producing RAZR (Fig. 5H and S12E), indicating that mRNAs are targets of RAZR during infection. We also observed cleavage of rRNAs during phage infection in this RNA-seq experiment (Fig. S12F). Combined with our gel analysis in Fig. S12B-C and *in vitro* studies (Fig. 5I), we think that upon activation during phage infection RAZR cleaves single-stranded regions of mRNAs, tRNAs, and rRNAs.

3. Figure 3g-h and S8a-c. The experiments showing that the RAZR dimer/dimer interface is important carry a caveat that these mutations could cause folding problems in the active site, resulting in an enzymatically dead protein. I cannot think of a counter screen that would demonstrate that the point mutants are simply not at all capable of folding. If these proteins could still interact with GP77 or portal proteins by immunoprecipitation, this would show some functionality to the mutant protein. This should work for the D179/180A mutant because it still appears to form a half sized oligomer (Figure S8c).

Our western blot of protein levels in Fig. S10B showed that the mutant proteins were produced to similar levels as the wild-type protein, suggesting they are not trivially unfolded and consequently degraded. In addition, the RAZR(D179A/D180A) mutant still assembled into an oligomer (Fig. S10C), further supporting the notion that this protein is not globally unfolded. We also consider it unlikely that these mutations specifically disrupt the active site, as the substituted residues are located on the opposite face of the HEPN domain (now indicated in Fig. 3G). As suggested, we also performed co-immunoprecipitation experiments in cells co-producing Gp77-HA and either the V21W or D179A/D180A variants of FLAG-tagged RAZR. We did not observe binding of Gp77 to either mutant (data shown below). Because Gp77 binding occurs in the context of higher-order RAZR assemblies, disruption of oligomerization would be sufficient to reduce binding affinity or complex stability. Consistent with this interpretation, we observed a similar phenotype for the RAZR(Δ AGN) mutant (Fig. 3I-J and S10D-G), which also impaired oligomerization and abolished Gp77 binding. Together, these results suggest that proper oligomerization of RAZR is critical for recognition of its phage-encoded trigger.

4. The authors state that RAZR does not recognize the portal protein of the SECP27 (while simultaneously recognizing GP77 of this phage). It seems odd that a phage where the portal was not detected would simultaneously have another protein that is detected. Are there GP77 homologs in the other three phages, and have these been tested for a RAZR response? This should be discussed.

There are no Gp77 homologs in the other three phages (T3, T7, and SEC ϕ 18), likely because these phages belong to different families that are distantly related to SEC ϕ 27 - this is noted on line 127 of the revised manuscript. Gp77 is conserved in SEC ϕ 27-related phages (see

new Fig. S1H) where it, rather than the portal protein, is the key trigger. Importantly, Gp77 is also an essential protein, so in all cases RAZR is recognizing an essential component of the phage. This is noted in the Discussion on lines 423-427. It is possible that there are some other phages that encode both Gp77 and a portal protein that are detected by RAZR, but we feel it is too speculative to note this particular point in the paper.

Minor comments

1. I think the authors did not comment much about the fact that GP77 is a 24-mer, whereas the portal proteins are 12-mers. Is 12-fold symmetry the key to RAZR function? That its targets must have 12-fold (or multiples thereof) symmetry in order for RAZR to oligomerize around them?

We thank the reviewer for this insightful comment. We have now highlighted in the manuscript the symmetry mismatch between the 12-fold RAZR outer ring and the 24-fold Gp77 inner ring (lines 184-187). As noted above, mutants that form smaller RAZR oligomeric assemblies fail to associate properly with Gp77, suggesting that complete RAZR assembly is required to satisfy binding and form a stable complex. Whether 12-fold (or multiples thereof) symmetry is a general requirement for all RAZR systems remains to be determined. However, as the reviewer correctly pointed out, at least in the case of an unrelated phage protein we tested, the trend that proper oligomeric assembly of RAZR is required for complex formation appears to hold true.

2. The authors claim that RAZR targets are of a discreet size and that no endogenous complexes would be the same size and shape. With this in mind, there are many polymeric structures in bacteria that form circular structures. It should be easy to look at the known sizes of some of these to determine if they are potential RAZR targets that would need to not be detected, or whether all the potential targets are not encoded by bacteria that encode RAZR. I can think of several ring-like bacterial structures that would be worth considering: the cytosolic components of T3SS, flagellar basal body, T4SS, T6SS. In fact, T6SS has a whole circular complex that crosses the width of the bacterial cytosol. I am sure there are many more that are known as well. It may be worth commenting upon.

We thank the reviewer for raising this important point. Our mutational analyses of the RAZR:Gp77 complex demonstrate that both correct higher-order assembly and specific intermolecular contacts are required for activity. These features likely contribute to recognition specificity and minimize the risk of spurious interactions with unrelated bacterial complexes. In addition, differences in the dimensions of known bacterial ring-like assemblies may provide another level of discrimination, perhaps suggesting that RAZR has evolved mechanisms to distinguish self from non-self structures. The *E. coli* strains we worked with do not have T3SS, T4SS, or T6SS, but they do encode a flagellum. Notably, the flagellar basal body has a diameter of ~260 Å which is significantly larger than the ~170 Å diameter of Gp77 and the portal protein. Additionally, the basal body is largely embedded

within the cell envelope and so likely not present in the cytoplasm. We now comment on this point in the revised Discussion (see lines 421-423).

3. It seems fairly easy to evade RAZR with point mutations. Can the authors comment upon why these phages have not already evolved to evade RAZR? Are there RAZR negative *E. coli* that are the natural hosts for these phages? Also, are there phages that infect the strain of *E. coli* that encodes RAZR? Such phages should have evolved to evade RAZR detection. Did the authors survey of phages include others that have portal proteins or GP77 homologs but against which RAZR fails? If so, these would be worth mentioning.

This is an interesting and important idea that comes up in the study of virtually all escape mutants. One possibility is that the escapers isolated have subtle defects that would put them at a competitive disadvantage in the wild if those phages have not been recently subject to RAZR-based defense. In regards to the question of whether there are related phages that can infect *E. coli* encoding RAZR: as the figure below shows, the T7-related phage Bas64 (and to a lesser extent Bas67 and Bas68) can indeed overcome RAZR. Each of these phages have a portal protein similar to that from T7. This raises the intriguing possibility that they might encode inhibitors of RAZR, which, of course, enable escape without mutation of trigger proteins. We feel these results, which we intend to pursue in the future, are too speculative to include in the current manuscript.

4. It should be possible to generate mutations in GP77 or a portal protein that prevents ring formation. This should prevent RAZR activation, and would support the conclusion that the ring is detected. I mention this because it would be very uncomfortable for the authors if a monomeric GP77 or portal was subsequently found to activate RAZR. That said, I think that is very unlikely that monomeric proteins would interact with RAZR, and given that it is not essential for the current paper.

We agree with the reviewer that testing whether disruption of Gp77 oligomerization prevents RAZR activation would provide valuable support for our proposed model. We attempted to generate mutations at the predicted oligomerization interface of Gp77; however, these constructs did not alter oligomerization. Designing such mutants is not trivial, as it requires considerable effort to identify variants that affect oligomeric state

without compromising protein stability. Therefore, while we could not experimentally evaluate this point, our biochemical and structural data strongly support the conclusion that ring formation is essential for RAZR activation.

5. Figure 2E, is it possible to annotate the active site residues so that the reader can see where they are and how they move (instead of a bar) that is only showing the AF3 predicted active site. It is hard to see precisely.

We thank the reviewer for this helpful suggestion. However, because the HEPN domain is resolved at lower resolution, and following Reviewer #1's recommendation, we chose not to annotate specific side-chain positions, as doing so could give a misleading impression of confidence in their placement. Instead, we have indicated the active-site region more generally to reflect the current resolution limits.

6. Figure 4E. I assume that the whole phage was sequenced and that only one mutation was found, and it was the one described for Figure 4E. If so, please state that.

These phages have been sequenced and T7^{Ec} clone 1 only has S223P mutation in Gp8. T7^{Kv} clone 1 and 2 have other mutations in its tail fiber protein or the tail tubular protein, in addition to the mutation in Gp8. We don't think the mutations in tail fiber or tail tubular protein leads to escape, as Gp8 is the only gene mutated in all phage clones and we showed that Gp8 is sufficient to activate both RAZR homologs on its own (Fig. 4D). This information has now been added as table S2.

7. Based on the P22 portal structure and its predicted position with the phage capsule published for example by Cingolani (doi: 10.1038/ncomms14310), would the authors predict that RAZR is incorporated into forming phage heads? Or if the portal complex assembled first and then the capsid assembles around it or dock onto it, would RAZR prevent the assembly of the capsid and portal into a phage head?

We don't think RAZR is incorporated into the forming phage heads, because if RAZR affects proper phage head formation due to binding to the portal protein, then we would expect any RAZR proteins capable of binding to the portal protein to disrupt phage propagation. However, we showed that the active site mutant (H154A) of RAZR does not have any effect on plaquing efficiency of phage T3, T7, and SECφ18 (new Fig. S2A), but it can bind to the portal proteins of these phages (Fig. S2E). Therefore, we think the phage-defensive activity of RAZR comes from its RNase activity instead of disrupting phage assembly.

8. I am confused about what Gp77 does, and why it forms a ring. From Line 161 the authors mention that the ring structure of Gp77 is similar to another phage, and that this ring may be important for circularizing the genome. Could the authors provide some speculation on why a ring structure might be needed to circularize the genome? No reason for this comes to my mind as a reader immediately. And this property seems to be the key to RAZR function. The explanation hypothesized by reference 28 published in 1983 could be incorrect. In contrast, for the portal proteins it is very apparent from prior work why they

form the ring structure as part of their structure in phage assembly so that is a very satisfying rationale for their detection by RAZR.

We appreciate the reviewer's comments. The precise role of Gp77 in phage biology is not fully established, and we agree that the prior hypothesis from the reference may not provide a complete explanation. However, structural parallels with other ring-shaped proteins involved in DNA metabolism suggest possible functional roles. In particular, we note that several single-strand annealing proteins, such as the human Rad52 protein, assemble into ring structures with positively charged grooves that interact with ssDNA and promote annealing (doi: 10.1073/pnas.212449899). By analogy, we speculate that Gp77 and related Erf-family proteins may similarly exploit ring formation to facilitate single-strand annealing and recombination during genome circularization. Such a geometry could allow multiple weak DNA contacts to be simultaneously engaged, thereby stabilizing intermediates in recombination or genome circularization, but we felt it was too speculative to include this in the manuscript.

9. Figure 2B. I suggest using the same magnification for each panel, thus allowing the reader to directly compare the size, for example, of the rings of Gp77 to the size of those seen in the RAZR-Gp77 structure. This would be a more intuitive presentation (sacrificing resolution of the Gp77 alone structure, but to the same extent as is lost by shrinking the co-structure).

We thank the reviewer for this suggestion. While we have generally tried to keep the scale consistent across figures, it is not possible to achieve this in all cases. To aid interpretation, we have provided scale bars in Fig. 2B to allow the reader to accurately distinguish size differences. Additionally, it is worth noting that during cryo-EM data processing, the diameter of the particles is explicitly considered when defining the extraction box size. This ensures that structural features of interest are captured consistently across datasets, regardless of magnification differences.

10. Figure S7D is remarkable in contrast to Figure 3C. Is there space to move it to the main figure? I wonder if Figure 3B could be moved to the supplement to make space, as the take home for me from 3B and 3C was the same in that the mutations seem to all have a big effect. In contrast, 3C vs S7D was quite impressive showing specificity.

As suggested, previous Fig. 3B has been removed to the supplement as new Fig. S9C, and previous Fig. S7D has been moved to the main figures as new Fig. 3D.

11. In Figure 4e, it was intriguing to see that the chimeric protein on the far right is stronger in defense than the E. coli RAZR against T7-Kv clones 1 and 2. Can the authors speculate as to why the chimera is superior in a way that appears independent of the ZFD?

We think there could be many reasons why this chimeric RAZR protein provides stronger defense to both wild-type T7 and T7-Kv clones. It's possible that this chimeric RAZR is expressed to higher levels (as it contains the native promoter from RAZR^{Kv}), or the chimeric

protein is more stable. It's also possible that there are differences in the catalytic efficiency of the HEPN domains of RAZR^{Ec} and RAZR^{Kv}, and that affects the phage defense phenotype.

12. You may want to mention the similarities to RNase L in the mammalian system in your discussion.

We thank the reviewer for noting the similarity to RNase L in mammalian innate immunity and we now note this striking similarity in the revised Discussion (lines 439-442).

- Ed Miao

Referee #3 (Remarks to the Author):

I co-reviewed this manuscript with one of the reviewers who provided the listed reports.

Referee #4 (Remarks to the Author):

In this study authors provide biochemical and structural characterization of the RAZR antiviral defense system. Most notably, they show a novel activation mechanism of antiphage defense that relies on the RAZR recognition of circular assemblies of phage proteins through the RAZR ZFD domains triggering the transition of linear RAZR filaments into ring-shaped structures and activation of HEPN RNase. Activation of antiviral defense system and discrimination of the self vs non-self through the shape recognition certainly adds a new dimension to the currently known mechanisms for sensing phage infection by antiphage defense proteins.

Overall, the study is well thought out, technically sound and well presented, providing compelling evidence for the proposed mechanism.

Points that should be addressed by the authors are listed below.

1. An important question that remains open is how the formation of the circular structure on the circumference of the gp77 disc leads to the activation of the RNase activity of the HEPN domains. The overlay of the linear-like AlphaFold3 prediction on a fragment of the circular structure is presented in Fig. 2E. However, the overlay is based on the ZFD domains, not the HEPN domains, making it difficult to judge the overall similarity between the AF3-predicted HEPN structure and the experimentally modeled HEPN domain. It is also unclear whether the transition from the linear AF3-predicted filament to the experimentally observed ring-shaped structure results in the rearrangement of RAZR (HEPN) homodimers relative to each other or to RAZR (HEPN) domains within the dimers, leading to the rearrangement/assembly of the catalytic centers at the HEPN domain interfaces.

We thank the reviewer for this thoughtful question. As noted, the HEPN domains in our cryo-EM reconstructions are resolved at comparatively low resolution ($>7 \text{ \AA}$), which precludes confident model building. Consequently, we cannot determine whether the transition from the linear AlphaFold3-predicted filament to the experimentally observed ring involves rearrangements of the HEPN catalytic centers. What we can state with confidence is the relative positioning of the RAZR zinc-finger domain and the HEPN domain. Continuous density connecting these two domains supports the overall orientation of the HEPN domain with respect to the ZFD. For this reason, our overlay in Fig. 2E was based on alignment of the ZFD domain, which is resolved at higher resolution, rather than the HEPN domains, which are represented by weaker density.

We have revised the manuscript to clarify these points and to avoid suggesting that the current data resolve the detailed arrangement of the HEPN catalytic centers. Instead, we now emphasize that our model supports the general placement and outward orientation of the HEPN domains, as well as their orientation relative to the ZFD domains.

2. In addition to the expected HEPN-HEPN domain interactions within the RAZR primary dimers observed in the structure, the primary HEPN domains seem to form additional interaction surfaces between the primary dimers. How large are these interaction surface areas? Which structural elements that form these interfaces could be reliably modeled into the density, and how does this overlap with the AlphaFold3 model shown in Fig. 3G, which was used to design interface-disrupting point mutations?

We thank the reviewer for raising this important point. As mentioned above and in the manuscript, the limited resolution of the HEPN ring prevents us from reliably calculating interaction surface areas between HEPN domains. Providing such estimates could create the impression that these features are resolved with high confidence, which is not the case.

What we can model is the density corresponding to the loop connecting the ZFD and HEPN domains, as well as the first α -helix of the HEPN domain (residues 76–93). The remainder of the HEPN domains could not be confidently built and were instead positioned by manual docking of the AlphaFold3-predicted model, followed by refinement in Phenix to minimize steric clashes. Accordingly, while the general outward orientation of the HEPN domains and their canonical dimeric arrangement are consistent with the AlphaFold3 model shown in Fig. 3G, the precise details of the inter-dimer HEPN contacts remain uncertain. We have revised the Results section on lines 197-203 and the relevant section of the Methods to clarify these limitations (see lines 1310-1318).

3. Authors show that activated HEPN RNase cleave various RNA molecules including tRNA. Where tRNA cleavage occurs?

Although we do not know the exact cleavage sites, we think RAZR is likely cleaving ssRNA regions of tRNAs, for example the anti-codon loop, because we showed in Fig. 5I that RAZR specifically cleaves ssRNA but not dsRNA.

4. Would the authors predict that all potential activators of the RAZR system would form rings/discs with 12-fold symmetry? The RAZR system is activated by unrelated proteins from different phages (Gp77/Erf-like protein from SECphi27, Gp8/Gp31 portal proteins from T7/T3). However, can the authors explain, based on structural predictions, why the SECphi27 portal protein and T3/T7 Gp77 homologs (if present) are not activators for the RAZR system? Can some other structural requirements (beyond the 170Å outer diameter) for the activator protein be detected based on these observations?

RAZR relies on a combination of protein interfaces and a ring structure of the appropriate diameter/curvature to sense its activators. Notably, the region where the T7 escapers were mutated is a highly divergent region when comparing the portal proteins from T7 and SECphi27 (see figure below), which might explain why the SECphi27 portal protein fails to activate RAZR despite forming a ring with comparable diameter to that of T7. Without fully characterizing the interface, we think it's too speculative to include in the current manuscript.

5. The cryo-EM processing figure (Fig. S11) does not provide the local resolution for the final C1 reconstruction obtained after symmetry expansion/3D classification. Since the obtained C1 map was used to build the atomic model of the HEPN domain (Fig. S5g), it should be deposited to the EMDB.

In the course of revision, we reprocessed our data and generated new maps with reduced anisotropy, which provided improved map quality. As a result, we no longer used the previous C₁ reconstruction after symmetry expansion for building the HEPN domain. All the models and corresponding maps, which were used for model building, have been deposited to the PDB and EMDB.

6. The ConSurf-colored RAZR sequence and AF3 model are provided in Fig. S1d and Fig. 4d. Please provide more information on the RAZR sequences that were used to generate the respective data in the Methods section.

This has now been added in the Methods section under 'Homology search and conservation analysis' (see lines 1496-1500).

7. The conserved linker (aa 66-79), which is subjected to deletion analysis (lines 278-281), should be indicated in Fig. S1d. Does the 3aa linker deletion characterized in your study alter the AF3 prediction of the RAZR filament? Does AF3 co-fold Gp77 with RAZR (say, in a 4:4 subfragment)?

The linker region has now been indicated in Fig. S1d. The 3 aa linker deletion does not alter the AF3 prediction of the RAZR filament as a tetramer, but the prediction has lower confidence (ipTM of 0.35). Additionally, we tried to co-fold Gp77 with RAZR with different copies of each protein, but AF3 was able to co-fold them with a reasonable confidence score. For example, the 4:4 subfilament has only an ipTM score of 0.23, and overall does not agree with our structure.

8. Lines 243-244: The "slightly smaller colonies" observation, referring to data in Fig. 1g, could be supported by quantification of actual colony sizes in the +glu/+ara experiments.

The colony sizes have been quantified in 3 biological replicates and are reported in Fig. S1K.

9. Showing denoised micrographs in Fig. S3f and Fig. S4a could provide a better representation of the actual particles observed in the respective samples.

We thank the reviewer for this suggestion. The images shown in Fig. S3f and Fig. S4a are from screening datasets that were collected as single-exposure micrographs rather than dose-fractionated movies. As such, they cannot be subjected to motion correction or denoising procedures, and we therefore present them in their raw form. With a small dataset that we collected we tried to do denoise micrographs but they did not provide a better visualization than what we have so we therefore decided to keep them as they are.

10. Throughout the Methods section, the oligomeric states of the proteins/complexes to which the listed concentrations refer should be indicated.

The oligomeric states of proteins/complex have now been added in the Methods.

Referee #5 (Remarks to the Author):

I co-reviewed this manuscript with one of the reviewers who provided the listed reports.